# Evidence of megathrust earthquakes and seismic supercycles in subtropical Japan from millennia-old coral microatolls

Sophie Debaecker [1,2,9] ✉, Nathalie Feuillet [1], Kenji Satake [2,10], Kohki Sowa[3,11], Masaki Yamada [2,12], Tetsuro Sato[2,13], Mamoru Nakamura[4,5], Atsushi Watanabe[2], Ayaka Saiki[2], Jean-Marie Saurel [1], Giovanni Occhipinti [1], Tsai-Luen Yu[6] & Chuan-Chou Shen[7,8]

Megathrust earthquakes in subduction zones often go unreported because they are rare and the historical record is short. On the Ryukyu subduction zone of southwestern Japan, unlike neighboring Nankai Trough, the history and future potential of great interplate earthquakes are not well known. While the geodetic measurements on the islands suggest that the plate coupling is very weak, recent observations of slow seismic events as well as offshore geodetic measurements imply the presence of coupled patches along the megathrust. Furthermore, the historical and geological studies indicate evidence of great tsunamis. Here, we use fossil microatolls in Ishigaki island to reconstruct the relative sea level in the Holocene. The coral record reveals several relative emergence episodes clustering between 5-4 and 3-2 thousand years ago (ka). Elastic modeling shows that the observed motions can correspond to coseismic uplift associated with megathrust earthquakes. The clusters of megathrust events suggest possible supercycles of earthquakes with a recurrence interval of more than 2 ka. Such results imply a strong seismic hazard for the upcoming centuries. The devastating 1771 Meiwa earthquake and associated tsunami may mark the onset of the most recent seismic supercycle.

The seismic behavior of the Ryukyu subduction zone in Japan (Fig. 1) is a matter of debate. Geodetic data in the last decades tend to indicate that the plate interface is weak (coupling rate lower than 5%), and that the convergence is accommodated by aseismic creep[1,2] or slow slip events[3,4], large megathrust earthquakes being unlikely[5,6]. However, two major earthquakes occurred on April 24, 1771 (Meiwa) in the southern Ryukyus and on June 15th, 1911 offshore Kikai island (Fig. 1), although their source remains debated. Both events generated large tsunamis with wave run-ups reaching tens of meters for the Meiwa tsunami[7,8] and up to 5 m for the Kikai event[9]. In addition, geological records provided evidence for prehistorical large tsunamis in the southern Ryukyus[7,8,10–14] and for a sudden coseismic coastal uplift at ~0 B.C. in

[1]Université Paris Cité, Institut de Physique du Globe de Paris, CNRS, Paris, France. [2]Earthquake Research Institute, The University of Tokyo, Tokyo, Japan. [3]Japan Agency for Marine-Earth Science and Technology, Kanagawa, Japan. [4]Faculty of Science, University of the Ryukyus, Nishihara, Okinawa, Japan. [5]Disaster Prevention Research Center for Islands Region, University of the Ryukyus, Nishihara, Okinawa, Japan. [6]Marine Ecology and Conservation Research Center, National Academy of Marine Research, Kaohsiung, Taiwan, Republic of China. [7]High-Precision Mass Spectrometry and Environment Change Laboratory (HISPEC), Department of Geosciences, National Taiwan University, Taipei, Taiwan, Republic of China. [8]Research Center for Future Earth, National Taiwan University, Taipei, Taiwan, Republic of China. [9]Present address: LIttoral ENvironnement et Sociétés (LIENSs), UMR 7266 CNRS, La Rochelle Université, La Rochelle, France. [10]Present address: Department of Earth Sciences, National Central University, Taoyuan, Taiwan. [11]Present address: KIKAI Institute for Coral Reef Sciences, Kikai, Japan. [12]Present address: Department of Geology, Faculty of Science, Shinshu University, Matsumoto, Japan. [13]Present address: Marine Core Research Institute, Kochi University, Kochi, Japan. ✉e-mail: debaecker.sophie@gmail.com

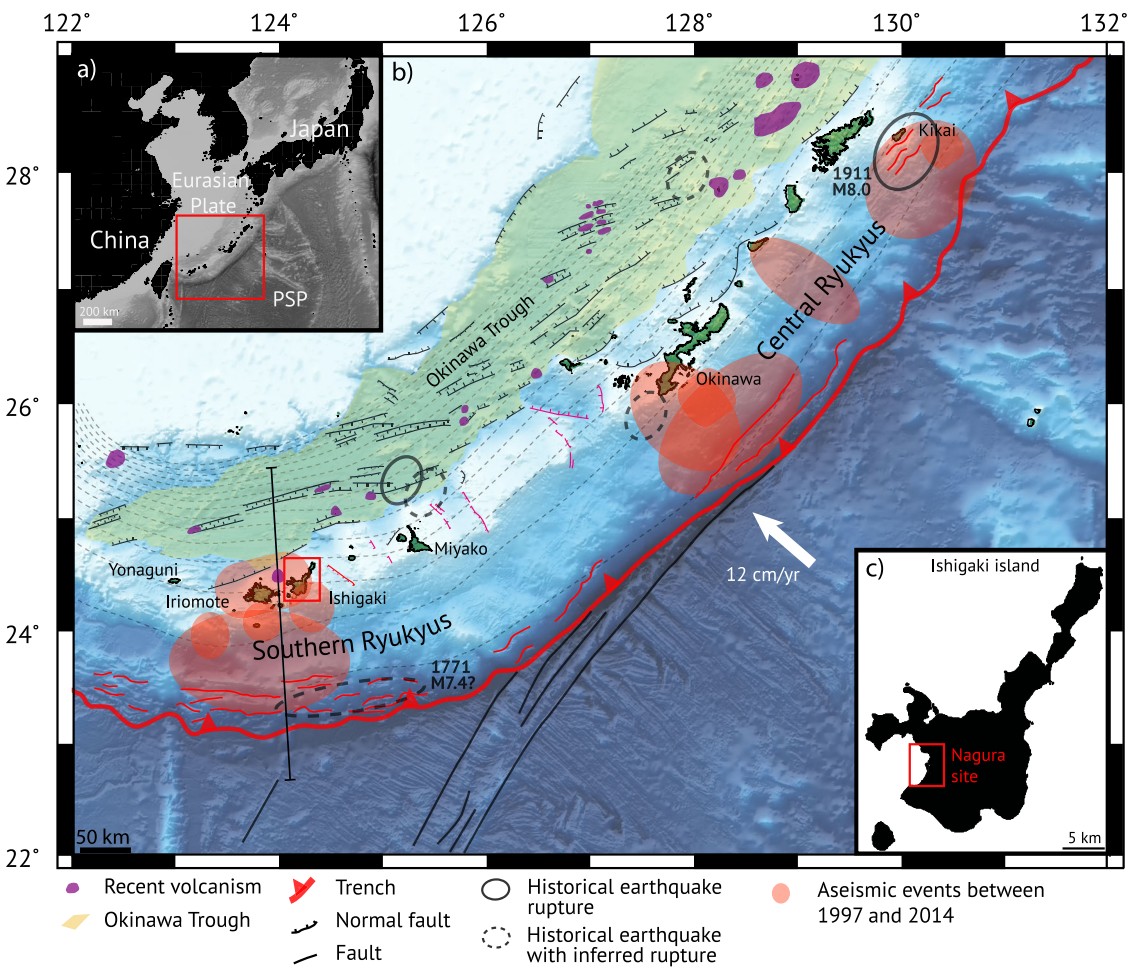

**Fig. 1 | Geodynamic setting of the Ryukyu archipelago.** Upper inset: Location of the Ryukyu archipelago. PSP Philippine Sea Plate. Main figure: Tectonic setting in the Ryukyus. Pink faults are related to the arc-parallel extension. Splay faults are in red. Straight black line is location of the cross-section in Fig. 5. Faults and volcanism are from refs. 10,38,75,76. Historical earthquakes are from refs. 43,77. Aseismic events are from refs. 4,49. Plate convergence rate is from ref. 70. Dashed lines are isodepths of the megathrust every 10 km from ref. 37. Lower inset: Location of Nagura site in Ishigaki island. Figure produced using the Generic Mapping Tool software (https://www.generic-mapping-tools.org/).

Ishigaki island[15,16]. Moreover, interseismic strain accumulation on patches of the subduction interface has been inferred for the last one to two centuries from modern microatoll coral analysis[17,18], and over the past decades from gravimetric or offshore geodetic data[19,20]. A seismic coupling of 20% has been estimated for those patches[12].

In this work, we aim to better understand the seismogenic behavior of the Ryukyu subduction zone and to bridge the gap between short-term geodetic series and paleo-tsunami record. We focus on an exceptionally long and unique record of centimetric-scale sea-level changes from fossil microatolls in Ishigaki island. We found, measured, sampled and dated several generations of fossil and living microatolls in Nagura site. We show that the development of these corals is controlled by seismic supercycles and permanent deformation. This study provides an example of multiple earthquake cycles over the past five millennia in a subduction zone, based on coral microatolls analysis. It challenges previous inference from inland geodesy that this subduction zone is mainly aseismic.

## Results

### Nagura microatolls measurement and sampling

The Nagura site extends over a 3.4 km² area, 130 km far from the Ryukyu trench in the southwestern part of Ishigaki island (Fig. 1). We found multiple generations of in-situ *Porites* fossil corals within a mangrove on the large alluvial fan of the Nagura river, and on a sandy tidal flat forming a bay open to the ocean (Fig. 2a, b). Microatolls have

been previously documented around Ishigaki[21] but to our knowledge, Nagura is the only site where fossils specimens have been observed[22].

Microatolls are massive corals that grow radially from a single polyp in a tropical intertidal zone. Once the coral reaches its highest level of survival (HLS), its uppermost part dies. The lowermost part keeps growing below the HLS, thus recording annual relative sea-level (RSL) changes with a centimetric precision (see Method).

Based on field observations and using high-resolution drone imagery, a total station and real-time kinematic Global Navigation Satellite System (RTK-GNSS), we measured the position and elevation of 73 fossil microatolls, 1–11 m wide in diameter, up to 40 cm thick, as well as a living coral located 1.2 km offshore (Fig. 2c). Using a hydraulic chainsaw, we sampled seven of the largest and best-preserved fossil microatolls by extracting a ~10-cm-thick slab from which 1–2-cm-thick slices were X-rayed, CT-scanned and interpreted. In addition, we sampled the outer part of 12 specimens from various generations of microatolls over the entire site for U/Th dating[23] (see Method and Supplementary Tables S1, S2). Photos of the sites, slabs or samples of each microatoll, X-ray interpretation and HLS curve are presented in Supplementary Information S1, S2.

### Relative sea-level changes over the past 5ka

The upper surfaces of the microatolls, whose altitudes have been measured with a precision of ±5 to ±7 cm, define 23 plateaus that

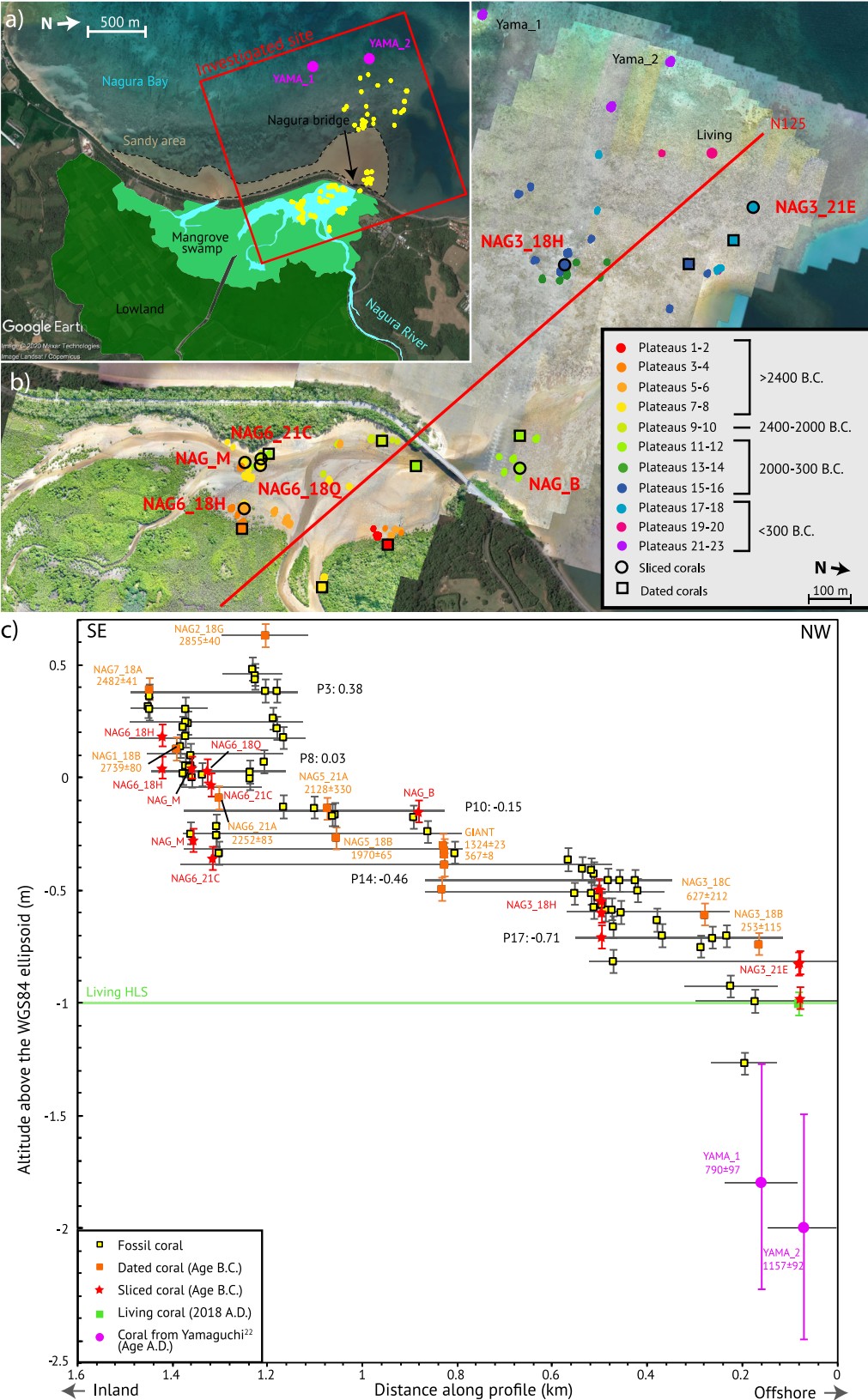

**Fig. 2 | Nagura site. a** Geomorphology of Nagura site with main geomorphic elements superimposed on Google Earth screenshot (© 2020 Maxar Technologies, CNES/Airbus, Google Earth), Yama_1 and Yama_2 are corals studied by Yamaguchi[22]. Red rectangle: location of (**b**). **b** Location of microatolls from total station and RTK-GNSS survey, superimposed on Google Earth screenshot (as in (**a**)) and drone imagery. Red line: location of the cross-section in (**c**). **c** Plot of the absolute altitudes of the corals as measured with RTK-GNSS and total station. Uncertainties on a single coral related to HLS measures and combination of total station and GNSS are of ±5 to ±7 cm (see Method).

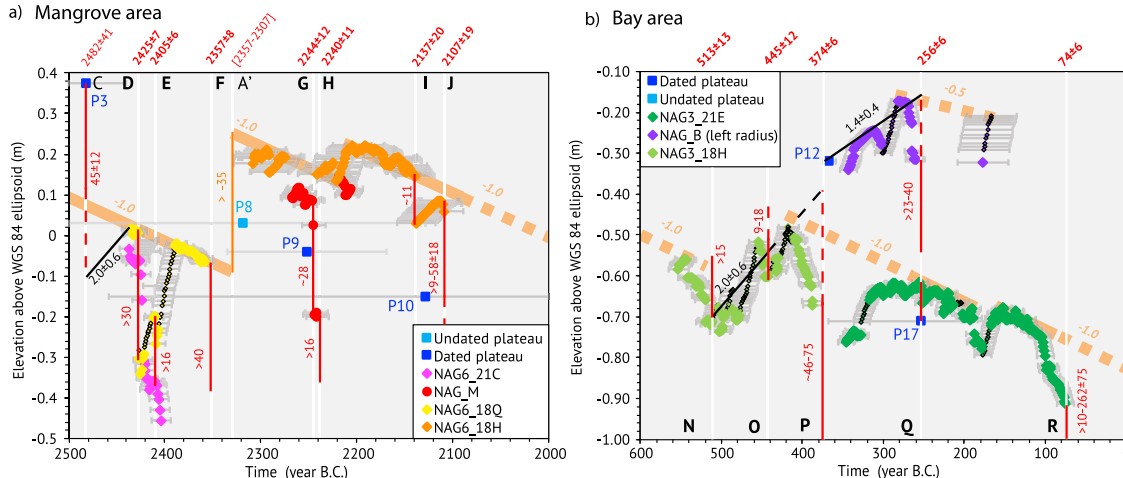

**Fig. 3 | Relative Sea-Level reconstructions from coral microatolls data. a** In the Mangrove area between 2500 and 2000 years B.C., and **b** in the Bay area between 600 and 0 years B.C. Initial periods of growth (Supplementary Fig. S14) are not considered. We used the altitude and RSL records from sliced microatolls in the mangrove and in the bay with a color for each microatoll (Supplementary Information S1, S2 for details on X-ray interpretation and HLS reconstructions). Smaller black points show Highest Level of Growth (HLG) years. We used the altitudes of sampled blocks of microatolls to date several plateaus that are indicated with dark blue squares. Light blue square is undated plateau. Black and red lines with numbers are relative submergences (in mm/year, Supplementary Table S3) and drops (in cm) of the RSL, respectively, dashed lines when inferred (see Method). Thin vertical orange line shows a potential sudden RSL increase event. Thick light orange lines (dashed when inferred): centennial-scale RSL decreases. Gray lines: age uncertainties (see Method). Vertical white lines: dates of emergence events, with letters numbering the events in bold when inferred from slice analysis.

represent topographic features (Fig. 2c and Supplementary Table S3). They are first order-paleosealevels, with the highest—and generally landward—being the oldest. The overall increase in elevation and age landward supports the interpretation that the microatolls are in-situ. Two deeper fossil microatolls (Yama_1 and Yama_2) measured at −1.80 m and −2.00 m and radiocarbon-dated at A.D. 790 ± 97 and A.D. 1157 ± 92 by Yamaguchi[22] are included in our analysis (see Method).

Individual coral slabs record a complex RSL history, alternating between 1) periods of free growth below the HLS, 2) periods of growth close to or at the HLS, and 3) sudden drops or increases of the RSL (see Method and Supplementary Fig. S12). The drops were large enough to partially or entirely kill the corals. The microatolls sliced in the mangrove recorded seven sudden RSL drops with values ranging from 11 to 40 cm over approximately three centuries between 2500 and 2000 B.C. (Fig. 3a). Coeval corals (within age uncertainties) recorded the same HLS variations, indicating that the record is reliable. Microatolls in the bay recorded five sudden RSL drops, with values between 9 and 46 cm over five centuries between 600 and 0 B.C. (Fig. 3b).

These RSL drops are followed by years of HLS stability (with RSL changes at small rates of less than 10 mm/yr, Supplementary Table S4), overall indicating that they are of tectonic origin and due to earthquakes[24]. Long lasting RSL changes at small rates are promoted by interseismic tectonic motion (long term strain accumulation or transient displacements[18,25]). They likely also include a climatic component (eustatic sea-level changes or glacial isostatic adjustment, GIA) for which we have no precise information. Here, RSL contributions from GIA was inferred to be 0–0.2 mm/year for the last 4–5 ka[26,27], which is one order of magnitude smaller.

Although NAG3_21E is coeval with NAG_B, it is 30–40 cm lower and did not record RSL drops of similar magnitude. Variations of tens of centimeters in altitude among corals of the same generation have been observed elsewhere, reflecting the variability of the coral record[28]. Ponding of NAG_B coral could explain such differences, although we did not observe any traces of past geomorphic features supporting this hypothesis.

## Seismic supercycles

Overall, the RSL changes display an irregular and truncated sawtooth pattern of interseismic subsidence or uplift interrupted by sudden coseismic uplifts (Supplementary Table S5). A long-term centennial-scale RSL decrease rate of about 1.0 mm/yr is superimposed on this sawtooth pattern (light thick orange line in Fig. 3). At the mangrove site (Fig. 3a), NAG6_21C and NAG6_18Q are older than NAG_M and NAG6_18H yet lie below them. This may indicate a sudden sea-level increase of ~35 cm.

Altitudes and dating of the external parts of other microatolls (dated or undated plateau, squares, Fig. 4, Supplementary Table S2 for ages and Supplementary Information S1 for information on samples) complete the sawtooth record. We used an interseismic RSL rate of 2.0 ± 0.6 mm/yr (as determined from the best-preserved signal from NAG3_18H, Supplementary Table S4) to estimate the magnitude of the events that killed the microatolls. We consider our approach is reasonable, although variations in the rate of interseismic sea-level changes were observed at smaller time scale in the RSL records of the slices and documented in the literature[29] (Supplementary Table S3). We inferred that the ages of the undated plateaus are bounded by those of the plateaus just higher or lower in elevation (Supplementary Table S3).

In addition to the 12 major earthquakes documented from the slices (Fig. 3), at least nine additional sea-level drops with amplitudes ranging from 15 ± 43 to 262 ± 75 cm (events named with letters A to U, Supplementary Table S5) are required to explain our observations. It is striking that over a time period exceeding 1300 years, between 2000 and 700 B.C., there are almost no indication of the RSL. The RSL record is also missing between 75 ± 6 B.C. (death of NAG3_21E) and A.D. 790 ± 97 (Yama_1). This could be due to coral absence or scarcity. Only few fossil microatolls (most undated) suggest the occurrence of earthquakes during these periods (event L and dotted portion in Fig. 4). It is possible that these long periods of missing corals followed major earthquakes, i.e. with greater uplift and a longer recovery time for the reef flat, as observed in Sumatra[30]. From our reconstruction, we estimate values of 136 ± 41 and up to 262 ± 75 cm for the main events K and R (Supplementary Table S5).

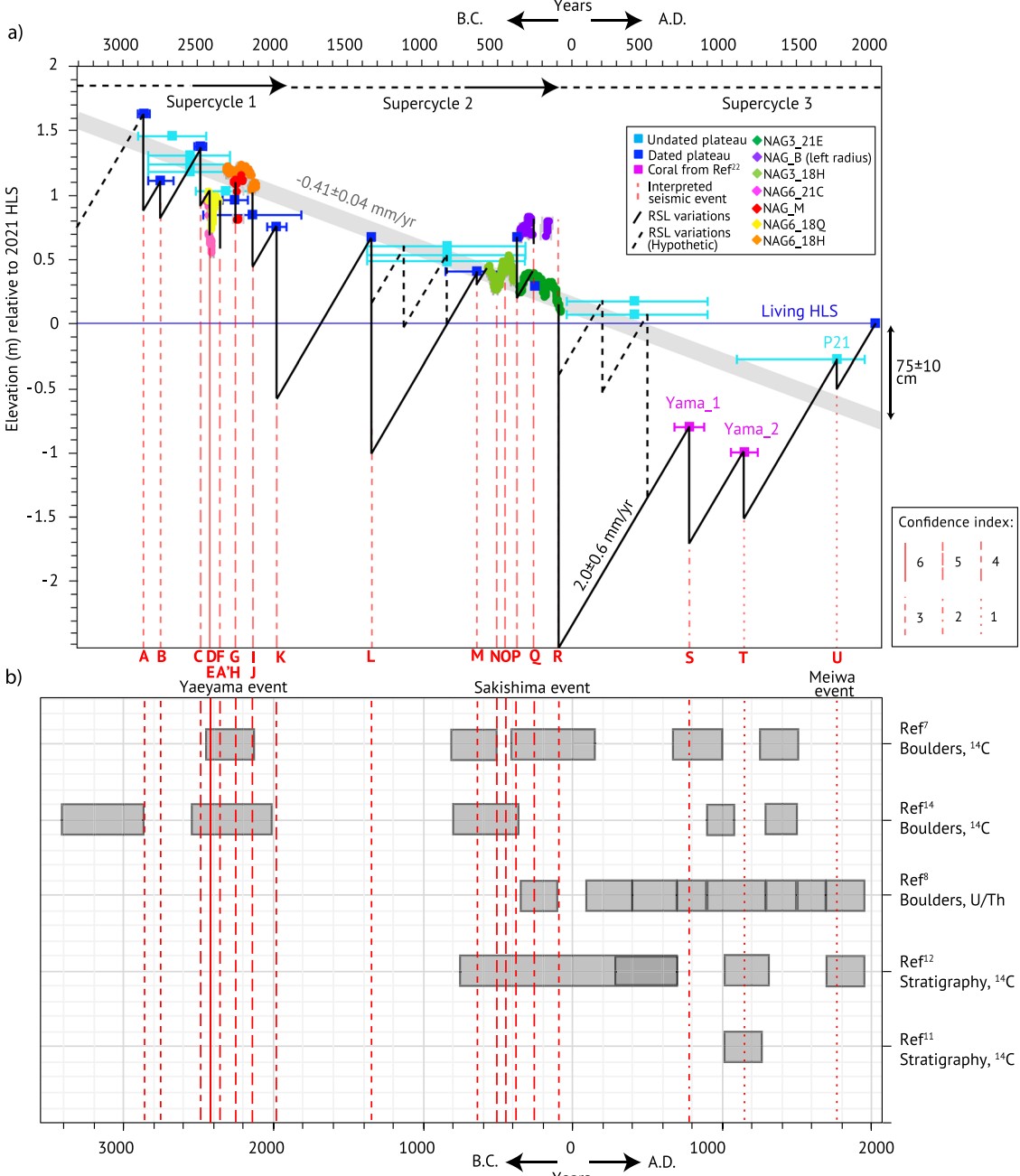

**Fig. 4 | Holocene seismic supercycles in the southern Ryukyus. a** Possible RSL reconstruction over the past 5ka in Nagura (black line) from sliced coral microatolls and other dead microatolls forming *"plateaus"* with associated age uncertainty (see Method; dark blue squares are U/Th dated corals, light blue are undated). Dashed portions are inferred, when using information from the altitude of undated plateaus. Dashed red lines: emergence events, with age and letters numbering the events. Line symbol reflects the confidence index (Supplementary Table S5). Light gray thick line is millennial emergence rate inferred from dated plateaus. Horizontal black arrows in the top part of **a** indicate the timing of each supercycle, with dashed part being the quiescence period and continuous part being the failure sequence[30]. **b** Traces of tsunami events (with names) in the Southern Ryukyus from various studies[7,8,11,12,14]. Red lines as in (**a**).

The most recent part of the sawtooth signal includes an undated microatoll marking the lowest plateau P21 (Supplementary Table S2). As its altitude is lower than NAG3_21E, it is probably more recent and may have developed between 75 ± 6 B.C and the present day. If it is younger than A.D. 1157 ± 92 (age of Yama_2), this coral may have been killed during the Meiwa event in 1711.

The coseismic motions we infer have an average amplitude of 59 cm. At a similar distance to the trench, this amplitude is comparable to that modeled from past Mw-8 class subduction earthquakes in Sumatra (e.g. between 0 to 1 m[31]). We verified with elastic models (Supplementary Information S3) that ruptures on either shallow

normal faults on Okinawa Trough or on splay faults in the forearc domain cannot account for the observed deformation. The sea-level changes recorded are most probably due to strain accumulation and release during the seismic cycle on the Ryukyu megathrust.

At first order, the sawtooth pattern characterizing the RSL signal appears to define supercycles. It shows periods of failure sequence with frequent and discrete emergence events (C-J and M-Q, Fig. 4) before larger events (K, R). Those periods are followed by periods of quiescence (characterized by coral absence or scarcity). Supercycle 1 lasted at least ~900 years, contains at least 11 earthquakes with amplitudes ranging from 7 to 136 cm and a mean recurrence time of

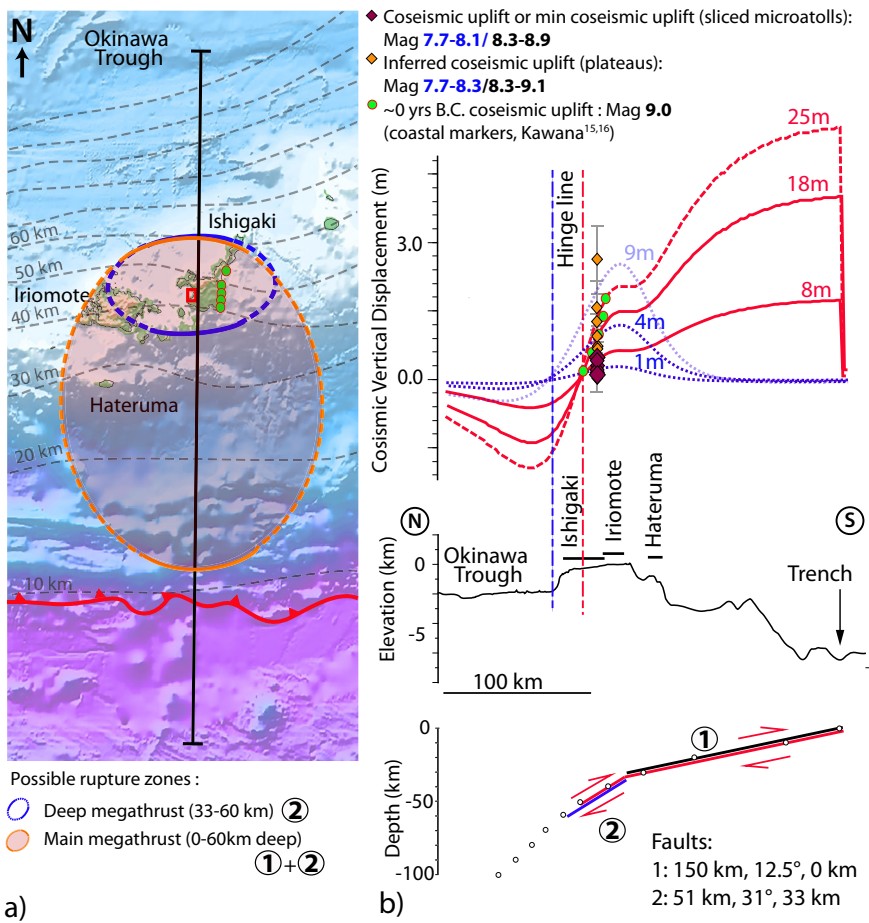

**Fig. 5 | Modeling of coseismic uplift events in Nagura (see Method). a** Map of the potential rupture zones along the plate interface (blue and orange ellipse). Green dots: 0 B.C. event data location[15,16]. Red rectangle: site location. Dotted gray lines: plate interface isodepth every 10 km (0 at the trench). Dark gray line: Location of cross-sections in (**b**). Red line with arrows: the Ryukyu trench. Ellipses are investigated potential rupture zones, with uncertain length. Panel produced using the Generic Mapping Tool software (https://www.generic-mapping-tools.org/) b) The colored points with gray error bars are inferred uplift events with their associated uncertainties (Supplementary Table S5). Red curves: vertical deformation at the surface promoted by slips between 8 and 25 m on the plate interface between 0 and 50 km depth, associated hinge line: vertical dashed red line. Blue curves: vertical deformation at the surface promoted by slips between 1 and 9 m on the plate interface between 33 and 60 km depth, associated hinge line: vertical dashed blue line. Magnitude of earthquakes corresponding to dislocations on the portion of the megathrust between 33 and 60 km and on the megathrust between 0 and 50 or 60 km are indicated in blue and black, respectively. Maximum magnitude is estimated using a length of ~300 km of the rupture zone along the southern Ryukyus, between Yonaguni and Miyako islands (Fig. 1).

~88 years during its failure sequence. It ended by the large 1970 ± 65 B.C. K event. We cannot confirm that this cycle is complete but we have not found any coral older than 2900 B.C. Supercycle 2 lasted about 2000 years and contains at least 6 seismic events of 15–262 cm of amplitude. It ended with the large 75 ± 6 B.C. R event. During the failure sequence, the mean recurrence time of earthquakes is ~110 years. The few earthquakes that occurred during the more quiescent periods at the start of supercycles 2 and 3 have a recurrence time of ~671 and 615 years respectively. This behavior was also observed in Sumatra where the recurrence time of earthquakes within supercycles is ~150 years, and of several hundred years during periods of quiescence following a major earthquake[32,33]. In this study, supercycles are characterized based on a single paeloseismic site only. To improve their description, it would be necessary to assess the spatial extent of the seismic ruptures documented here[33], which is beyond the scope of this paper.

## Source of RSL: elastic models
We used dislocation in a 2D elastic half-space to model the vertical tectonic deformations[34–36] (see Method, Fig. 5). We adopted the geometry of the SLAB 2.0 model[37], which is consistent with that derived from seismic multichannel profiles offshore southern Ryukyus[5,6,38]

(Fig. 1b). In Fig. 5, RSL drops could be promoted by coseismic uplifts due to a uniform slip of 8–25 m along faults 1 + 2, or by 1–9 m between 33 and 60 km on fault 2. By considering an arbitrary yet reasonable and conservative length of ~300 km for the ruptured patch (between Yonaguni and Miyako islands[39], Figs. 1 and 5), the formulation from Aki[40] and Hanks and Kanamori[41] with a rigidity of 32 GPa results in magnitude between Mw 7.7 and 8.3 if the rupture is restricted to the deepest segment (fault 2), and between Mw 8.3 and 9.1 if the earthquake ruptured the entire plate interface (faults 1 + 2). The hinge line (0 deformation), indicating the downdip limit of the rupture, would be located somewhere between Ishigaki island and the edge of the Okinawa Trough (Fig. 5). We also modeled the ~0 B.C. coseismic uplifts values documented by Kawana[15,16] and found that they could be reproduced by 25 m of slip on the entire plate interface between 0 and 50 km depth. A rupture of 300 km long would imply a magnitude Mw 9.0 for this earthquake.

In Fig. 4, some old corals are lower than younger generations. Our models suggest that this could be due to a sudden coseismic subsidence generated by a shallow megathrust rupture or a transient interseismic deformation, possibly SSE[18] (Supplementary Information S3). We cannot, however, exclude the possibility of local coseismic compaction of the soil.

Interseismic surface deformation varies over time, as also observed in Sumatra[29]. This can be explained by the occurrence of transient events and loading of more or less coupled asperities at various depths[42]. From modeling, we estimated the coupling rate of asperities of -15–100% (Supplementary Information S3). The lower bound is comparable to the 20% estimated by Ando et al.[12]. Slow interseismic uplift or subsidence can be explained by alternating loading of the shallowest and deepest segments, respectively (Supplementary Information S3).

## Discussion

From elastic models, we infer that the -0 B.C. coseismic uplift documented by Kawana[15,16] may have been promoted by a Mw 9.0 megathrust earthquake. This event coincides with the Sakishima Tsunami[43] and event R, which marks the end of the seismic supercycle 2 (Fig. 4b). Another older tsunami, that we will refer to as Yaeyama event, occurred around 2250 B.C.[7,14] and is coeval with events G to K at the end of the seismic supercycle 1. Given the age uncertainties of tsunamis, they could correspond to several events recorded by the microatolls; however, it is striking that they occurred close to the end of the supercycles that we documented. For the most recent period (supercycle 3), up to four tsunami events are documented, including the most recent Meiwa tsunami. The corals documenting this period are underwater and could not be sampled. Our observations are thus sparce, but we infer three to five events from our reconstruction. One of them may correspond to the megathrust earthquake that may have generated the Meiwa tsunami[44]. During supercycles 1 and 2 before the Yaeyama and Sakishima events respectively, sparce tsunami deposits were documented (Fig. 4b), and only discrete seismic uplifts occurred. Although we cannot distinguish between shallow and deep dislocation models, the absence of tsunami traces during these periods may indicate deep ruptures for the smaller earthquakes. Shallow seismicity and numerous small earthquakes along the deep portion of the interplate below the Moho occurred in Japan and Chile subduction zones before major megathrust earthquakes[45]. It is possible that deep asperities also exist along the Ryukyu megathrust. Repeated ruptures of deep asperities may contribute to the loading the shallower part of the megathrust which may eventually rupture on the whole interface, producing a giant earthquake.

The supercycle 2 is the only one to have been fully recorded by corals. It lasted two millennia, which is also the time span between the major events of Yaeyama and Sakishima. This recurrence interval is similar to that of the main Mw-8 class earthquakes that have uplifted the Kikai island over the last 6 ka[46]. It is also consistent with that of tsunamis and/or earthquakes estimated from submarine gravity-flow deposit records over up to 10 millennia[47,48] (e.g. 1–3 ka). From our reconstructions, we inferred a possible coseismic uplift of -23 ± 21 cm during the 1771 Meiwa (Supplementary Table S5). Elastic models show that such amplitude would imply a maximum magnitude between 7.7 and 8.6, comparable to that estimated for smaller earthquakes within supercycles. The Meiwa earthquake could be part of the ongoing supercycle 3 quiescence period rather than being of the same category as the largest earthquakes interpreted in the failure sequences at the end of seismic supercycles 1 and 2.

Supercycles may result from a combination of physical mechanisms operating over multiple seismic cycles. These include the presence of structural barriers on the megathrust that occasionally allow rupture to propagate over larger segments than usual, as well as long-term stress redistribution through postseismic processes such as afterslip, viscoelastic relaxation, or poroelastic rebound[33]. Nakamura[49] notably observed that deep aseismic events (at depths greater than 30 km) may trigger shallower ones (12–25 km), potentially due to changes in frictional conditions at depth. This suggests that frictional coupling within the subduction interface could influence the timing of large earthquakes. Similarly, numerical simulations[50] indicate that

spatial variations in fault friction may induce seismic supercycles. Over a period of 5000 years, a long-term average uplift rate of -0.4 mm/year is superimposed on the sawtooth curve of the seismic cycle. At first order, this long-term vertical motion is greater than the RSL variations due to the GIA and could represent an anelastic component of the vertical movement accumulated from one cycle to the next. This anelastic component, not balanced by coseismic subsidence, may accumulate during interseimic loading of the shallowest part of the megathrust, coseismic strain release on deeper segments, or a combination of both. Such irrecoverable deformation may have resulted in long term Quaternary topography, Ishigaki island being surrounded by several Quaternary marine terraces[51].

Extending this -0.4 mm/year trend line into the present, the current RSL would be expected to be up to 75 ± 10 cm below the HLS of the living microatolls. Certain seismic ruptures of the shallowest megathrust segment may have contributed to lowering the topography, as observed in Sumatra[52] (Supplementary Fig. S13). An alternative hypothesis is that interseismic loading of the megathrust beneath the islands is responsible for drowning the youngest generations of corals (Supplementary Fig. S14) and that a future earthquake is preparing. This earthquake could release the strain accumulated since the last major Sakishima event, either partly during a discrete event or completely, marking the end of the supercycle 3. Regardless of its magnitude, this earthquake may trigger a devastating tsunami.

## Methods

### Sampling site study

We measured and sampled numerous sets of well-preserved fossil and living microatolls over an extended area in the Nagura bay (Fig. 2). Fossil microatolls provide highly useful and valuable information to retrieve past RSL variations due to tectonics or climate[30,53–55]. They were first characterized by Stoddart and Scoffin[56], as they present the same morphology as atolls, typically a round shape at the surface with a dead plateau at the center. Microatolls grow at a first-order constant rate underwater and their development is controlled by RSL variations at a scale of centimeters. Their upward growth is limited by their highest level of survival, or "HLS". The HLS is the theoretical elevation above which the exposed part of the coral dies due to excessive subaerial exposure. Once the microatoll records its first HLS, the growth of its surviving lower part is conditioned by the variations of the HLS. We distinguish HLS and HLG[57–59] (for "highest level of growth"). "HLG" is used for years during which the coral grew freely upward and the HLS was not reached (green points in Supplementary Fig. S12); the distance to the theoretical coral HLS is unknown[60]. In the coral stratigraphy, a HLS year preserved from erosion corresponds to an unconformity over which younger growth bands curl over, while the growth band is complete during a HLG year[57]. The coral upward growth is often disrupted by temporary centimetric sea-level drops, commonly named "die downs"[24,59].

The microatoll morphology evolves accordingly and therefore records the variations of the RSL. When the RSL is stable, the microatoll develops laterally and its surface evolves into a flat plateau. When the RSL decreases, the upper part of the microatoll is emerged and die, and the part which lies below the new HLS continues to grow mostly laterally. The emergence can be sudden or progressive. In the case of sudden emergence, the morphology of the coral will show a clear lower step in its external part. In the case of progressive emergence, the coral height will decrease towards its most external part. In both cases of emergence, the coral has a hat-shape morphology. In the case of submergence, when the RSL increases, the coral grows upward and has a cupshape morphology. As the coral upward growth is limited by the coral growth rate, no distinction can be made between a sudden or a progressive submergence context. Alternation between periods of relative submergence and relative emergence along the coral life

implies the formation of concentric rings on its top surface. Such morphological features can often be observed and measured.

The exact relationship between HLS and RSL is not fully understood today. The growth rate is 5–25 mm/yr for *Porites* species[28]. Each year, the coral microatoll records a dark and a bright growth band that correspond to variations in aragonite density related to seasonal changes. Each of the growth bands will provide information on the HLG/HLS of the coral for its corresponding year. In our study, we use bright growth bands which correspond to rainy season in the middle of the year. The HLS is often associated with the lowest tide level, or Extreme Lowest Water[24,57,61] (ELW). The survival of a microatoll may also be associated with other phenomenon such as the sunlight exposure[24]. Thus, the HLS may lie tens of centimeters above the corresponding ELW[17,18,60]. This observation is very specific to each study site as the survival of the coral depends on the species and on the annual tidal characteristics of the sampling site.

The RSL changes deduced from the morphology of a microatoll can be of climatic, oceanographic and/or tectonic origin. During a period of stable absolute sea level, a seismic uplift would induce a RSL decrease. At the contrary, a seismic subsidence would correspond to a RSL increase. An event of tectonic origin is a few to few tens of centimeters of persistent vertical motion of the land in a restricted geographic area. It induces a RSL change and may be coeval with a historical earthquake[24,58]. Annual or multidecadal sea-level changes induced by climatic and/or oceanographic events, such as the El Niño Southern Oscillation[17] are recorded over a wider area. In the coral stratigraphy, they are characterized by an abrupt decrease followed by a rapid increase of the RSL. The modern RSL record of microatolls can be compared with that of the tide gauge, satellite and GNSS data to decipher between the tectonic and climatic and/or oceanographic signals. At a longer time scale (Holocene period, few millennia), the microatolls may record sea-level changes related to the glacial isostatic adjustment (GIA) following the ice sheet melting in northern hemisphere or Antarctica[54,55,62,63].

During field missions in July and December 2018, and November 2021, we mapped each set of microatolls by using a total station, a tool combining a theodolite for angle measurements with an integrated laser distance meter, allowing rapid acquisition of relative three-dimensional coordinates. We also used a Real-Time Kinematic GNSS with WGS 84 coordinates (RTK-GNSS, Fig. 2b, c). We measured in a systematic way the altitude of the external ring (younger part of the microatolls, i.e. most recent HLG), the center of all microatolls and the elevation of the substrate on which they developed, when possible. When visible, we also measured the altitude of other morphological features such as markers of possible older HLS records (older concentric rings) along diameters in several directions. We measured the altitude of the HLS of one living microatoll (Fig. 2c), which was necessary to compare the elevation of the old HLS recorded by the fossil microatolls and the modern HLS.

Altitudes and distances relative to the total station base are measured with millimetric precision. We used reference screws and reference microatolls to adjust total station measurements to GNSS results with a closure error of <1 cm, and with a resulting precision related to the GPS-RTK method of ±5 cm. The vertical variability on a given concentric ring is estimated at ±3 cm in average. This last uncertainty is considered as a statistical type A uncertainty, $\sigma_{mean}$, and the error on a single concentric ring elevation (above the 0 of the WGS 84 ellipsoid), $\sigma_{Elev}$, is the sum of the two errors added in quadrature, i.e. ±5 cm to ±7 cm.

$$\sigma_{Elev} = \sqrt{\sigma_{mean}^2 + 0.05^2} \quad (1)$$

Other errors related to erosion or compaction of the site are complex and difficult to quantify. As we consider that they mostly affect all corals similarly, they are unlikely to bias the RSL curve

reconstruction of the site. To include deeper fossils documented by Yamaguchi[22] (Yama_1 and Yama_2, Fig. 2), we plotted the microatolls documented in Yamaguchi[22] against our measurements. As their elevation was estimated and not measured with RTK-GNSS, we apply an uncertainty of ±50 cm. In Yamaguchi[22], Yama_1 and Yama_2 were dated with $^{14}$C method, with ages of 1063-1257 cal BP and 700-885 cal BP, respectively. This corresponds to ages of 790 ± 97 and 1157 ± 92 years A.D., respectively. High-resolution of 3.2 cm at 75 m of altitude drone imagery was also acquired in the site of Nagura with a Promark120 drone. A mosaic of images was made using the Photoscan software.

## Coral slices analysis

After the topographic mapping, we sampled, among the largest and best preserved, seven microatoll fossils. Before the sampling we verified that they were not tilted nor overturned. We sampled ~10-cm-thick sliced of coral skeleton along the radii or the diameters of the microatolls with a hydraulic chainsaw. Before sampling, we installed screws at ~10 centimeters interval along the slice and we measured their altitude in order to retrieve the growth position of the slice before sampling. A reference screw was also installed and measured near the microatoll to compare the measurements made during different days and for future investigations if needed. Thin cutting of the coral slabs was further performed at the Ogyu Manufacturing Inc., also known in Japan as Milano Seisakusho Co. Ltd (equipped with stable diamond wire) to obtain 1–2-cm-thick slices. The thin slices were then X-rayed at the Veterinary Medical Center of the University of Tokyo to image in thin section the internal architecture of the coral skeleton and to evidence the alternation of white and darker growth bands for counting. This is also important to distinguish and discard from our analysis new overgrowths that commonly develop on top of dead corals. We also used CT Scan imagery available at the National Institute of Advanced Industrial Science and Technology in Tsukuba. CT scan advantage is to provide a 3D tomography of the slice in which numerical virtual slices (tomographic cross-sections) as thin as 2 mm can be extracted by using the Horos software (Horos Project; https://www.horosproject.org/) without cutting the sample. The resolution of those virtual sections is lower than X-Ray however. The X-Ray or CT scan images are reoriented within an uncertainty of ±1 cm in its initial growth position by using the altitude of the screws. We then measured and characterized each growth band to reconstruct the RSL change curve.

Absolute dating is needed to estimate the chronology of the sea-level changes. We collected and dated with the U/Th method[23,64] several samples both along the coral slabs and on many other microatolls of different generations (Supplementary Table S2, Supplementary Information S1). Along the slabs, we collected, by pairs with a small core drill, several micro-core samples of the coral skeleton on one growth band. We used a drill to ensure that the hole created from the core drill extends through the whole slice and can be located later on X-ray imageries of the slice. On the other fossil corals not sampled with the chain saw, we collected with a hammer decimetric samples on the youngest external part of the microatolls. Considering an average growth rate of ~1 cm/yr for *Porites* corals[28], we estimate that their age approximates that of the most external growth band, i.e. the year of their death, by several years. Coral boulders samples were cleaned by ultrasonically bath using tap water for 20 min. After that they were cut within several cm thicknesses. The coral slices sampled were also cleaned by ultrasonically bath using tap water for 20 min and dried in the incubator at 40 °C for several days. All samples were divided into two subsamples, one for U/Th dating, the other one, grounded to powder, for X-Ray Diffraction (XRD) measurements. Those last samples were analyzed with a MiniFlex II (RIGAKU corporation) diffractometer. The scan-angle, speed, and step were from 25 to 32°, 1°/minute and 0.01°, respectively. The aragonite and calcite in each sample were detected using PDXL v.1.8.03 Software (RIGAKU corporation).

All samples were measured by XRD to estimate the percentage of aragonite and screen for calcite. Only samples with more than 95% of aragonite were dated at the High-Precision Mass Spectrometry and Environment Change Laboratory (HISPEC), National Taiwan University[23]. The results are presented in Supplementary Table S2.

Uncertainties on the chronology of the RSL changes may be large when established by using fossil corals[31,53,65]. It depends both on the uncertainties on U/Th ages and on the growth band counting. We assign an age and uncertainty to each growth band from a combination of U/Th dating results and growth band counting. To estimate uncertainty related to growth band counting, we used a numerical estimation from Weil-Accardo[66]. We followed the formula detailed by Meltzner and Woodroffe[28].

## Reconstruction of the RSL curve

We estimated the amplitude of the identified diedowns using the difference in elevation between the highest HLG point (eroded or not) preceding the diedown[60], and of the HLS point (eroded or not) marking the diedown itself. We calculated the rate of sea-level rise of decrease from our microatoll record by using different methods provided in Zachariasen[58] and Zachariasen et al.[57] (Supplementary Table S4).

The methods used to estimate RSL trends are all based on simple linear regression and depends on the number and type of data used: Zachariasen et al.[57] used the maximum height of all growth bands except those corresponding to the initial upward growth of the coral (Zach-1 method), Zachariasen[58] used only the elevation from the HLS bands (Zach-2 method), and we propose Zach-3 method, derived from Zach-2, that consider only non-eroded HLS bands. Following Meltzner et al.[60], we also use when available the highest eroded or preserved HLG point preceding a diedown (Meltzner method). We distinguish the Zach-3 and Meltzner methods that provide well-constrained results. In contrast, the Zach-1 and Zach-2 methods include HLG points and/or eroded HLS/HLG points, which can lead to an underestimation of the RSL and produce results that do not reflect the true RSL changes. We refer to these results as "apparent" RSL changes. However, when the RSL record is robust, all methods give similar results[67,68] (Supplementary Table S3). We use Zach-3 method to reconstruct the RSL history in Nagura and Zach-1 method for slice analysis, when Zach-3 method could not be applied. The HLS curve of a single microatoll has a relative uncertainty of ±1 cm, due to the uncertainty in the horizontal reorientation of X-Ray and CT scan images. The amplitude of the RSL drops that were only measured in one slice has therefore an uncertainty of ±1 cm. When a drop was observed in several slices (e.g. NAG6_21C and NAG6_18Q, Fig. 3), we observe an average range of 15 cm between measurements, leading to an estimated uncertainty of ±7 cm. This value is in the same range as the HLS variability observed in a given site from microatoll study in the Ryukyus[17,18], therefore we use an HLS variability of ±7 cm. As an indicator of past RSL, each plateau is assigned an uncertainty of ±7 cm. Each plateau elevation is estimated as the average elevation of 1–10 corals or concentric rings in coral surface. For the uncertainty in elevation of each coral $\sigma_{HLS}$, we added in quadrature the uncertainty of ±7 cm related to HLS variability.

$$\sigma_{HLS} = \sqrt{\sigma_{Elev}^2 + 0.07^2} \quad (2)$$

To estimate the uncertainty of each plateau $\sigma_P$, we calculate the uncertainty on the weighted mean using the following equation:

$$\sigma_P = \sqrt{\frac{1}{\sum_{i=1}^n \frac{1}{\sigma_{HLSi}^2}}} \quad (3)$$

To reconstruct the RSL history of Nagura site, we use a constant interseismic RSL change of 2.0 ± 0.6 mm/year (see text). Uncertainty of

the amplitude of the RSL drops we infer from a given plateau $P_i$ is therefore controlled by (1) the age uncertainties of the plateau $P_i$ and the following younger one $P_{i-1}$, (2) the vertical uncertainty of both plateaus $\sigma_{Pi}$ and $\sigma_{Pi-1}$ and by (3) the uncertainty of the interseismic RSL rate $\sigma_R$ we use. Considering those latter, we calculated for each documented drop its related uncertainty with the following equation:

$$\sigma_{Drop} = \sqrt{\sigma_{Pi}^2 + \sigma_{Pi-1}^2 + \sigma_{inter}^2} \quad (4)$$

Estimation on the uncertainty related to the interseismic rate $R + \sigma_R$ and the age of each plateau $Age_{Pi} + \sigma_{agePi}$ and $Age_{Pi-1} \pm \sigma_{AgePi-1}$ is given by:

$$\sigma_{inter} = R * \Delta_T * \sqrt{\left(\frac{\sigma_R}{R}\right)^2 + \left(\frac{\sqrt{\left(\sigma_{agePi}^2 + \sigma_{agePi-1}^2\right)}}{\Delta_T}\right)^2} \quad (5)$$

With $\Delta_T$ being the time lapse between the age of the two plateaus.

The diversity of our dataset affects the robustness of our interpretation. To account for this variability, we assign a confidence index ranging from 1 to 6 to each interpreted seismic event (Supplementary Table S5). Earthquakes identified within coral slices receive a confidence index of 6 when affected growth bands are preserved from erosion, and of 5 otherwise. Those interpreted based on the death of a coral observed at the same elevation as several other colonies are assigned a confidence index of 4. Events inferred from the death of a single coral colony are given a score of 3. Earthquakes corresponding to the observations reported by Yamaguchi[22] receive an index of 2, and the event attributed to the hypothesized Meiwa earthquake has a score of 1. For all events whose estimated vertical motions are close or smaller than their associated uncertainties, the initial confidence index is reduced by one point. The same stands for events identified in one slab that are not observed in coeval slices. We deduced the subsidence event A' from the difference in elevation of several corals (NAG6_21C and NAG6_18Q, ang NAG_M and NAG6_18H). It was assigned a confidence index of 4. However, we do not interpret it further due to the uncertainty of our 35 cm amplitude estimation.

## Modeling

Our elastic model consists of a simple dislocation in a 2D elastic half-space[34] with a rigidity of 32 Gpa. The geometry of the plate interface is determined from SLAB 2.0 model from Hayes et al.[37] (Fig. 5).

According to SLAB 2.0, for simplicity, we modeled coseismic deformation on the plate interface with two faults: Fault 1 with a dip of 12.5° between 0 to 33 km and Fault 2, with a dip of 31° between 33 and 50 km or between 33 and 60 km. We also attempted to reproduce the coseismic deformation promoted by slip on a normal fault at the southern margin of the Okinawa trough. In Supplementary Fig. S13, we used a 20 km-wide dislocation dipping northward by 45° with a slip of 3 m (implying a magnitude 7.7 for a length of 100 km).

During the interseismic period, the slip deficit accumulation on the megathrust can be modeled by a back-slip on a planar fault[35]. Such a model has been developed on curved geometry, and when a kink in the slab dip is observed above the downdip limit of the locked zone, surface deformation can be better reproduced by dislocation on a planar fault tangential to this downdip limit[36,69]. In Supplementary Fig. S14, we therefore used back-slip dislocation on Fault 3 tangential at a depth of 33 km to the locked zone corresponding to Fault 1, and on Fault 4, which is tangential at a depth of 60 km to Fault 2. We used different dislocations on Faults 1–4, corresponding to seismic coupling between 0 and 100%, 100% indicating a fully-locked zone. When the plate interface is fully coupled, the slip deficit corresponds to 12 cm/yr, which is the convergence rate between both plates[70].

## Data availability

The authors declare that most of the data supporting the findings of this study are available within the paper and its Supplementary Information. Samples are accessible on site at KIKAI Institute for Coral Reef Sciences. The data used in this study (Drone images, slices images and elevation data for each plateau) are available in the dataset[71] on the following link: https://doi.org/10.18715/IPGP.2025.mh417kg4.

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

## Acknowledgements

The corals were sliced thank to the Milano Industry Co. Ltd (Ogyu Manufacturing Inc.). The slices were X-rayed with the kind help of Prof. M. Mochizuki from the Department of Veterinary Medical Sciences, the University of Tokyo, and CT-scanned with the contribution of K. Seike's team from National Institute of Advanced Industrial Science and Technology (AIST). We are very grateful to J. Weil-Accardo for her helpful advices on the coral analysis process, and to A. Yamazaki for her help to identify the corals specy. We are also grateful to A. Henry and Jo. C. Grall

for their very constructive help with GIA matters. We thank T. Yamaguchi for fruitful discussions about Nagura microatolls. Maps were created using Generic Mapping Tools[72], and Adobe Illustrator (https://www.adobe.com/). Topography and bathymetry are from SRTM15+[73]. We used Coulomb 3.3[74] to model the surface deformation promoted by a dislocation in an Elastic half space. Satellite imagery used in this study was obtained from Google Earth Pro. Figure 2 is based on imagery from 2020 (© 2020 Maxar Technologies, CNES / Airbus, Google Earth), and Supplementary Fig. S9 is based on imagery from 2025 (© 2025 Maxar Technologies, CNES / Airbus, Google Earth). All imagery was used in accordance with the Google Earth Terms of Service (https://www.google.com/permissions/geoguidelines/). The coral samples were collected with the permission of Ministry of Environment of Japan and Okinawa Prefecture. Fieldwork and coral analyses were funded by the JSPS KAKENHI Grant JP16H01838 and JP21H01167. U/Th dates using solution MC-ICPMS protocol for stalagmites were determined at the High-Precision Mass Spectrometry and Environment Change Laboratory (HISPEC), Department of Geosciences, National Taiwan University, supported by grants from Taiwan ROC MOST (111-2116-M-002-022-MY3 to C.-C.S.) and National Taiwan University (112L894202 to C.-C.S.).This work was supported by the Institut National des Sciences de l'Univers, Centre National de la Recherche Scientifique (CNRS-INSU Alea program 2019), the Institut de Physique du Globe de Paris (BQR IPGP 2018), the Ministère de l'enseignement supérieur, de la Recherche et de l'Innovation and the Ecole Doctorale UPC-IPGP.

## Author contributions

Conceptualization, Funding acquisition: N.F., K.S. Field observations and sampling were performed by S.D., N.F., K.S., M.N., K.Sowa, M.Y., T.S., A.W., A.S., J.-M.S., G.O. S.D. prepared the slices with assistance of M.Y., T.S. and K.S. S.D., K.Sowa, T.-L.Y. and C.-C.S. processed the XRD measurements and T.-L.Y. and C.-C.S. performed the U/Th dating. N.F. provided the elastic models. S.D. wrote the paper with assistance of N.F. and K.S. All authors reviewed the manuscript.

## Competing interests

The authors declare no competing interests.
