## [Transparent Peer Review file · Nature Communications]

Evidences of megathrust earthquakes and seismic supercycles in subtropical Japan from millennia-old coral microatolls

Corresponding Author: Dr Sophie Debaecker

Redactions – published data

Version 0:

Reviewer comments:

Reviewer #1

(Remarks to the Author)

Debaecker et al. present relative sea-level (RSL) reconstructions from coral microatolls located along the southeastern coast of Ishigaki Island, located north of the Ryukyu subduction zone. Combining the coral slab interpretations with elastic dislocation modelling, the authors suggest several coseismic uplift events superimposed on millennial-scale long-term relative sea-level (RSL) fall that may be attributed to permanent deformation (irrecoverable uplift) over seismic cycles. Considerable effort was taken to provide detailed documentation to support the findings of this study, both in the main text and the supplementary material.

We believe that the coral record presented has important implications for advancing the understanding of the seismogenic potential of the southern Ryukyu subduction zone.

We recommend publication of the manuscript after addressing the following concerns.

We find that there are important details that need to be addressed, particularly with regards to the inferred coseismic uplifts. We believe that greater focus should be placed on the periods of RSL rise recorded internally within the coral slabs, which may indicate interseismic or postseismic subsidence. The uplift estimates are problematic due to erosion of the coral surface. Glacial isostatic adjustment (GIA) drives far-field RSL fall during the mid to late Holocene, which may explain a non-negligible part of the RSL fall that is currently inferred as long-term net land uplift. The inference of RSL changes between different coral generations (vs internally within a slab) is more challenging as it requires 1) clearer demonstration that the corals were in-situ and were not moved by past tsunamis; and 2) consideration of the natural variability in coral growth that is currently underestimated.

We provide detailed comments below.

Major comments

1. There seems to be a misinterpretation of diedowns, highest level of survival and highest level of growth, and missed recognition of erosion.

Lines 406-407: "... a HLS year corresponds to a truncated growth band, while the growth band is complete during a HLG year". A truncated growth band indicates erosion, not the HLS. The HLS is the level the coral dies down to during a diedown event. In cross section, the HLS can be recognised as an unconformity, where the annual bands are seen to curl over at the diedown (resulting from outward growth at the HLS after the diedown).

In Text S2 of the supplemental information, years that were described as the 'first HLS' being reached were marked by the first (oldest) annual bands that were truncated (supplemental lines 143-144; 146). The first HLS should be where the first diedown is recorded by the coral, not where the first annual bands get truncated.

In lines 406-409, HLS was described first, followed by a separate explanation of diedowns as 'centimetric sea-level drops', suggesting the authors do not recognise that HLS are only formed during diedowns.

As a result of the misunderstanding of HLS, several consecutive years were misinterpreted as HLS years and therefore stable RSL (Figure S2b) (supplemental material lines 119; 150; 188; 199), when in fact we cannot know because of erosion of the top of the coral.

Additionally, in several instances erosion was misinterpreted as relative emergence/ periods of RSL fall (supplemental material lines 123; 200; 213; 237). In these cases, often the slab profiles indicate the truncation of near-vertical growth bands, which are a clear sign of erosion. Near the base of a coral, the growth direction is near horizontal, so the annual bands (which are perpendicular to the growth direction) are near vertical. If there were no erosion, the surface of each concentric ring of the microatoll should show annual bands that are near-horizontal, as growth direction is near vertical at the ring crest.

Figure S2b:

- many 'eroded HLS' points are actually 'eroded HLG'
- Some of the 'HLG' points are eroded, but there is no distinction between eroded HLG and preserved HLG in this figure. HLG is preserved when the ring crest of a particular annual band is preserved; if the annual band is truncated, then the HLG in that year is eroded.

2. The authors do not account for natural variability in the coral HLG/ HLS.

For the RSL reconstruction, only two living microatolls were surveyed (line 343). This likely does not capture the natural variability in microatoll HLG. The authors report levelling uncertainties (line 348) and a vertical uncertainty of +/- 3 cm on a single concentric ring (lines 348-349), but it is unclear what the latter refers to. Published studies indicate that HLG of *Porites* sp. microatolls may vary by up to about +/- 20 cm across a study site. When reconstructing centimetric-scale RSL changes between different corals this uncertainty in the natural variability of coral HLG becomes important.

Related to the above, the authors also do not mention the genus of corals being studied. Different genus can survive at different elevations with respect to the tides. Can this explain the vertical offset between the corals NAG_B and NAG3_18H? Based on the caption of Figure 4.2 in the Yamaguchi et al. 2016 chapter, it seems that there were fossil *Porites* sp. and *Favia* microatolls at this site.

3. Possibility of reworking

This is an area with historical tsunamis (Figure 4b). Is there a possibility that the corals could have been transported during a tsunami? NAG6_21C looks like a fragment of a coral that was broken off and was found within the main river channel. Is it possible it was transported during the Yaeyama tsunami event?

Were there dated fossil corals at similar elevations to those slabbed, which would support the in-situ nature of those slabbed corals? Currently, it is difficult to discern based on the plateaus shown in Figure 3. The plateaus are presumably combined from different dated (but not slabbed) microatolls? It may be more useful to show individual sea-level index points for each dated microatoll.

As coseismic uplift events were inferred from the relative elevations of dated coral blocks, we believe it is also important to consider if these dated coral blocks were in-situ.

4. Coseismic uplift events

The authors reconstructed coseismic uplift events by assuming an "arbitrary interseismic RSL rate of 1.4 +/- 0.2 mm/yr" (line 139), supplemented by coral analyses (Table S2b).

The application of this arbitrary interseismic RSL increase (i.e., interseismic subsidence) forces coseismic events even when the coral elevations and ages themselves may not require so. Events L, M and N were recorded by dated plateaus at rather similar elevations. However, these were inferred as coseismic uplifts because of the assumed interseismic RSL rate. It is possible that this period was marked by gradual RSL fall, rather than coseismic uplift followed by interseismic subsidence.

Of the 22 uplift events reported, only three were supported by RSL fall internally within the slab profiles (events E, G, and O). A third was not inferred but is evident in the NAG6_18H slab (the annual band for year 2089 that was sampled for the U/Th date A2); event J was inferred a few years later than year 2089 but the annual bands do not suggest a diedown in the year of event J. For events E, G and O, the slab profiles suggest evidence of a RSL fall, as indicated by a step-down in elevation from the pre-diedown HLG (be it eroded or preserved) to the HLS at the diedown. However, the HLS in events G and at the 2089 band in NAG6_18H were not preserved, so it is also possible that the step down in elevations are from erosion rather than a true RSL fall. The pre-diedown HLG in all of events E, G and O are not preserved, so the inferred uplifts can only be minimum uplift estimates. The diedowns for events O and G also appear from the slab annotations to be a few years earlier than is currently inferred.

Event D is obscure because of the "non-uniform growth" that was described in lines 184-185 of the supplementary material.

If the diedown was indeed at the top part of the discontinuous growth, then the diedown is small and may be due to other oceanographic/climatic drivers rather than coseismic uplift. However, the annual bands in the lower portion of this discontinuous growth appear to be curling in a different direction than the upper part. Is it possible that the upper part is out-of-sequence overgrowth?

For event P, the pre-diedown HLG is also eroded, therefore the uplift for this event should also only be treated as a minimum uplift estimate. There appears to have been continuous upward growth between events O and P. It is possible that postseismic or interseismic subsidence after the uplift in event O outpaced the coral growth rate, and the coral only caught up to sea level during event P. Event P may also be due to other oceanographic/climatic drivers rather than coseismic uplift, as the magnitude of the diedown appears small.

For event R, the HLG/HLS is not preserved, so it is not clear there was a diedown (or RSL fall), and it should not be used to quantify or characterise coseismic uplift. Based on the slab, there seems to be two to three earlier diedowns before event R, and one after event R – all of which seem to be separated by a similar number of years. This suggests that if event R were a diedown, it is likely to be caused by periodic oceanographic drivers, rather than coseismic uplift.

Six uplift events were inferred from the thickness of the outer margin of the coral, which was assumed to have resulted from an uplift event that killed the coral entirely (events F, H, K, Q, R for NAG_B, S). It is common that coral microatolls produce RSL records that are temporally discontinuous, and which cluster in certain time periods. Therefore, we are not certain that the death of a microatoll necessarily implies that it was killed by an uplift event.

Several uplift events (Table S2b) were inferred from the relative elevations of dated coral blocks (i.e. sudden 'drops' in the elevations of younger corals): events A, B, I, J, K, L, R, T, U. We suggest further investigation into the possibility that the lower corals were eroded.

5. Attribution to RSL drivers

Overall, the elevations and ages of the corals in figure 4 appears to indicate a gradual RSL fall between ~3000 B.C. and present, apart from the two low coral blocks marking events T and U. Can the magnitude of the long-term RSL fall be explained by glacial isostatic adjustment (GIA)? Given that the study site is in the far field, a mid- to late Holocene RSL fall from a highstand can be expected. We suggest showing a RSL predictions from GIA models to illustrate the magnitude of RSL fall that can be expected from GIA.

Within this record, the slab profiles of NAG6_18Q and NAG3_18H indicate clear and abrupt RSL fall followed by gradual RSL rise; and that of NAG_B indicates gradual RSL rise. We believe that these periods of postseismic/interseismic subsidence (RSL rise) are more robust than the coseismic uplifts, whose magnitudes are not well-constrained due to erosion of the pre-diedown HLG.

Minor comments

Line 28: Remove hyphen from "relative sea-levels". As a noun, sea level should not be hyphenated.

Lines 30-31: "Elastic modelling shows that the observed motions are coseismic uplift associated with megathrust earthquakes." Rephrase this to suggest that there are other possible interpretations. In this study, a constant interseismic rate was assumed between majority of uplift events, but this is only one possible mechanism.

Line 46: "A seismic coupling of 20% was inferred in this area." Clarify what "this area" refers to.

Line 52: "This study provides the first example of multiple earthquake cycles over five millennia in a subduction zone." Is this referring to just the Ryukyu subduction zone? Or anywhere in the world?

Line 71: "... its upper part dies and it keeps growing laterally" Rephrase this. After the coral dies during a diedown, the coral polyps grows out radially (both upwards and outwards), not just laterally.

Line 80: "Photos of the sites, slabs, or samples of each microatolls." Correct "each microatolls" to "each microatoll".

Line 93: "... measured with an accuracy of +/- 6 cm" Is this referring to the precision of the instrument, rather than accuracy?

Line 125-126: "..., it grew 30 cm below and was too deep to record RSL drops of the same magnitude." Even if so, NAG_B should have died down to a similar elevation (within the natural variability of HLS) as NAG3_21E, just that the magnitude of the coeval diedown in NAG_B is larger. In figure 3, it appears that NAG_B died down to an elevation (-0.3m) that was 40 cm higher than the elevation that NAG3_21E died down to (-0.7m) at ~256 B.C.

Line 137: "... the external parts other microatolls". Add "of" between "external" and "parts".

Line 139: "We used an arbitrary interseismic RSL rate of 1.4 +/- 0.2 mm/yr (as determined from the best-preserved signal from NAG_B, Table S2a)..." In figure 3b, the interseismic rate labelled for NAG_B is 1.4 +/- 0.4 mm/yr, not 1.4 +/- 0.2 mm/yr as is described in lines 139 and 421; please make consistent.

Table S2 a does not indicate the rate 1.4 +/- 0.2 mm/yr, but it is described in the footnote of Table S2 that “Underlined values from NAG_B coral are used to reconstruct the RSL at a larger time scale in Fig. 4”. Explain how the 1.4 +/- 0.2 mm/yr was derived from the underlined values (1.6 +/- 0.2 mm/yr and 1.2 +/- 0.2 mm/yr), and how those specific underlined values were chosen?

Line 145: “ages of the undated plateaus are bounded by those of the plateau just above or below” Is there a stratigraphic relationship between the corals (i.e., does “above” and “below” refer literally to corals stacked on top of one another?) The extrapolation of ages of “undated plateaus” would make sense only if the stratigraphy supports it: a coral that grew physically on top of another coral must be younger than the coral beneath it. In Figure S1j, it appears the corals were all in separate locations. If this is the case, the relative ages of undated corals to the dated corals is not clear.

Line 207: Typo mistake here “(fault 2) 2”

Line 228: As per line 145, the sentence here “Some old corals are below younger generations” is vague. Does “below” mean that some corals are physically stacked on top of other older corals below?

Line 231: Given that the site is located at the mouth of a river, and some corals are located amongst mangroves, can there also be sediment compaction?

Lines 254-255: “Although we cannot decipher between shallow and deep dislocation models, the absence of tsunami traces could be in favour of deep ruptures for the smaller earthquakes.” The “absence of tsunami traces” seems contradictory given that the paragraph talks about documented tsunamis. Please clarify.

Lines 269-270: “... and not of the same category as the largest earthquakes interpreted in the area.” The phrase is vague, it is not clear what “the area” and the “largest earthquakes” refers to.

Lines 270-271, 277-278: Add line spacing between paragraphs.

Lines 281-282: If interseismic subsidence is currently occurring, is this evident in the geomorphology of the site? Are the living microatolls recording continuous relative sea-level rise? Are there other independent drowned landscapes?

Line 297: “Fossil microatolls permit to have extremely useful...” A subject is missing between “permit” and “to”?

Line 299: “Stoddard” is spelt wrong

Line 300: “Microatolls grow at a constant rate underwater” Corals can grow at variable rates throughout their lifetimes, causing the annual band thicknesses to vary across the microatoll.

Line 301: “their development is controlled by sea-level variations...” Please change “sea-level” to RSL, to clarify that corals microatolls also record land-level changes

Line 303: “excessive subaerial and sunlight exposure” It is not clear if excessive sunlight is needed for a coral microatoll to experience a diedown

Lines 303-304: “Once the HLS is reached, the growth is restricted to the horizontal direction” This statement implies that the coral simply grows upwards until a limit (the HLS), then proceeds to grow horizontally. In reality, the HLS is the elevation a coral dies down to (from its pre-diedown highest level of growth), rather than the elevation it grows up to. Additionally, following a diedown, the coral polyps continue to grow radially outwards (both horizontally and vertically).

Line 318: “the coral microatoll record a dark ...” Add an ‘s’ to ‘record’

Line 320: “will provide information on the HLS of the coral ...” change “HLS” to “HLG/HLS”. Depending on whether there was a diedown in a given year, the annual band could be representing a HLS (during a diedown), or HLG (between diedowns).

Lines 332: “The most recent RSL record” The term “most recent” is vague. Modern?

Lines 339-340: “the altitude of the external ring (younger part of the microatolls, i.e., most recent HLS)...” The outermost ring crest is not the most recent HLS, it is the most recent HLG. For fossil corals, the outermost ring crest is the most recent pre-diedown HLG (HLG just before the most recent diedown).

Line 344: “... one being ball-shaped” What does a ball-shaped microatoll mean? Does it have concentric rings? If not, is it just a hemispherical coral head, and not a microatoll?

Line 348-349: “The average vertical uncertainty on a single concentric ring measurement is of +/-3 cm”. This statement is vague. Is this referring to the precision of the levelling equipment? Or the natural variability in HLG across a given concentric ring, derived by multiple elevation measurements of the same ring?

Lines 351-352: “Other errors related to erosion or compaction of the site are complex but small ...” The coral slabs suggest significant erosion, as evident from the truncation of near-vertical growth bands.

Lines 353-354: Were the Yamaguchi et al dates calculated using the same half-lives as the dates in this study?

Line 363: Some coral slabs were sampled along the radius of the coral, not diameter.

Lines 371-372: "... to distinguish and discard from our analysis new colonies that commonly develop on top of dead corals." Colonies B and C of NAG_B look like they belong to an independent coral colony that's distinct from colony A.

Line 389: Is the uncertainty in the sampling of coral chunks (described here to approximate the most external growth band "by several years") accounted for in the uncertainty of the coral ages during subsequent interpretation?

Line 408: We recommend shifting this part describing the "die downs" together with the part that introduces the HLS, as they are related concepts.

Line 417: Clarify that the +/- 1 cm relative uncertainty in the HLS curve of a single microatoll relates to the uncertainty in the reorientation of the X-Ray and CT scan images (described in lines 377-378). What about band-counting uncertainty, associated with uncertain interpretation in the coral growth bands when trying to reconcile the annual bands with the U/Th dates?

Lines 418-420: "... we observe an average range of 15 cm between measurements, leading to an estimated uncertainty of +/- 7 cm." Does this mean that coeval diedowns were observed at different elevations in different corals, therefore deriving an uncertainty of +/- 7cm? If so, is this implying the natural variability in the HLS is +/- 7 cm?

Line 426: "... permit to make..." A subject is missing between "permit" and "make"

Lines 427-428: It is not clear how the 21 cm is derived.

Supplementary line 95: How was mean sea level derived?

Supplementary Table S2b: Event P – is there a typo for the age of the event (442 BC not 44 BC)?

Supplementary Table S2b: what is the difference between the "Age" column and the "All-data age average" column? How were both derived?

U/Th ages:

- Were the samples pre-screened for secondary calcite?
- NAGM_A1 and NAG6_18N have initial d234U values that deviate by more than 5 ‰ from modern seawater (145 ‰), suggesting possible open system behaviour. Were these used in the analyses?
- Samples NAG321E_A1, NAG618H_A2, NAG3_18C, NAG3_18D, NAG3_18B, NAG5_21A have anomalously low 230Th/232Th ratio, suggesting presence of non-negligible amounts of initial (non-radiogenic) detrital Th. How sensitive are the ages to the atomic 230Th/232Th ratios used for the age corrections (footnote d)?

Figure comments

Figure 1

- What do "low frequency" and "very low frequency" earthquakes mean?
- Difficult to identify/ distinguish the "Graben" and "Okinawa Trough" colours

Figure 2

- Can the coral locations in panel b be drawn shown also in panel a?
- It was very difficult to notice Yama_1 and Yama_2 in panel a as they are blue points on a blue background
- Is there a clearer image of the reef in panel b?
- Panel c: what is the datum for the elevation measurements? Specify in the caption or y axis title.
- Panel c: Do the "Fossil coral" symbols refer to the "undated corals"? If so, how were they placed along the x axis?
- Panel c: we recommend colouring the error bars according to the symbol colour, so it is easier to read which error bars belong to which samples
- Panel c: it is difficult to distinguish the red and orange symbols
- Panel c: why is the horizontal error bar for one of the living corals dashed?
- Panel c: how was the tidal range measured? It seems odd that the living coral microatolls are out of the tidal range. It may also be useful to show the lowest astronomical tide as an estimate for extreme low water level at this site – which is what controls the microatoll diedowns.

Figure 3:

- The caption describes this as a relative sea level plot, but the y axes are labelled as "Altitude (m)". Please clarify if this is indeed showing the absolute elevation of the corals, or relative sea level.
- For NAG_B, which radius is shown in panel b, given that the full diameter was slabbed?
- Uplift events estimated from the eventual death of coral microatolls should be minimum uplift estimates. We recommend annotating an arrow and indicating "> XXX cm" to illustrate that the uplift (or RSL fall) could be greater than the vertical red line drawn.

Figure 4: As with figure 3, please clarify if the y axis is showing the absolute elevations of the corals or relative sea level.

Figure 5: The ellipses for the deep and main megathrust in panel a do not match the cross section in panel b. In the bottom panel of b, fault 2 (deep megathrust) ends at the northern end of Ishigaki (middle panel of b), but in panel a the blue ellipse does not extend to the northern end of Ishigaki island. The bottom panel of b also shows the blue ellipse extending further beyond the main megathrust, but the northern limit of both ellipses in panel a are aligned.

Figure S1b – S1h: There seems to be a mislabelling of the screw positions. Lines 363-365 describe that elevations were surveyed before the slabs were extracted, so it seems unlikely that the screws would be along the yellow line, and should instead be on the coral surface.

Figure S2a: Please indicate scales for each of the coral slabs.

Reviewer #2

(Remarks to the Author)

Reviewer #3

(Remarks to the Author)

This is an exciting paper that the authors present a comprehensive study of the history of megathrust earthquakes over the past five thousand years in the southern Ryukyu Trench using coral microatolls. The manuscript is well-organized and clearly written. I only have few minor comments and support publication.

Lines 193-195: The inferred supercycles in the southern Ryukyu Trench are based on data from a single paleoseismic site. Are there any published studies using sediment cores from the seafloor, marine terraces, or tsunami boulders dated in Okinawa island that support the existence of these supercycles found in southern Ryukyu Trench?"?

Lines 232-238: The Okinawa Trough is rifting southward at a rate of approximately 5 cm/year in this region (e.g. Nishimura et al., PEPI, 2004). As a result, the convergence rate between the Philippine Sea plate and the southern Ryukyu forearc exceeds 8 cm/year (e.g. Chen et. al. <https://doi.org/10.1029/2022GL098218>). Authors need to refine interseismic modeling accordingly.

Reviewer #4

(Remarks to the Author)

Please see my comments in the PDF

Version 1:

Reviewer comments:

Reviewer #1

(Remarks to the Author)

See attached

Reviewer #2

(Remarks to the Author)

Reviewer #3

(Remarks to the Author)

The authors have thoroughly addressed all comments raised by me and the other reviewers. I have no further comments.

Reviewer #5

(Remarks to the Author)

The manuscript "Evidence for megathrust earthquakes (...)" provides clear evidence for paleoseismic activity along the

Ryukyu trench that can reasonably be attributed to megathrust earthquakes. The interpretation of the coral record seems robust. However, the paper is poorly written and the figures are of poor quality. I recommend minor revisions to improve the quality of the manuscript.

Line 26: replace "infer" with "imply". Write "Furthermore, historical and geological studies show evidence of great tsunamis." But perhaps indicate when and where, as the sentence, as is, is vague.

Line 27: "Here, we use". Use present tense and active voice.

Line 28: "relative sea level". Singular. "We use fossil microatolls in Ishigaki Island to reconstruct relative sea level in the Holocene. The coral record reveals several emergence episodes clustering 5-to-4 and 3-to-2 thousand years ago (ka) compatible with rapid uplift from megathrust earthquakes."

Line 33: remove "would". "The paleoseismic behavior implies a strong seismic hazard for the upcoming centuries. The devastating 1771 Meiwa earthquake and associated tsunami may mark the onset of a seismic supercycle."

Line 36: remove "," in the middle of the sentence. Remove "strong".

Line 43: "evidence". Singular. Remove "," before "and".

Throughout the text: There are problems with virtually every sentence of the manuscript. I will stop correcting, but this is a major issue. They are many useful tools to correct and improve the writing. See DeepSeek, ChatGPT, Mistral, and the like. The language and writing must be seriously improved.

Line 49: "evidence" -> "record". "To this goal" -> "Toward this goal"

Line 97: "motion". Singular.

Line 98: "motions"->"displacement".

Line 215: "2 ky apart"

Line 216: "over the last 6 kr".

Line 238: Permanent or irrecoverable deformation is often attributed to splay faulting and folding in the accretionary prism. Ishagaki is a forearc high built by such deformation. See a recent review:

Qiu Q. and S. Barbot, "Tsunami excitation in the outer wedge of global subduction zones". Earth-Science Review, 2022.

Figure 1: add contours of slab2.0 depth. Is it "Ruykyu" or "Ryukyus"? Probably singular. Use similar color themes for the main map and the two insets. Fix the width of the black rectangles on the side. Some have a black frame, others don't. Date the aseismic events.

Figure 2: can probably be split into two figures.

Figure 2c: "Distance along plot (km)" What is "plot". Is it a plot of land? Or plot as in "figure"? Perhaps find a more suitable axis label.

All the figures are pixelated. Hopefully, vector-graphic figures will be provided eventually.

Version 2:

Reviewer comments:

Reviewer #1

(Remarks to the Author)

No further comments

Reviewer #2

(Remarks to the Author)

Summary

This manuscript presents a new data set of relative sea-level over the last 5000 years on Ishigaki Island (tropical Japan). The authors acquire the data from coral microatolls and interpret their noisy record in form of three supercycles of 1'000 years with more earthquakes occurring in the last parts of those supercycles. They use the 1771 Meiwa earthquake to suggest the end of the third supercycle is approaching. I find the data somewhat over-interpreted, while the elastic dislocation models are too simple and have clear limitations. Together this makes some messages a bit too suggestive. However, the potential of such a long data set would be valuable to assess the potential and recurrence behaviour in this region with little data, where recent geodetic data suggest only weak coupling, suggesting limited hazard. However, I also have difficulty following and accepting all interpretation done in order to be able to use the data. I would like to see this data set published after correcting and answering various points, because it will make an important contribution for our understanding of how vertical motions over 1000's of years evolve. However, I strongly doubt whether your data interpretation is robust. Since I do not think the data allow a much more robust interpretation, I doubt that your warning of a possible imminent M9.0 earthquake is robust enough for publication in Nature Communications.

Next time please use line and page numbers. My review is now delayed in part due to the increased difficulty of providing my comments. Since I do not want to spend too much time to indicate where my comments relate to. I now ask the authors to search for the word combinations that I copy or search longer if needed.

Major points

Robustness and validity of the data

First of all I would like to note that I am no expert in paleosealevel changes and coral measurements. However, as an informed earthquake scientists with some expertise, I did my best to decipher the data presented. I found it hard to understand the presented data and follow several aspects of assumptions seemed very severe. Your descriptions makes some jumps in text and only to a limited degree refer to Figures, such that I needed to spend quite some time to follow and interpret what you are trying to show (e.g., sentences above Fig 3; are 3a and 3b a part of a longer record you that you only show for the first time in Fig 4? Then it makes more sense to first show the complete record (4) and then zoom into a few parts)).

Above figure 3 you write about *nine* sudden Relative Sea Level (RSL) drops with values ranging from 6 to 50 cm. If I would analyse Fig 3a without the interpretation lines you draw over the data I would probably not identify at least 3 events you do. For example:

- 2482: Where is the minimal data point of your event? I only see the red and black line you draw
 - 2357: Again no minimum point are any data at the time you draw
 - 2108: No minimum to connect to
- Some other dots e.g., the blue ones seem unrelated to your interpretation. How do they fit?

Additionally, I could identify at least half a dozen more drops that you do not identify. For example:

- Between 2208 and 2240
- Between I and J

- Also where from come the black lines beneath H and before G?

- Also in Fig. 4 I wonder what data support the drops and black lines that you draw in Fig 4a.
- And how do you determine the values of the minima in your sawtooth pattern in Fig. 4a?

In general for many lines in that key graph it is not clear where they come from. Please explain what the lines are and where they come from.

Even when you draw the interpretations for me I have a really hard time to make any sense out of those interpretations. Even after starring at it for more than 15 minutes I can not do it. It also makes me wonder whether you should draw so many lines on top of it, as the data themselves are not clear anymore.

In all, I do not find the presence of those nine events convincing at all.

The same holds for figure 3b. I could interpret many events after $t=518$ and after $t=256$ that look the same as the lines that you do show in red. I also see 7 interpreted vertical lines that I think you want to show as vertical drops, but the text here refers to *four* sudden drops. All these points make me sincerely doubt your interpretation and the robustness of your story.

That questionability about the robustness of your data and interpretation is strengthened when reading that the uncertainty is $\pm 6\text{cm}$ (caption Fig 2). Several drops that are in the data are around that 6 cm, so I do not think you can interpret that.

Overinterpretation of a super cycle

This data interpretation is then used to suggest strong seismic hazard for the next centuries (end of the abstract / conclusion section). Based on the data, which could also represent other earthquakes, this is suggestive.

- In Fig.4a you draw a large drop / event around 1300 BC.
 - (a) Based on what data is that drop / event justified? I do not see that in the figure.
 - (b) That very large event in the middle of your supercycle #2 does not really fit the supercycle thinking and process. Since supercycle #2 is your only complete one in your story line (Supercycle #1 you have the end and of #3 the potential approach of the end of another supercycle), I do not see one clear and complete supercycle. Hence I wonder how reliable your claim for recurrent supercycles is, and how likely it is

Overall, following my comments above I do not think the "RSL signal is strikingly similar to what was observed in Sumatra by Sieh et al.". The true supercycle signal presented here seems less evident.

Potential M9.0 not justified

You suggest that a M9.0 earthquake can explain the coseismic uplift data by Kawana, but I do not agree with that interpretation:

(1) The calculation does not add up. Following your numbers I calculate a seismic moment M_0 of

$$M_0 = \text{rigidity} * \text{slip area} * \text{average slip} = 32\text{e}9 * 300\text{e}3 * 50\text{e}3 * 25 = 1.2\text{e}22 \text{ N m}$$

Using empirical scaling relations that leads to a moment magnitude of

$$M_w = 2/3 \text{ LOG}_{10} (M_0) - 6.05 = \mathbf{8.67}$$

That is significantly smaller than a M9.0, as more than 10x less energy is released than what you suggest. A factor 10 is a large discrepancy.

To reach a M9.0 80 meters of slip would be required on average on the fault plane you use. So much slip has never been observed.

(2) Also an average of 25 meters of slip over depths from 0 - 50 km is already a lot and is only rarely observed. I wonder if the 25 meter estimate is biased towards high values by the simplified dislocation modelling assumptions that you use. Tsunami modellers regularly observe that unrealistic values of elastic parameters need to be used to make seafloor displacements match slip estimates from seismological sources.

(3) In Figure 5b the top panel also shows that the 25 meter line is constrained by very little data, because all data are located very near to the hinge line. Moreover, coincidentally your data are only located right above the top of the deepest part of the deepest fault patch (segment 2 out of 2). Your simplified modelling required you to make gross approximations to make these estimates

(see need major point). Those approximations in combination with the large extrapolations required lead to a large uncertainty on this 25m estimate and corresponding earthquake size.

(4) By slightly moving the location of that second linear patch to the right (trenchward) a much smaller deep earthquake with 9 meters slip could also agree with all of your data (see Fig. 5b). Since the actual slab interface is curved and already not very well imaged and located, such a 10-15km shift is well within the uncertainties. Following your suggestion of the kink "likely being located between 25 to 35 km depth" that translates to a horizontal uncertainty of 24km (using an average dip of $(30+15)/2 = 22.5$ degrees). Hence twice as large as the shift required to fit all data with a deep event that slips 9m.

- How can you robustly separate where slip occurred?
- And what true evidence exists for the presence of a kink over a curved geometry?

Simplified fully elastic modelling on a planar fault can not lead to accurate fault slip estimates

Modelling of vertical motions is difficult and we hardly ever model vertical motions correctly. Not even at the shorter earthquake cycles time scales we observe (e.g., Govers et al., Reviews in Geophysics, 2018). This is partially because we do not understand the larger-scale geodynamic processes that cause them in combination with fault slip (e.g., Govers et al., Reviews in Geophysics, 2018). You follow a suggestion to balance our inadequate understanding and modelling simplifications by largely changing the dip of the fault (from 12.5 to 31 degrees, which is very significant!) over the most relevant depth intervals of 0-33km. These very large changes to compensate for largely simplified modelling assumptions (e.g., full elastic medium, back slip, planar fault) can in my opinion not lead to robust estimates of fault (back) slip rates.

- We know viscoelastic medium behaviour significantly influences vertical motions (e.g., Wang et al., Nature, 2012, Weiss et al., Science Advances, 2019).
- As mentioned above, we know the interface is curved. The paper that you cite (Kanda & Simons, Tectonophysics, 2012) also explains that the hinge location and vertical motions are very sensitive to planar or curved faults. Since you have only a very narrow band of data very close to the hinge location, the usage of a planar versus curved fault is also said to have a very large impact on the location of the hinge location. Hence this would critically estimate your estimates, which required large extrapolation to that estimate the amount of slip (and hence earthquake size) somewhat. The impact of this assumption on the coseismic motions is also demonstrated by Ragon et al., GJI, 2018.
- We know geodynamic processes impact the vertical motions as well (e.g., Govers et al., Reviews in Geophysics, 2018).
- Additionally, the very distinct topography near the islands could play a role (e.g., Landers et al., Tectonophysics, 2020).

I know inverting for at least the first and/or second and/or third point requires state-of-the-art modelling that has become available since five years (see several works of Thea Ragon and Sylvain Barbot referred to below), but I think that level of robustness is justified for Nature Communications.

Moderate points

- Novelty of the data

As written in the abstract geodetic measurements and slow slip events suggest coupled patches along the megathrust and there is evidence for great tsunamis.

Hence the author provide another piece of the puzzle using a new and longer data set. However, it is not very clear from the text what the novelty of that data set is. The authors other paper in 2023 in GGG (below) have a similar introduction, but they then show similar data on a much shorter record from 1800 to 2020. Please better clarify in the introduction already that it is the duration of your record that makes the largest difference. You have that in the next paragraph, but then quantify the length of the other data sets in the first paragraph of page 2. For example for these two citations (i.e., 1 to 2 centuries rather than thousands of years)?

Weil-Accardo, J. et al. Relative Sea-Level Changes Over the Past Centuries in the Central Ryukyu Arc Inferred From Coral Microatolls. *Journal of Geophysical Research: Solid Earth* 125, e2019JB018466 (2020). - *last century*

Debaecker, S. et al. Recent Relative Sea-Level Changes Recorded by Coral Microatolls in Southern Ryukyus Islands, Japan: Implication for the Seismic Cycle of the Megathrust. *Geochem Geophys Geosyst* 24, e2022GC010587 (2023).: *time range is difference; 1800-2020*

Moreover, these texts in the abstract seem to contradict a sentence at the end of page 2, which states “ mainly aseismic “. It rather seems that observations are not in agreement and a new very long record is added to understand the earthquake potential and recurrence behaviour of this subduction segment.

- English language should be checked. Relatively often words are missing, or plurals are missing or added unnecessarily.
- “Elastic modelling *shows* that the observed motions are coseismic”. Modelling can not *show* that. Your simplified modelling only shows what scenario might and might now work. Hence it can not more than “*support*” your interpretation that motions are coseismic.
- Some terminology is not explained at a level suitable for the broad readership of *Nature Communications*. Examples include e.g., “total station” (p.3,4).
- Some sentences are incredibly hard to follow for the reader
 - For example, see the top few at page 6 .
 - The complete story in the “Seismic supercycles” section is not easy to follow and understand and has a lot of if’s present.
- Moreover, a clear explanations of the vertical motions of the tectonics and the coral and what then the relative changes mean is missing.
- Additionally, an explanation of the physical mechanisms governing super cycles is missing. If you want to suggest they exist a brief explanation of why they can exist would be useful for the reader. Papers such as Herrendorfer et al., *Nature Geoscience*, 2015, explain how stress build up near the down dip limit can explain quiescence periods, whereas subsequent earthquakes can transfer the stresses loaded from below. That transfer loads the entire mega-thrust and prepares it for complete rupture in a super event that you anticipate.
- P.6: Sudden coseismic uplifts of 210 cm are extremely large and much larger than the largest coseismic uplifts we have measured for e.g., Tohoku. This also contradicts other values in the paper, so is this reasonable? Could you also add mean, min, max values there, such that we can better interpret your values?
- P/6 You always use inter seismic rates of 1.4 mm/yr, but for the period from 1771-2018 you use 3.0 mm.yr without much explanations of how you get to these rates. You even name them “arbitrary”. Since this is key for some of your calculations latter it would be good to understand how you get to those estimates and what their uncertainty and the impact thereof is. Sometimes I wonder wether your work simply needs to be described with more details and requires a longer journal.

Minor points

- Improve several aspects of the figures
 - Fig. S2b is missing axis labels (please also check for other axis labels)

References

- 1.Govers, R., Furlong, K. P., Wiel, L., Herman, M. W. & Broerse, T. The Geodetic Signature of the Earthquake Cycle at Subduction Zones: Model Constraints on the Deep Processes. *Rev Geophys* **56**, 6–49 (2018).
- 1.Wang, K., Hu, Y. & He, J. Deformation cycles of subduction earthquakes in a viscoelastic Earth. *Nature* **484**, 327–332 (2012).
- 1.Ragon, T. & Simons, M. Accounting for uncertain 3-D elastic structure in fault slip estimates. *Geophys J Int* **224**, 1404–1421 (2020).
- 1.Ragon, T., Sladen, A. & Simons, M. Accounting for uncertain fault geometry in earthquake source inversions – I: theory and simplified application. *Geophys. J. Int.* **214**, 1174–1190 (2018).
- 1.Langer, L., Ragon, T., Sladen, A. & Tromp, J. Impact of topography on earthquake static slip estimates. *Tectonophysics* **791**, 228566 (2020).
- 1.Weiss, J. R. *et al.* Illuminating subduction zone rheological properties in the wake of a giant earthquake. *Sci Adv* **5**, eaax6720 (2019).
- 1.Herrendörfer, R., Dinther, Y. van, Gerya, T. & Dalguer, L. A. Earthquake supercycle in subduction zones controlled by the width of the seismogenic zone. *Nature Geoscience* 1–5 (2015) doi:10.1038/ngeo2427.

We thank the authors for their thoughtful revisions and detailed responses, which have strengthened the manuscript. The introduction of the confidence index and distinction between eroded and preserved HLG/HLS enhances the clarity and robustness of the record, and addition of event labels in Figure S2a makes the descriptions in the text easier to follow. The study provides important evidence of past tectonic uplift and potential subsidence along the southeastern coast of Ishigaki Island. We believe the manuscript remains a meaningful contribution to the field and offer a few additional suggestions for consideration prior to publication.

1. Erosion and sea-level tendency

It is now much clearer with preserved/ eroded HLG/HLS illustrated in Figure S2b. The interseismic rates are also now inferred using the authors' best estimate of the interseismic RSL trend using the longest, best-preserved record from NAG3_18H. However, eroded HLGs were still used to quantify coseismic uplift estimates, which we draw caution to.

While it is true that eroded HLG/HLS have been used to estimate RSL changes (as cited in the examples by Sieh et al. 2008 and Meltzner et al. 2015), the focus for this is to quantify interseismic RSL rates. The premise for doing so is that the coral only experiences a diedown once it grows close enough to sea level that a temporary negative sea-level anomaly/ drop in sea level can cause a diedown. Therefore, the HLG immediately preceding a diedown (i.e., highest point on the coral before a diedown) best approximates HLS, even if the highest points are eroded. Additionally, the pre-diedown HLG is less affected by the magnitude of a given diedown event (which can be influenced by the severity of a sea-level anomaly), compared to the post-diedown HLS. (See *Meltzner et al., 2010, section 3.3.2*)

In the examples regarding Sieh et al. 2008 and Meltzner et al 2015, coseismic uplifts were only estimated where there were preserved HLS points to indicate how far the corals died down to. Without a preserved HLS/ diedown, it is not clear if there truly was a diedown that later got eroded, or that there was no diedown to begin with.

We recommend that events G and H should be assigned a lower confidence index due to the lack of preserved HLS and lack of agreement with coeval records. During the timing of events G and H when NAG_M was inferred to have died down, NAG6_18H recorded continued upward growth, even though NAG6_18H was at a higher elevation. The two coeval coral records do not seem to agree, even considering the +/- 7 cm HLS variability, so it seems possible that events G and H in NAG_M reflect erosion horizons, rather than diedowns. The concentric nature of this "step-down" marking event G is also not clearly illustrated in the field photo of NAG_M provided in the rebuttal:

Therefore, we recommend that event G is downgraded from confidence index of 5 (the most confident) to a lower confidence, and hedge the wording regarding the “major sea-level decrease” and “second sudden sea-level decrease” in supplementary lines 142-144. This is unlike event D, where coeval diedowns were recorded in both NAG6_18Q and NAG6_21C (despite NAG6_21C not having a preserved diedown); and event E, where NAG6_21C died down completely around the same time when NAG6_18Q recorded a diedown (despite there being no preserved HLS in NAG6_21C).

Supplementary text lines 117-119: the two emergence events inferred are from eroded parts of NAG_B – there may not have been a diedown as the HLS is not preserved.

Figure S2a: The upward growth of a coral does not necessarily indicate sea-level rise tendency (as shown by black lines in Figure S2a indicating “apparent RSL increase”). The coral could still be catching up to sea level, during which upward growth reflects the coral growth rate, not sea level. Instead, it is the stepping upwards of successive concentric rings/ diedowns that indicate RSL rise, not the increase in elevation of successive growth bands. Similarly, RSL fall should be interpreted as successively lower concentric rings/ HLS, not successively lower growth bands.

Supplementary text lines 138-139: the words ‘first HLS’ were removed, and eroded HLG/HLS shown in Figure S2b, but the text and interpretation did not change (i.e., erosion is still interpreted as RSL fall, as indicated by the blue arrow in Figure S2b).

Supplementary text lines 180-181: similar comment to the above; RSL decrease of -1.4 mm/yrs is inferred from eroded HLG, but HLS are not preserved

Supplementary text line 191: similar comment to the above; RSL decrease is inferred from eroded HLG, but HLS are not preserved

2. Offset of NAG3_21E and NAG_B not explained by HLS variability applied

It is unclear from the text how the HLS variability of +/- 7 cm was derived from two corals with coeval diedowns. Which corals / diedowns was this based on?

In lines 102-104 of the main text, the current explanation for why NAG3_21E is lower than NAG_B seems to suggest that this offset could be explained by natural variability in the coral HLG/HLS. However, the elevation difference between the preserved HLS of the two

corals is nearly 35 cm at 301 yrs BC based on figure 3, which is larger and seems to contradict the the +/- 7 cm applied as the HLS variability.

Could NAG3_21E have slumped? Or NAG_B be ponded at a higher elevation?

Alternatively, is it possible that NAG3_21E was growing too deep compared to NAG_B, such that the uplift event that killed NAG_B (event Q) was not large enough to kill NAG3_21E? If so, then this would provide an upper limit on the amount of uplift for event Q. This would require that the 301 yrs BC and 329 yrs BC inferred diedowns were not diedowns. From figure S2a, the 342 yrs BC diedown is not shown on NAG3_21E, and the 301 yrs BC diedown in NAG3_21E is also not as clear as that in NAG_B.

Minor comments

1. **Line 99:** Should be “glacial isostatic adjustment” not “glacio-isostatic adjustment”
2. **Lines 181-183:** Please specify which corals were used to infer possible coseismic subsidence (it's indicated in figure 4 caption as “potential sudden RSL increase event”) but could be described better in the main text. It will also be helpful to add the inferred subsidence event(s) to table S2b
3. **Lines 234-236:** Consider adding that different glacial isostatic adjustment models could also produce variable RSL predictions.
4. **Supplementary text line 119:** 26' +/- 8 yrs BC seems to be a typo
5. **Supplementary lines 190-191:** NAG6_21C was described as starting to record HLS variations at 2436 years BC, but there was no preserved diedown/ HLS here.
6. **Figure 3:** Similar to figure 4, also distinguish the confidence index in the vertical lines that indicate diedown events?
7. **Figure 4:** Rather than bolding letters with confidence index 5, consider having distinct line styles for the different confidence indices.
8. **Please check that table S2b is consistent with figure 3.** For example, events N and O are indicated as >15 cm and > 18 cm in figure 3 and in the main text, but reported as ~15 cm and ~18 cm in table S2b. Please check for other events as well.
9. **NAG3_18H, events N and O:**
 - It seems odd that the erosion line was not extrapolated to the diedown for event N

- Why is the vertical downward dashed arrow for event O (which indicates the tectonic diedown) drawn from the HLG/ring crest **after** the diedown, not the pre-diedown HLG?
- Rather than crudely estimating the amount of erosion based on the dashed blue erosion lines drawn, since the uplift amounts are reported and described as minimum uplift estimates anyway, would it be better to simply take the difference between the highest (eroded) pre-diedown HLG and the preserved HLS elevation as the minimum uplift estimate? Alternatively, perhaps extrapolate the pre-diedown HLG from the RSL trend prior to the diedown, similar to the examples cited in the rebuttal for Sieh et al. 2008 and Meltzner et al. 2015?

10. The complete die-off of any individual coral microatoll colony may not always require a tectonic uplift event.

By “temporally discontinuous” records, we refer to the fact that even in relatively tectonically stable locations, coral microatoll ages often cluster into age clusters. This may be due to biological life spans of the coral microatolls, as according to Smithers (2011):

“Large microatolls demonstrate continuity of growth for several centuries, but most microatolls are < 2 m in diameter. This probably reflects the relatively low probability of extended survival in shallow reef environments where corals are most vulnerable to a range of debilitating conditions. Microatolls on the northern GBR surveyed by Scoffin and Stoddart (1978, p.105) averaged 0.5 m in diameter, although other than stating that they “vary in size from a few centimeters to a few meters” no statistical description of the variance was offered.”

Redacted

References

Hallmann, N., Camoin, G., Eisenhauer, A., Botella, A., Milne, G. A., Vella, C., ... & Fietzke, J. (2018). Ice volume and climate changes from a 6000 year sea-level record in French Polynesia. *Nature communications*, 9(1), 285.

Meltzner, A. J., Sieh, K., Chiang, H. W., Shen, C. C., Suwargadi, B. W., Natawidjaja, D. H., ... & Galetzka, J. (2010). Coral evidence for earthquake recurrence and an AD 1390–1455 cluster at the south end of the 2004 Aceh–Andaman rupture. *Journal of Geophysical Research: Solid Earth*, 115(B10).

Meltzner, A. J., Sieh, K., Chiang, H. W., Wu, C. C., Tsang, L. L., Shen, C. C., ... & Briggs, R. W. (2015). Time-varying interseismic strain rates and similar seismic ruptures on the Nias–Simeulue patch of the Sunda megathrust. *Quaternary Science Reviews*, 122, 258–281.

Sieh, K., Natawidjaja, D. H., Meltzner, A. J., Shen, C. C., Cheng, H., Li, K. S., ... & Edwards, R. L. (2008). Earthquake supercycles inferred from sea-level changes recorded in the corals of west Sumatra. *Science*, 322(5908), 1674-1678.

Smithers, S. (2011). Microatoll. In D. Hopley (Ed.), *Encyclopedia of Modern Coral Reefs: Structure, Form and Process* (pp. 691–696). Springer Netherlands. https://doi.org/10.1007/978-90-481-2639-2_111

Tan, F., Horton, B. P., Ke, L., Li, T., Quye-Sawyer, J., Lim, J. T., ... & Meltzner, A. J. (2024). Late Holocene relative sea-level records from coral microatolls in Singapore. *Scientific Reports*, 14(1), 13458.

The revised manuscript shows significant improvement over the previous version and has satisfactorily addressed most of the comments raised by me and the other reviewers. I have no further comments.

We thank the authors for taking the time and effort to address the comments raised. We find that the manuscript is now much clearer in its assumptions and inferences, providing important data to enhance understanding of seismogenic nature of the Ryuku subduction zone. It is especially interesting that the clusters of high-confidence abrupt uplifts coincided with the ages of tsunami deposits, with lack of both coral data and tsunami data between the inferred supercycles.

We only have a few remaining minor comments and recommend publication thereafter.

Main text

1. Line 101: please specify over what time period the quoted GIA rate is referring to.
2. There are places where there seems to be missing line spacing between paragraphs. Please check to ensure that the spaces are added accordingly (e.g., between lines 102–103, 135–136...)
3. Line 405: Here it says you use Zach-3 method, but in table S2a the footnote mentions you use the Zach-1 method. Please check to make sure it is consistent.
4. Line 436: Earthquakes inferred from within the coral slices are assigned with confidence rating of 5. While we appreciate that the authors downgraded the confidence ranking of event G to grade 4 to address our earlier comment, the confidence rating now is inconsistent with the description in the text (because event G is inferred from the slab but is assigned grade 4 and not 5). We suggest adding a new confidence rank to differentiate between diedowns inferred from *preserved* HLG/HLS (ranking of 6) in slabs versus those inferred from *eroded* HLG/HLS in slabs (ranking of 5).
5. Figure 1: panel labels are missing (e.g., 1a, 1b)
6. Figure 4: where are the dotted lines?
7. Figure 4: we appreciate the attempt to address our earlier comment by introducing different line thicknesses for different confidence rankings. However, the different line thicknesses are more difficult to read than the bolded letters initially. Could you use different line symbols (e.g., dotted, dashed, solid, twodash) to represent different confidence rankings, and add a legend for this?
8. Figure 4: please arrange the red line for event R to be on top of the black vertical line as now it is difficult to see the red line.
9. Figure 5: please indicate in the caption which corals/ uplift events were used to constrain the elastic dislocation models. Maybe I missed this but was this mentioned explicitly in the text? If not, please add a line to clarify this.

Supporting text

1. Line 209: change “at the sooner” to “at the soonest”
2. Table S2b: to make it easier for the reader, please add to the description in the table header that a confidence ranking of 5 is the best, so that the reader does not have to look for this in the text.

REVIEWER COMMENTS

Reviewer #1 (Remarks to the Author):

Debaecker et al. present relative sea-level (RSL) reconstructions from coral microatolls located along the southeastern coast of Ishigaki Island, located north of the Ryukyu subduction zone. Combining the coral slab interpretations with elastic dislocation modelling, the authors suggest several coseismic uplift events superimposed on millennial-scale long-term relative sea-level (RSL) fall that may be attributed to permanent deformation (irrecoverable uplift) over seismic cycles. Considerable effort was taken to provide detailed documentation to support the findings of this study, both in the main text and the supplementary material.

We believe that the coral record presented has important implications for advancing the understanding of the seismogenic potential of the southern Ryukyu subduction zone.

We recommend publication of the manuscript after addressing the following concerns.

We find that there are important details that need to be addressed, particularly with regards to the inferred coseismic uplifts. We believe that greater focus should be placed on the periods of RSL rise recorded internally within the coral slabs, which may indicate interseismic or postseismic subsidence. The uplift estimates are problematic due to erosion of the coral surface. Glacial isostatic adjustment (GIA) drives far-field RSL fall during the mid to late Holocene, which may explain a non-negligible part of the RSL fall that is currently inferred as long-term net land uplift. The inference of RSL changes between different coral generations (vs internally within a slab) is more challenging as it requires 1) clearer demonstration that the corals were in-situ and were not moved by past tsunamis; and 2) consideration of the natural variability in coral growth that is currently underestimated.

We provide detailed comments below.

Major comments

1. There seems to be a misinterpretation of diedowns, highest level of survival and highest level of growth, and missed recognition of erosion.

Lines 406-407: "... a HLS year corresponds to a truncated growth band, while the growth band is complete during a HLG year". A truncated growth band indicates erosion, not the HLS. The HLS is the level the coral dies down to during a diedown event. In cross section, the HLS can be recognised as an unconformity, where the annual bands are seen to curl over at the diedown (resulting from outward growth at the HLS after the diedown).

There is a misunderstanding on the understanding of what we call HLS. A coral may record the HLS as an unconformity in its morphology (an unconformity in the microatoll internal stratigraphy) when it reaches its HLS (Zachariasen, 1998) but this unconformity may be eroded. We used the term HLS for both preserved and eroded HLS. Depending on the studies, various markers are defined in the coral morphology (pre-diedown HLS or HLG, post-diedown HLS, HLG eroded or Preserved HLG, HLS. See legends of Figures below from Sieh et al. 2008, and Meltzner et al., 2015).

Depending on the methods (that we described in the method section), various markers are used to estimate the relative sea-level changes. Some studies used only the HLS after diedown to estimate the RSL (Zachariasen et al., 2000) as they consider that they are better preserved from erosion by coral regrowths after diedown, others use non-eroded HLG before diedown (Meltzner et al., 2010). Other estimated the RSL by using eroded HLS (Sieh et al., 2008), the estimation is less robust however, unless the erosion is small. It is sometimes difficult to estimate the erosion, it is why the best-preserved corals were slabbed in this study. The shape of the coral (cup, hat, flat) is a first indicator of the sea-level changes.

We followed the advice of the reviewer and distinguish the best-preserved HLS level from others to further use them to estimate the sea-level variations. We changed the related text in the Method section, and all text, figures and tables of supplementary materials S2. This slightly modified the interseismic rate used for the seismic cycle model in Figure 4.

In Text S2 of the supplemental information, years that were described as the 'first HLS' being reached were marked by the first (oldest) annual bands that were truncated (supplemental lines 143-144; 146). The first HLS should be where the first diedown is recorded by the coral, not where the first annual bands get truncated.

This point has been addressed.

In some studies, depending on the coral morphology and if well preserved, the first unconformity in the coral stratigraphy can be considered as the first HLS (see Philibosian et al., 2022). We however addressed this point to have a better estimation of the RSL changes but the results remain unchanged.

In lines 406-409, HLS was described first, followed by a separate explanation of diedowns as 'centimetric sea-level drops', suggesting the authors do not recognise that HLS are only formed during diedowns.

Again there is a misunderstanding here on the term HLS we used. We clarified this in the Figures and the text. In the literature, for example Zacharisen (1998) wrote "If at some point the coral reaches its HLS, it will, for the first time, record the HLS as an unconformity in its morphology". It is true that this unconformity may be eroded (Sieh et al., 2008) leading to errors in the estimation of the relative sea-level changes (RSL). It is why "preserved HLS" are used to better estimate the RSL. In the literature (Sieh et al., 2008; Meltzner et al., 2015), HLS occurs before die-down (they are mentioned "eroded HLS" but are used sometimes to estimate first order RSL changes, see attached figures).

As a result of the misunderstanding of HLS, several consecutive years were misinterpreted as HLS years and therefore stable RSL (Figure S2b) (supplemental material lines 119; 150; 188; 199), when in fact we cannot know because of erosion of the top of the coral.

We believe that Eroded HLS could be used to crudely estimate the RSL changes as in Sieh et al. (2008), see Figure below. However, we followed the recommendation of the reviewer and discarded the eroded HLS or HLG from the reconstruction of RSL changes (we however indicate the tendency by a black line).

Additionally, in several instances erosion was misinterpreted as relative emergence/ periods of RSL fall (supplemental material lines 123; 200; 213; 237). In these cases, often the slab profiles indicate the truncation of near-vertical growth bands, which are a clear sign of erosion. Near the base of a coral, the growth direction is near horizontal, so the annual bands (which are perpendicular to the growth direction) are near vertical. If there were no erosion, the surface of each concentric ring of the microatoll should show annual bands that are near-horizontal, as growth direction is near vertical at the ring crest.

Again, this is a misunderstanding. In papers such as Sieh et al. 2008 (see Figure below) eroded HLS unconformities (i.e. truncations, red squares) have been used to estimate crudely the RSL changes (0.1 ± 0.3 mm/yr) by using well preserved medieval coral. The authors included error bars that they were able to quantify them. In the paper by Meltzner et al., (2015), they identified eroded HLS or HLG but also HLS before diedowns (see caption of the figure below) and inferred the RSL from pre-diedown HLS or HLG sometimes eroded. We may have some erosion in some cases, but the slabbed corals are well preserved and we are confident that we can estimate their morphology in the field (cup, hat, flat) in absence of regrowth colony. In that case, we have a first order estimation of the RSL changes. We followed the advice of the reviewer and remain conservative on our estimation of the interseismic RSL trend and used the best estimate in our model in Figure 4.

REDACTED

Redacted

Figure extracted from Sieh et al. (2008).

Figure S2b:

- many 'eroded HLS' points are actually 'eroded HLG'
- *Some of the 'HLG' points are eroded, but there is no distinction between eroded HLG and preserved HLG in this figure. HLG is preserved when the ring crest of a particular annual band is preserved; if the annual band is truncated, then the HLG in that year is eroded.*

We now better distinguished eroded and preserved HLG and we modified our interpretation of the slabs draws accordingly.

2. The authors do not account for natural variability in the coral HLG/ HLS.

*For the RSL reconstruction, only two living microatolls were surveyed (line 343). This likely does not capture the natural variability in microatoll HLG. The authors report levelling uncertainties (line 348) and a vertical uncertainty of +/- 3 cm on a single concentric ring (lines 348-349), but it is unclear what the latter refers to. Published studies indicate that HLG of *Porites* sp. microatolls may vary by up to about +/- 20 cm across a study site. When reconstructing centimetric-scale RSL changes between different corals this uncertainty in the natural variability of coral HLG becomes important.*

The mean uncertainty of ± 3 cm refers to the HLG variability per microatoll.

We agreed with the reviewer that living microatoll measurements are not sufficient to estimate the natural variability of their HLG (we also made a mistake in Figure 2 by adding the elevation of a hemispherical coral head that we removed in this revised version). In Nagura, no variability in microatoll HLG can be directly or exactly measured from our survey. We used the average difference between growth band elevation between two corals for coeval diedowns, which is of ± 7 cm. It is consistent with the HLG variability per site measured in sites with comparable tidal range (between 5 to 15 cm for tidal range between 1.1 to 3 m, the tidal range in the Ryukyus being of 2 m (Meltzner and Woodroffe, 2015; Hongo & Kayanne, 2009; Tsuji, 1993; Yamano et al., 2019). It is also consistent with estimations from studies from nearby sites in the Ryukyu archipelago (Debaecker et al., 2023; Weil-Accardo et al., 2020). Those sites show standard deviation around the living HLS between 1 and 7 cm, ~ 4 cm in average.

Related to the above, the authors also do not mention the genus of corals being studied. Different genus can survive at different elevations with respect to the tides. Can this explain the vertical offset between the corals NAG_B and NAG3_18H? Based on the caption of Figure 4.2 in the Yamaguchi et al. 2016 chapter, it seems that there were fossil Porites sp. and Favia microatolls at this site.

Genus of all dated corals, slabbed and sampled, is *Porites*. They have been identified by Dr. Kohki Sowa, and by Dr. Atsuko Yamazaki (Nagoya University/ Kikai Institute for coral reef sciences). We warmly thank the latter for her help.

Difference of genus can therefore not explain the vertical offset between corals NAG_B and NAG3_18H.

There are several hypotheses on why NAG_B is up to 40 cm higher than NAG3_21E:
- NAG_B may have been ponded, and the geomorphic feature that ponded the microatoll (and probably other corals nearby at the time) broke when NAG_B died, at 258 ± 7 years B.C. NAG_B is closer to the shoreline. This is unlikely however, as there are no traces of past geomorphic features that could strengthen this hypothesis.

- This difference in elevation occurred after the death of both corals, although it seems unlikely since they are at ~ 800 m from each other.

3. Possibility of reworking

This is an area with historical tsunamis (Figure 4b). Is there a possibility that the corals could have been transported during a tsunami? NAG6_21C looks like a fragment of a coral that was broken off and was found within the main river channel. Is it possible it was transported during the Yaeyama tsunami event?

A tsunami would indeed have the energy to carry corals blocks on the land. However, in Ishigaki, evidence of tsunamis in the bibliography (such as the Meiwa tsunami for example) have shown that transported boulders and other tsunami traces are only observed in the southern and western coasts of the island (see figure below, Suzuki et al., 2008; Goto et al., 2010). This distribution seems also supported by numerical modelling of tsunami wave propagation led by Hisamatsu et al. (2014). We found no evidence of reworking in the area.

Redacted

Figure extracted from Goto et al. (2010).

Furthermore, NAG6_21C is at similar elevation than NAG6_18Q, and their RSL records are strikingly similar. Its lower part was buried, it did not show any tilting. These corals are in-situ.

Photograph of NAG6_21C, showing its underwater buried lower part. We extracted part of the sand before taking the picture.

Were there dated fossil corals at similar elevations to those slabbed, which would support the in-situ nature of those slabbed corals? Currently, it is difficult to discern based on the plateaus shown in Figure 3.

The plateaus are presumably combined from different dated (but not slabbed) microatolls? It may be more useful to show individual sea-level index points for each dated microatoll.

Plateaus as shown in Figure 2 and in Table S1c are topographic features. We did not consider the ages of their corals to define them. For example, plateau 12 is composed of several colonies that were dated at 367 ± 8 and 1324 ± 23 years B.C. We refined our definition in lines 95-96.

Our dating strategy when investigating Nagura site was to date corals at different elevations in order to get the most complete RSL history. However, several dated coral boulders were found at similar elevation than those slabbed, with comparable age considering their uncertainty:

- Given their age uncertainties, plateau 9 dated with NAG6_21A and plateau 10 dated with NAG5_21A are consistent with RSL records from NAG_M, and NAG6_21C, NAG6_18Q, NAG_M or NAG6_18H, respectively.
- One colony of GIANT coral, GIANT_18, in plateau 12 is associated with RSL record of NAG_B;
- Plateau 17 dated with NAG3_18B sample can be associated with NAG3_21E RSL record.

Simplified Figure 3 showing coeval plateaus with sliced microatolls.

As coseismic uplift events were inferred from the relative elevations of dated coral blocks, we believe it is also important to consider if these dated coral blocks were in-situ.

We agreed with the reviewer on this point, and we precised in the text (Method section) and in Figure 2 evidence supporting the in-situ nature of the coral samples.

We believe that the microatolls located in the mangrove, that are higher in elevation and older (~4 ka), are in-situ for 2 ka (minimum) because they are partially buried under the mangrove peat that settled 2 ka ago (Fujimoto et al., 1995; Miyagi et al., 2003). Microatolls located in the bayside of the site, that are lower and younger (<2 ka), are also partially buried, but in this area the sedimentation rate has not been investigated yet.

In the figure below, we present in a more general way the spatial distribution of topographic plateaus we identified, with indication of their age range base on U/Th results (Table S1c).

Map of the distribution of microatolls according to their age and elevation in Nagura. This figure has been included in the revised version of Figure 2.

This figure shows that the spatial distribution of the fossil corals can be related to their age at first order, and to their elevation. This observation supports the in-situ nature of the corals.

4. Coseismic uplift events

The authors reconstructed coseismic uplift events by assuming an “arbitrary interseismic RSL rate of 1.4 ± 0.2 mm/yr” (line 139), supplemented by coral analyses (Table S2b).

The application of this arbitrary interseismic RSL increase (i.e., interseismic subsidence) forces coseismic events even when the coral elevations and ages themselves may not require so. Events L, M and N were recorded by dated plateaus

at rather similar elevations. However, these were inferred as coseismic uplifts because of the assumed interseismic RSL rate. It is possible that this period was marked by gradual RSL fall, rather than coseismic uplift followed by interseismic subsidence.

We have strong evidence from sliced slab corals that we have large sea-level drops due to earthquakes. We also inferred from well preserved HLS an interseismic strain rate of 2 mm/yr (this new estimate is the RSL we inferred from four preserved HLS bands in NAG3_18H coral, as suggested by the reviewers). We inferred that each generation of corals composing the plateau were killed by earthquakes and mark a past HLS at the scale of the entire site. With this simple hypothesis, we present a model of possible seismic cycles on figure 4: the corals composing the plateau were killed partly or entirely by earthquakes and the interseismic strain rate is constant. It is why we inferred that Events L, M and N are coseismic.

The RSL between events L and N could indeed correspond to a gradual fall. However, this would not account for the sudden death of all microatolls composing the plateau. Their death could be either tectonic-related, as we suggest, or climate-related. Reconstructions of past climate in the Ryukyus are rare and not well constrained, but to our knowledge, none of them document significant sudden climatic events (Sowa et al., 2014; Garas et al., 2023)

We also agree with the reviewers that applying an arbitrary interseismic RSL rate between the documented events is a strong assumption as it has been shown that interseismic rate can vary through time (Meltzner et al., 2015) but again we are presenting a model as simple as possible to discuss process at the megathrust at mid to long term time scale.

Of the 22 uplift events reported, only three were supported by RSL fall internally within the slab profiles (events E, G, and O). A third was not inferred but is evident in the NAG6_18H slab (the annual band for year 2089 that was sampled for the U/Th date A2); event J was inferred a few years later than year 2089 but the annual bands do not suggest a diedown in the year of event J.

For events E, G and O, the slab profiles suggest evidence of a RSL fall, as indicated by a step-down in elevation from the pre-diedown HLG (be it eroded or preserved) to the HLS at the diedown. However, the HLS in events G and at the 2089 band in NAG6_18H were not preserved, so it is also possible that the step down in elevations are from erosion rather than a true RSL fall.

The pre-diedown HLG in all of events E, G and O are not preserved, so the inferred uplifts can only be minimum uplift estimates. The diedowns for events O and G also appear from the slab annotations to be a few years earlier than is currently inferred.

-First, there seems to be a misunderstanding concerning the event J. The reviewer mentioned the date “year 2089” for a diedown event. Indeed, this diedown occurred at 2137 ± 20 years BC. This age came from U/Th ages, slice analysis, band counting and uncertainties (growth band counting uncertainty method from Weil Accardo (2014), uncertainty estimation from Meltzner and Woodroffe (2015)). We agree that the representation is confusing as the dashed red line indicating the 2137 ± 20 date of the diedown seems to occur few years before the obvious step-down in the morphology of the upper part of the coral. This is because the growth bands are undulating. The diedown in the morphology is thus not in correspondence with the growth band dated at 2089 ± 102 years BC.

We applied the same thinking to NAG_M and NAG3_18H, from which events O and G were inferred.

The reviewer also mentioned that corals may have been incised by erosion creating apparent diedowns. It is true that the growth bands before and after the die-downs G and K (now referred as event J) in slabs NAG_M and NAG6_18H, respectively, are eroded. It is also difficult with the slab stratigraphy to prove that these events are coseismic sea-level drops. It is why we carefully checked before slicing that the step-down in elevation we observed on the colony was present all around the microatoll. We are thus confident that these events are not related to erosion. We present here photos of NAG_M and NAG6_18H showing this step-down at other places than the slab axis. It is unlikely that erosion explains such morphology of the microatolls. Moreover, this drops also coincide with the altitude of other dead microatolls in the area, suggesting massive deadly events which amplitude attest for a coseismic event.

Photograph of NAG6_18H after extracting its slice. The location of the slice can be seen in the upper left part of the photo, whereas we measure the step-down in elevation elsewhere around the coral.

Photograph of NAG_M, showing its buried lower part.

We agree that the erosion of the growth bands before and/or after the die downs observed in the slabs are important to consider when characterizing the uplift amplitude. For events E and O (now event N), in NAG6_18Q and NAG3_18H slices, respectively, the growth bands after the die down are preserved from erosion, the uplift estimate is thus a minimum, which we indicated. For events G and K (now event J), growth bands after the die down are eroded.

Event D is obscure because of the “non-uniform growth” that was described in lines 184-185 of the supplementary material. If the diedown was indeed at the top part of the discontinuous growth, then the diedown is small and may be due to other oceanographic/climatic drivers rather than coseismic uplift. However, the annual bands in the lower portion of this discontinuous growth appear to be curling in a different direction than the upper part. Is it possible that the upper part is out-of-sequence overgrowth?

We thank the reviewer for this remark. In NAG6_18Q, the gap between the top and lower part of the discontinuous growth is big (at least 10 cm), and the lower portion is preserved from erosion as it shows clear shapes of non-eroded HLG growth bands. For this discontinuous growth, erosion seems unlikely, and the hypothesis of a death of the coral on its middle part only is less consistent than the hypothesis of an overgrowth.

We re-interpreted the slab accordingly, with a diedown at 2427 ± 10 years B.C. of 30 cm, that is clearly more consistent with the coeval diedown of 27 cm at 2423 ± 9 years B.C. observed in NAG6_21C slab.

For event P, the pre-diedown HLG is also eroded, therefore the uplift for this event should also only be treated as a minimum uplift estimate.

There appears to have been continuous upward growth between events O and P. It is possible that postseismic or interseismic subsidence after the uplift in event O outpaced the coral growth rate, and the coral only caught up to sea level during event P. Event P may also be due to other oceanographic/climatic drivers rather than coseismic uplift, as the magnitude of the diedown appears small.

We agreed with the reviewer that because of the erosion, uplift for event P is only a minimum estimate, and we indicated it.

We believe that the coral regularly caught up to sea level after event O, as we observed preserved HLS bands at 502 ± 13 and 482 ± 13 years B.C. Those bands were affected by diedowns of at least 7 cm and ~5cm, respectively, that may be climatic events. However, we believe that events O and P (now events N and O in the revised version) are tectonic events, because considering the arrangement of the growth band, the slab appears to have been significantly eroded before the event (see dashed blue lines in Fig. S2a). This implies that the amplitude of the diedown is underestimated and may be larger than 10cm. We indicated this in the HLS curve of NAG3_18H in Fig. S2b.

For event R, the HLG/HLS is not preserved, so it is not clear there was a diedown (or RSL fall), and it should not be used to quantify or characterize coseismic uplift. Based on the slab, there seems to be two to three earlier diedowns before event R, and one after event R – all of which seem to be separated by a similar number of years. This suggests that if event R were a diedown, it is likely to be caused by periodic oceanographic drivers, rather than coseismic uplift.

Event R is coeval with a diedown observed in NAG3_21E, but after careful re-consideration of erosion in the slab stratigraphy, we consider that the occurrence of a diedown, whether it is related to climate or tectonic, cannot be interpreted from this slab only.

Event R is inferred from the deaths of NAG_B coral, as well as NAG3_18B (Table S1b). Although both corals do not have the same elevation, we consider that their death followed a sudden and large RSL fall related to tectonic rather than a climatic event. We applied the same argument for events F, H, K, Q and S (now events F, H, J, P and R in the revised version)

Six uplift events were inferred from the thickness of the outer margin of the coral, which was assumed to have resulted from an uplift event that killed the coral entirely (events F, H, K, Q, R for NAG_B, S). It is common that coral microatolls produce RSL records that are temporally discontinuous, and which cluster in certain time periods. Therefore, we are not certain that the death of a microatoll necessarily implies that it was killed by an uplift event.

We are not certain what the reviewer means by “temporally discontinuous” RSL record. The growth rate of microatoll can change, and as a result the space between the growth bands can temporarily and significantly change too, but this is not what we observe in our slabs. This is unlikely that the coeval death of the entire microatolls we sampled is climatic and this for two reasons. The first one is that we observed, measured and sampled numerous modern microatolls in several islands of the Ryukyus and we never observed such a massive death microatolls that would have result in sudden drastic change of their conditions of survival, 2) We could still explain

the death of corals by a sudden change of climatic conditions (thus without fall in RSL) but we observed sudden sea-level drops in the internal stratigraphy of the slabs and dead corals forming plateau as markers of past sea levels. The simplest hypothesis is that the microatolls were all killed by significant sea-level drops. It would be difficult to prove this with only one microatoll but we have here an entire site with dozens of dead microatolls at various elevations. Also as of today and to our knowledge, there is no particular late Holocene sudden (less than a year) climatic event that may have promoted a massive death of corals in the Ryukyus. We lack conclusive evidence to support the hypothesis of a climatic event, whereas we document here evidence of uplifts during the microatolls' life. The death of NAG6_21C, which corresponds to event E, is coeval with a die down of at least 7 cm in NAG6_18Q, which is close to the 10 cm proposed by Taylor et al. (1987) to decipher between climatic and tectonic events from slab analysis. Furthermore, a large earthquake is documented around 0 years B.C. (Kawana, 1985; 1987), which could correspond to NAG3_21E death. To remain conservative, we established a confidence index for the events we documented. The values of the index vary from 1 to 5. Index 5 is for drops inferred from slab analysis, index 4 corresponds to drops inferred from death of microatoll associated with other undated colonies, and events defined by the death of only one fossil were attributed a confidence index of 3 (Table S2b). Index 2 refers to the corals documented by Yamaguchi (2016) that we did not observe ourselves, and index 1 corresponds to the hypothesis that undated plateau 21 could correspond to uplift generated by the Meiwu event. We downgraded by one point the index of events with large uncertainties on their amplitude.

Several uplift events (Table S2b) were inferred from the relative elevations of dated coral blocks (i.e. sudden 'drops' in the elevations of younger corals): events A, B, I, J, K, L, R, T, U. We suggest further investigation into the possibility that the lower corals were eroded.

We are not entirely certain we understand the reviewer's concern here. If the comment refers to the possibility that erosion might artificially enhance the apparent elevation drops used to infer uplift events, we would like to clarify the following. We have explicitly addressed the uncertainties related to drop amplitude estimation both in the main text and in Table S2b. While we agree that localized and substantial erosion could bias some measurements—as indeed observed in slab NAG3_18H—we note that in this case, the concentric growth rings and preserved HLS bands in the stratigraphy argue against significant erosion having affected the key features. For coral boulders, a full assessment of potential erosion would require slabbing and internal inspection, which is beyond the scope of this study. However, we were careful to select and sample boulders showing minimal external signs of erosion.

5. Attribution to RSL drivers

Overall, the elevations and ages of the corals in figure 4 appears to indicate a gradual RSL fall between ~3000 B.C. and present, apart from the two low coral blocks marking events T and U. Can the magnitude of the long-term RSL fall be explained by glacial isostatic adjustment (GIA)? Given that the study site is in the far field, a mid- to late Holocene RSL fall from a highstand can be expected. We suggest showing a RSL predictions from GIA models to illustrate the magnitude of RSL fall that can be expected from GIA.

We agreed with the reviewer. In the long-term RSL fall we document, there is a

component that can be attributed to the GIA (we already mentioned this in the first version of the manuscript). In the southern Ryukyus, only one study tried to model its effect in nearby island (Iriomote island, ~20 km westward from Nagura site, Yokoyama et al., 2016). Their results show lower RSL fall, which would indicate that part of the RSL fall could be accounted by tectonic process.

As asked by the reviewer and thanks to the kind help of A. Henry and J.C. Grall., we now provide a first-order RSL variations curve related to GIA (see below). They used VM5a model (Peltier et al., 2015), with a lithospheric thickness of 90 km and a melting at 6 ka, as our results in Figure 4 suggest.

GIA reconstruction in Nagura, Ishigaki island from VRMa model (Peltier et al., 2015). The GIA-related RSL decrease rate for the last 5ka is of -0.040 ± 0.001 m.

Both our long-term RSL curve and the ones from GIA studies have similar patterns, but different RSL rates (-0.4 m in our reconstruction versus -0.04 m in this GIA model). It is possible, although unlikely, that the entire long-term uplift is related to the GIA. However, we have clear evidence of Quaternary tectonic uplift in Ishigaki as there is a set of uplifted quaternary reef terraces (Ota and Hori, 1980).

We are more cautious in the revised version about the origin of the long-term tectonic uplift as it is difficult to separate the local tectonics from the GIA signal.

Within this record, the slab profiles of NAG6_18Q and NAG3_18H indicate clear and abrupt RSL fall followed by gradual RSL rise; and that of NAG_B indicates gradual RSL rise. We believe that these periods of postseismic/interseismic subsidence (RSL rise) are more robust than the coseismic uplifts, whose magnitudes are not well-constrained due to erosion of the pre-diedown HLG.

We did indeed analyze the deformation trends over interseismic and postseismic periods, including through first-order modeling of their potential sources (Figure S3b). While these phases provide important context, we acknowledge that there are uncertainties on coseismic uplift magnitudes due to erosion. Nevertheless, they remain

sufficiently resolved to allow for meaningful interpretation and first-order modeling of their sources.

We would like to emphasize that the primary aim of our study is to demonstrate the existence of coseismic events and seismic supercycles.

Minor comments

Line 28:

Remove hyphen from “relative sea-levels”. As a noun, sea level should not be hyphenated.

Done.

Lines 30-31: “Elastic modelling shows that the observed motions are coseismic uplift associated with megathrust earthquakes.”

Rephrase this to suggest that there are other possible interpretations. In this study, a constant interseismic rate was assumed between majority of uplift events, but this is only one possible mechanism.

Done.

Line 46: “A seismic coupling of 20% was inferred in this area.”

Clarify what “this area” refers to.

“This area” corresponds to partially coupled patches on the megathrust identified with microatolls, gravimetric or geodetic studies in the Ryukyus. We clarified it.

Line 52: “This study provides the first example of multiple earthquake cycles over five millennia in a subduction zone.”

Is this referring to just the Ryukyu subduction zone? Or anywhere in the world?

It was referring to anywhere in the world, but we forgot to precise that it is the first example from microatolls study (For example, study from turbidites along the Cascadia subduction zone infer earthquake cycles over ~10 k (Goldfinger et al., 2012)). We clarified our point.

Line 71: “... its upper part dies and it keeps growing laterally”

Rephrase this. After the coral dies during a diedown, the coral polyps grows out radially (both upwards and outwards), not just laterally.

We corrected this part.

Line 80: “Photos of the sites, slabs, or samples of each microatolls.”

Correct “each microatolls” to “each microatoll”.

Done.

Line 93: “... measured with an accuracy of +/- 6 cm”

Is this referring to the precision of the instrument, rather than accuracy?

Yes it was. We apologize for this English mistake.

We reassessed our estimation of uncertainty on the elevation for each coral (see Method), which led to uncertainties between ± 5 cm to ± 7 cm.

Line 125-126: "..., it grew 30 cm below and was too deep to record RSL drops of the same magnitude."

Even if so, NAG_B should have died down to a similar elevation (within the natural variability of HLS) as NAG3_21E, just that the magnitude of the coeval diedown in NAG_B is larger.

In figure 3, it appears that NAG_B died down to an elevation (-0.3m) that was 40 cm higher than the elevation that NAG3_21E died down to (-0.7m) at ~256 B.C.

We modified our sentence to specify NAG_B is 30-40 cm lower than NAG3_21E today.

Line 137: "... the external parts other microatolls".

Add "of" between "external" and "parts".

Done. We added "of" between "parts" and "other".

Line 139: " We used an arbitrary interseismic RSL rate of 1.4 +/- 0.2 mm/yr (as determined from the best-preserved signal from NAG_B, Table S2a)..."

In figure 3b, the interseismic rate labelled for NAG_B is 1.4 +/- 0.4 mm/yr, not 1.4 +/- 0.2 mm/yr as is described in lines 139 and 421; please make consistent.

Indeed, there was an error, thank you for noticing. We corrected it, as we changed the interseismic RSL rate we use to that estimated from preserved HLS bands from NAG3_18H slab, 2.0±0.6 mm/year.

Table S2 a does not indicate the rate 1.4 +/- 0.2 mm/yr, but it is described in the footnote of Table S2 that "Underlined values from NAG_B coral are used to reconstruct the RSL at a larger time scale in Fig. 4". Explain how the 1.4 +/- 0.2 mm/yr was derived from the underlined values (1.6 +/- 0.2 mm/yr and 1.2 +/- 0.2 mm/yr), and how those specific underlined values were chosen?

Those values were averaged. We chose them as they correspond to different radii of a same coral (NAG_B), and they were estimated from relatively long period (~80 years) of record that included what were interpreted as HLS growth bands.

Since we corrected our slab interpretation and now consider that the record of NAG_B during this period is mostly eroded, we turn to the RSL trend estimated from preserved HLS bands in NAG3_18H band during ~100 years (513-417 years B.C.). This rate corresponds to 2.0±0.6 mm/year, which sensibly increase our estimation of the amplitude of inferred seismic events. This is the longest, clearest, and most consistent signal we could estimate from our slab analysis.

Line 145: "ages of the undated plateaus are bounded by those of the plateau just above or below"

Is there a stratigraphic relationship between the corals (i.e., does "above" and "below" refer literally to corals stacked on top of one another?)

The extrapolation of ages of "undated plateaus" would make sense only if the stratigraphy supports it: a coral that grew physically on top of another coral must be younger than the coral beneath it.

In Figure S1j, it appears the corals were all in separate locations. If this is the case, the relative ages of undated corals to the dated corals is not clear.

We made a wrong use of the words "above" and "below", as we meant "higher" and "lower" in elevation, respectively. We corrected this.

We did not observe any coral growing above another one, as the reviewer saw in Figure S1j, all corals are separated from each other. The hypothesis we made to

propose an age to undated plateaus is weak, but this is, to our sense, the most consistent one. Age of undated plateaus is only used to propose an alternate solution to the large slips documented for events M and S (now events L and R, lines in Figure 4), but they are not used in further interpretation of events.

We made an exception for event V (now event U), which corresponds to documented Meiwa earthquake, and that may have caused uplift in Nagura site.

Line 207:

Typo mistake here "(fault 2) 2"

Done.

Line 228:

As per line 145, the sentence here "Some old corals are below younger generations" is vague. Does "below" mean that some corals are physically stacked on top of other older corals below?

We made an English language mistake here. We meant lower in elevation. We corrected this.

Line 231:

Given that the site is located at the mouth of a river, and some corals are located amongst mangroves, can there also be sediment compaction?

The Nagura mangrove swamp developed at ~2 ka and consists today in a thin layer of ~30 cm of mangrove peat (Fujimoto et al., 1995; Miyagi et al., 2003). This mangrove peat is found above the lower part of oldest corals, which are >40 cm of thickness and which are located in the mangrove area of the site (for example, external stepdowns of NAG_M and NAG6_18H corals).

Younger (~2 ka) corals, located in the bay area, are located on consolidated sandy substrate and we estimate that compaction is also unlikely there.

Lines 254-255: "Although we cannot decipher between shallow and deep dislocation models, the absence of tsunami traces could be in favour of deep ruptures for the smaller earthquakes."

The "absence of tsunami traces" seems contradictory given that the paragraph talks about documented tsunamis. Please clarify.

We were referring to quiescence periods before the failure sequence in supercycles 1 and 2. We modified our sentence.

Lines 269-270: "... and not of the same category as the largest earthquakes interpreted in the area."

The phrase is vague, it is not clear what "the area" and the "largest earthquakes" refers to.

Done.

Lines 270-271, 277-278:

Add line spacing between paragraphs.

Done.

Lines 281-282:

If interseismic subsidence is currently occurring, is this evident in the geomorphology of the site?

Are the living microatolls recording continuous relative sea-level rise?

Are there other independent drowned landscapes?

Few other clues could indicate relative submergence around Nagura. Living microatolls are recording relative submergence in the southern Ryukyus (Debaecker et al., 2023). Studies of traces of past mangroves may also indicate a lower RSL in the past: Yamauchi et al. (1995) documented sedimentation close to Nagura site at 143 to 193 cm deep that was dated at 1920 ± 100 years B.P. (i.e. 30 ± 100 years A.D.) that they assimilated with mangrove traces. Other studies also infer seaward displacement of mangroves around 1 and 2 ka in Iriomote and Ishigaki islands (Fujimoto and Ohnuki, 1995; Fujimoto, 1997).

Line 297: "Fossil microatolls permit to have extremely useful..."

A subject is missing between "permit" and "to"?

We changed the verb to "provide".

Line 299:

"Stoddard" is spelt wrong

We corrected this mistake.

Line 300: "Microatolls grow at a constant rate underwater"

Corals can grow at variable rates throughout their lifetimes, causing the annual band thicknesses to vary across the microatoll.

The growth rate of microatolls is, at first-order, constant. This is also observable in X-Ray and CT scan images of the coral slabs. As this sentence is here in a general paragraph presenting in a simple way the microatolls, we specified that this is a first-order assumption.

Line 301: "their development is controlled by sea-level variations..."

Please change "sea-level" to RSL, to clarify that corals microatolls also record land-level changes.

Done.

Line 303: "excessive subaerial and sunlight exposure"

It is not clear if excessive sunlight is needed for a coral microatoll to experience a diedown

We agreed with the reviewer, we may have been too assertive here. We do believe that excessive sunlight can desiccate more quickly a microatoll, but we lack of strong evidences to say so. We corrected this sentence.

Lines 303-304: "Once the HLS is reached, the growth is restricted to the horizontal direction"

This statement implies that the coral simply grows upwards until a limit (the HLS), then proceeds to grow horizontally. In reality, the HLS is the elevation a coral dies down to (from its pre-diedown highest level of growth), rather than the elevation it grows up to. Additionally, following a diedown, the coral polyps continue to grow radially outwards (both horizontally and vertically).

We were not precise enough in our description of the formation of microatolls and we thank the reviewer for this correction. We changed our description.

Line 318: "the coral microatoll record a dark ..."

Add an 's' to 'record'

Done.

Line 320: "will provide information on the HLS of the coral ..."

change "HLS" to "HLG/HLS". Depending on whether there was a diedown in a given year, the annual band could be representing HLS (during a diedown), or HLG (between diedowns).

Done.

Lines 332: "The most recent RSL record"

The term "most recent" is vague. Modern?

We thank the reviewer for this suggestion, we made the change.

Lines 339-340: "the altitude of the external ring (younger part of the microatolls, i.e., most recent HLS)..."

The outermost ring crest is not the most recent HLS, it is the most recent HLG. For fossil corals, the outermost ring crest is the most recent pre-diedown HLG (HLG just before the most recent diedown).

We corrected this error.

Line 344: "... one being ball-shaped"

What does a ball-shaped microatoll mean? Does it have concentric rings? If not, is it just a hemispherical coral head, and not a microatoll?

Indeed, one of the living corals we surveyed is ball-shaped, which means it is only a hemispherical coral head. This coral only indicates a minimum level for the modern HLS, which is way better defined with the other living coral we surveyed, with a microatoll shape. We made a mistake in including its elevation in the plot and we corrected this in the new version.

Line 348-349: "The average vertical uncertainty on a single concentric ring measurement is of +/-3 cm".

This statement is vague. Is this referring to the precision of the levelling equipment? Or the natural variability in HLG across a given concentric ring, derived by multiple elevation measurements of the same ring?

This statement refers to the natural variability in HLG we estimate from measurements over a single ring. The elevation of each concentric ring we measured was averaged, and the standard deviation around this average, which can be assimilated to the natural variability in HLG, is of ± 3 cm in average. We specified this.

Lines 351-352: "Other errors related to erosion or compaction of the site are complex but small ..."

The coral slabs suggest significant erosion, as evident from the truncation of near-vertical growth bands.

We changed our formulation.

While we recognize that erosion is visible in some coral slabs—particularly through the truncation of near-vertical growth bands—other portions of the same slabs retain well-preserved surfaces and display oblique growth axes near the top, suggesting that erosion was not severe (e.g. NAG6_18Q and NAG_B slabs that shows lower part preserved from erosion, growth axis in NAG3_21E). Given the relative nature of our analysis at the site scale, we assume that all coral samples experienced similar

degrees of post-mortem erosion. Therefore, we consider erosion a secondary effect that minimally impacts the elevation differences central to our interpretation.

Lines 353-354: Were the Yamaguchi et al dates calculated using the same half-lives as the dates in this study?

The half-lives used to calculate Yamaguchi (2016) coral ages (Sasaki et al., 2006) were ones in Cheng et al., 2000 (Chemical Geology). In our coral age calculation, new half-lives of ^{230}Th and ^{234}U in Cheng et al. (2013) were used. Only 0.18% offset on ages between two sets of half-lives. For example, for corals with ages of 2-4 ka, the difference of calculated ages between two half-life sets are only $\pm 3-7$ yrs, which are insignificant comparing to the dating errors of $\pm 11-330$ yrs in our coral ages. For ages of 15-30 ka in Sasaki et al., 2006, the offsets are 27-53 yrs between two half-life sets, which are smaller than $\pm 520-1840$ yrs in Sasaki et al (2006). Therefore, no significant difference for coral age calculations using different half-lives.

The other dates calculated by Yamaguchi (2016), including samples Yama_1 and Yama_2 that are included in our study, are estimated from radiocarbon method.

Line 363: Some coral slabs were sampled along the radius of the coral, not diameter. We revised this sentence.

Lines 371-372: "... to distinguish and discard from our analysis new colonies that commonly develop on top of dead corals."

Colonies B and C of NAG_B look like they belong to an independent coral colony thats distinct from colony A.

We agreed with the reviewer on this point. We revised our sentence by changing "colonies" to "overgrowths".

Line 389:

Is the uncertainty in the sampling of coral chunks (described here to approximate the most external growth band "by several years") accounted for in the uncertainty of the coral ages during subsequent interpretation?

The dated coral chunks, that are decimetric, represent ~ 10 years of growth when considering a growth rate of 1 cm/yr. This temporal averaging is not accounted for in the reported age uncertainty, as it is minor compared to the U/Th age uncertainty of ± 100 years in average.

Line 408:

We recommend shifting this part describing the "die downs" together with the part that introduces the HLS, as they are related concepts.

We modified the text accordingly.

Line 417:

Clarify that the +/- 1 cm relative uncertainty in the HLS curve of a single microatoll relates to the uncertainty in the reorientation of the X-Ray and CT scan images (described in lines 377-378). What about band-counting uncertainty, associated with uncertain interpretation in the coral growth bands when trying to reconcile the annual bands with the U/Th dates?

Done.

To account for growth band counting uncertainty in our age assignment for coral slabs

analysis, we used numerical estimation from Weil-Accardo (2014). From numerical modelling, the author estimated a growth band counting uncertainty ε of

$$\varepsilon = 0.03*n+0.56,$$

n being the number of growth band counted, starting from the most external one. For each coral, we estimated the age of the most external growth band from dated growth band with U/Th methods, and from the number of growth band and associated counting uncertainty between dated growth band and the most external one.

Lines 418-420: "... we observe an average range of 15 cm between measurements, leading to an estimated uncertainty of +/- 7 cm."

Does this mean that coeval diedowns were observed at different elevations in different corals, therefore deriving an uncertainty of +/- 7cm? If so, is this implying the natural variability in the HLS is +/- 7 cm?

Indeed, the ± 7 cm uncertainty reflects the difference in elevation between coeval RSL records across different corals, mostly NAG6_21C and NAG6_18Q. We agree with the reviewer that this uncertainty can be considered as the natural HLS variability, in agreement with other results from other study sites in the Ryukyus (Debaecker et al., 2023; Weil-Accardo et al., 2020). We use this value for the HLS variability across all Nagura site. We added a detailed explanation in our uncertainty estimation in the Method section.

Line 426: "... permit to make..."

A subject is missing between "permit" and "make"

Done.

Lines 427-428:

It is not clear how the 21 cm is derived.

We re-estimated our uncertainty on RSL drops. We added them in Table S2b and explained our calculation in the Method part.

Supplementary line 95:

How was mean sea level derived?

We were referring to the WGS84 ellipsoid and not any mean sea-level. We corrected that in the text.

Supplementary Table S2b:

Event P – is there a typo for the age of the event (442 BC not 44 BC)?

Indeed, there was a typo that we corrected.

Supplementary Tabel S2b:

what is the difference between the "Age" column and the "All-data age average" column? How were both derived?

The "Age" column indicates the U/Th (or ^{14}C) age for each coral from which we inferred an event in our RSL reconstruction. The "All-data age average" column indicates the age estimated for a unique event. This age is the weighted average of the different ages measured for a single event, when applicable.

U/Th ages:

- Were the samples pre-screened for secondary calcite?

Yes, they were. We checked that all of them had a percent of at least 95% of Aragonite to proceed to U/Th dating. We refined it in the method part.

• *NAGM_A1 and NAG6_18N have initial d_{234U} values that deviate by more than 5 ‰ from modern seawater (145 ‰), suggesting possible open system behaviour. Were these used in the analyses?*

We agreed with this reviewer. Apparently the two samples are not pristine and altered. The two ages cannot be used. NAGM_A1 was not used, and NAG6_18N death was assimilated with previous event I, which we do not interpret anymore. Consequently, letters assimilated to events changed compared to the previous version of this manuscript.

• *Samples NAG321E_A1, NAG618H_A2, NAG3_18C, NAG3_18D, NAG3_18B, NAG5_21A have anomalously low $^{230}Th/^{232}Th$ ratio, suggesting presence of non-negligible amounts of initial (non-radiogenic) detrital Th. How sensitive are the ages to the atomic $^{230}Th/^{232}Th$ ratios used for the age corrections (footnote d)?*

Shen et al. (2008) carefully evaluated the non-radiogenic ^{230}Th using corals from diverse settings in different oceans. We followed Shen et al. (2008) and used an estimated $^{230}Th/^{232}Th$ atomic ratio of $4 (\pm 2) \times 10^{-6}$ to correct for the non-radiogenic ^{230}Th . The age uncertainty significantly increases from ± 45 yr for measured age to ± 240 yr for corrected age for sample NAG321E-A1-D with measured $^{230}Th/^{232}Th$ atomic ratio of $22.75 \pm 0.39 \times 10^{-6}$. For sample NAG3_18C with measured $^{230}Th/^{232}Th$ atomic ratio of $30.42 \pm 0.81 \times 10^{-6}$, the age uncertainty increases from ± 77 yr for measured age to ± 210 yr for corrected age. The age uncertainties for all corrected ages are given in Table 1.

Figure comments

Figure 1

• *What do “low frequency” and “very low frequency” earthquakes mean?*

Low Frequency Earthquakes (LFE), Very Low Frequency Earthquakes (VLFE) as well as Slow-slip Events (SSE) are types of slow earthquakes. They can be defined by their duration, which is of several Hertz for LFE (they can also be assimilated to low frequency tremors), tens to several tens of seconds for VLFE, and several days to several years for SSE (Obara and Kato, 2016; Nakamura, 2017).

As the distinction between those several types of slow earthquakes is not used to interpret our results, we chose to simplify our figure by referring to those events as “aseismic events”.

• *Difficult to identify/ distinguish the “Graben” and “Okinawa Trough” colours*

The existence of grabens within the Okinawa Trough is not mentioned in our study and has no impact on our observations and results. Therefore, in an effort to simplify our message we chose not to show them in Figure 1 anymore.

Figure 2

• *Can the coral locations in panel b be drawn shown also in panel a?*

We represented them in panel a too.

- *It was very difficult to notice Yama_1 and Yama_2 in panel a as they are blue points on a blue background*

We changed their color to pink.

- *Is there a clearer image of the reef in panel b?*

Unfortunately, this is the best image we could obtain from drone imagery. Drone images in Figure 2 may have been downgraded from PDF conversion. Drone images, where microatolls are observable, will be made available in the data repository.

- *Panel c: what is the datum for the elevation measurements? Specify in the caption or y axis title.*

We used GNSS-RTK survey of 2021 as a reference frame, so the measurements correspond to elevation above the WGS84 ellipsoid.

- *Panel c: Do the “Fossil coral” symbols refer to the “undated corals”? If so, how were they placed along the x axis?*

Yellow points are fossil corals that could not be dated. Dated and sliced corals are represented with orange and red symbols, respectively.

Panel c is a plot of the coral’s elevation along a line perpendicular to the coast. The corals age does not impact our plot, as it will in Figures 3 and 4.

- *Panel c: we recommend colouring the error bars according to the symbol colour, so it is easier to read which error bars belong to which samples*

We change the color of the error bar.

- *Panel c: it is difficult to distinguish the red and orange symbols*

We changed the tone of the orange color to a more vivid one, and we change the red square symbol to a star symbol.

- *Panel c: why is the horizontal error bar for one of the living corals dashed?*

This living coral corresponds to what was called a “ball-shaped” microatoll. It is a hemispherical coral head which has never reached its HLS, and therefore, it does not add any meaningful information to this study. We chose not to represent it anymore.

- *Panel c: how was the tidal range measured? It seems odd that the living coral microatolls are out of the tidal range. It may also be useful to show the lowest astronomical tide as an estimate for extreme low water level at this site – which is what controls the microatoll diedowns.*

Points of tidal measurements shown in Fig. 2 showed only tide level when we measured it in 2018. Upon re-evaluating this figure, we concluded that including this dataset did not meaningfully contribute to the interpretation. Investigating the tide level in microatolls study is crucial when trying to reconcile HLS to the absolute sea level. This is beyond the scope of this paper here, as in our reconstruction of the RSL with fossil *Porites* microatolls we assume that the relationship between the coral HLS and the sea level is the same for each coral, no matter what this relationship is exactly. This approach is well established on a study site when using a single proxy (here, *Porites* microatolls), and when its modern counterpart is also documented (Meltzner and Woodroffe, 2015; Majewski et al., 2018).

Figure 3:

- *The caption describes this as a relative sea level plot, but the y axes are labelled as “Altitude (m)”. Please clarify if this is indeed showing the absolute elevation of the corals, or relative sea level.*

In Figure 3, we use the absolute elevation above the WGS 84 ellipsoid. We specified this in the y label.

- For NAG_B, which radius is shown in panel b, given that the full diameter was slabbed?

We chose to only show the left radius, which we revised following your comment, for the sake of readability of the plot as both radii show very similar RSL history.

- *Uplift events estimated from the eventual death of coral microatolls should be minimum uplift estimates. We recommend annotating an arrow and indicating “> XXX cm” to illustrate that the uplift (or RSL fall) could be greater than the vertical red line drawn.*

We modified the Figure 3 accordingly.

Figure 4:

As with figure 3, please clarify if the y axis is showing the absolute elevations of the corals or relative sea level.

We adjusted the absolute elevations above the WGS 84 ellipsoid to that of 2021 HLS measured from the external ring of the living microatoll we document. Y axis in Figure 4 therefore refers to relative sea-level, to be clearer, we specified “Elevation (m) above 2021 HLS”.

Figure 5:

The ellipses for the deep and main megathrust in panel a do not match the cross section in panel b. In the bottom panel of b, fault 2 (deep megathrust) ends at the northern end of Ishigaki (middle panel of b), but in panel a the blue ellipse does not extend to the northern end of Ishigaki island. The bottom panel of b also shows the blue ellipse extending further beyond the main megathrust, but the northern limit of both ellipses in panel a are aligned.

We corrected our figure accordingly.

Figure S1b – S1h:

There seems to be a mislabelling of the screw positions. Lines 363-365 describe that elevations were surveyed before the slabs were extracted, so it seems unlikely that the screws would be along the yellow line, and should instead be on the coral surface.

Indeed, they are. Yellow dots correspond to holes drilled in the slab below screws located at the coral surface in order to retrieve the line of horizontality following the guidelines from Meltzner and Woodroffe (2015). We modified our label, as well as panel in Figure S2a, to refer to line of horizontality.

Figure S2a:

Please indicate scales for each of the coral slabs.

Done.

References:

Cheng, H., Edwards, R. L., Hoff, J., Gallup, C., Richards, D., & Asmerom, Y. (2000). The half-lives of uranium-234 and thorium-230. *Chemical Geology*, 169(1-2), 17–33.

Cheng, H., Edwards, R. L., Shen, C. C., Polyak, V. J., Asmerom, Y., Woodhead, J., et al. (2013). Improvements in ²³⁰Th dating, ²³⁰Th and ²³⁴U half-life values, and U–Th isotopic measurements by

multi-collector inductively coupled plasma mass spectrometry. *Earth and Planetary Science Letters*, 371, 82–91.

Debaecker, S., Feuillet, N., Satake, K., Sowa, K., Yamada, M., Watanabe, A., ... & Shen, C. C. (2023). Recent relative sea-level changes recorded by coral microatolls in Southern Ryukyus Islands, Japan: implication for the seismic cycle of the megathrust. *Geochemistry, Geophysics, Geosystems*, 24(6), e2022GC010587.

Fujimoto 1997:

藤本 潔. (1997). 太平洋島嶼域におけるマングローブ林の立地形成と海水準変動. *Tropics*, 6(3), 203-213.

Fujimoto, K. and Ohnuki, Y. (1995): Developmental processes of mangrove habitat related to relative sea-level changes at the mouth of the Urauchi River, Iriomote Island, Southwestern Japan. *Quarterly Journal of Geography* 47: 1-12.

Fujimoto et al., 1995:

藤本 潔・山内秀夫・目崎茂和・長谷川均・前門 晃, 1995. 石垣島名蔵川低地とマングローブ 林の形成過程. 8本地 留学会予結集 47 : 384~385.

Garas, K. L., Watanabe, T., & Yamazaki, A. (2023). Hydroclimate seasonality from paired coral Sr/Ca and $\delta^{18}\text{O}$ records of Kikai Island, Southern Japan: Evidence of East Asian monsoon during mid-to late Holocene. *Quaternary Science Reviews*, 301, 107926.

Goldfinger, C., Nelson, C. H., Morey, A. E., Johnson, J. E., Patton, J. R., Karabanov, E. B., ... & Vallier, T. (2012). *Turbidite event history—Methods and implications for Holocene paleoseismicity of the Cascadia subduction zone* (No. 1661-F). US Geological Survey.

Goto, K., Kawana, T., & Imamura, F. (2010). Historical and geological evidence of boulders deposited by tsunamis, southern Ryukyu Islands, Japan. *Earth-Science Reviews*, 102(1-2), 77-99.

Hisamatsu, A., Goto, K., & Imamura, F. (2014). Local paleo-tsunami size evaluation using numerical modeling for boulder transport at Ishigaki Island, Japan. *Episodes*, 37(4), 265-276.

Hongo, C., & Kayanne, H. (2009). Holocene coral reef development under windward and leeward locations at Ishigaki Island, Ryukyu Islands, Japan. *Sedimentary Geology*, 214(1-4), 62-73.

Kawana, T. (1987). Holocene crustal movement in and around the Sekisei Lagoon in Okinawa Prefecture, Japan. *Earth Monthly*, 9, 129-34.

Kawana, T. (1989). Quaternary crustal movement in the Ryukyu Islands. *Earth Monthly*, 11, 618-30.

Majewski, J. M., Switzer, A. D., Meltzner, A. J., Parham, P. R., Horton, B. P., Bradley, S. L., ... & Mujahid, A. (2018). Holocene relative sea-level records from coral microatolls in Western Borneo, South China Sea. *The Holocene*, 28(9), 1431-1442.

Meltzner, A. J., Sieh, K., Chiang, H. W., Shen, C. C., Suwargadi, B. W., Natawidjaja, D. H., ... & Galetzka, J. (2010). Coral evidence for earthquake recurrence and an AD 1390–1455 cluster at the south end of the 2004 Aceh–Andaman rupture. *Journal of Geophysical Research: Solid Earth*, 115(B10).

Meltzner, A. J., & Woodroffe, C. D. (2015). Coral microatolls. *Handbook of sea-level research*, 125-145.

Meltzner, A. J., Sieh, K., Chiang, H. W., Wu, C. C., Tsang, L. L., Shen, C. C., ... & Briggs, R. W. (2015). Time-varying interseismic strain rates and similar seismic ruptures on the Nias–Simeulue patch of the Sunda megathrust. *Quaternary Science Reviews*, 122, 258-281.

Miyagi, T., Ajiki, K., & Fujimoto, K. (2003). Mangrove habitat dynamics and human relation. *Kokon-Shoin Ltd., Tokyo, (193 pp)(in Japanese)*.

Nakamura, M. (2017). Distribution of low-frequency earthquakes accompanying the very low frequency earthquakes along the Ryukyu Trench. *Earth, Planets and Space*, 69, 1-17.

Obara, K., & Kato, A. (2016). Connecting slow earthquakes to huge earthquakes. *Science*, 353(6296), 253-257.

- Ota, Y., & Hori, N. (1980). Late Quaternary tectonic movement of the Ryukyu islands, Japan. *The Quaternary Research (Daiyonki-Kenkyu)*, 18(4), 221-240.
- Peltier, W. R., Argus, D. F., & Drummond, R. (2015). Space geodesy constrains ice age terminal deglaciation: The global ICE-6G_C (VM5a) model. *Journal of Geophysical Research: Solid Earth*, 120(1), 450-487.
- Philibosian, B., Feuillet, N., Weil-Accardo, J., Jacques, E., Guihou, A., Mériaux, A. S., ... & Deroussi, S. (2022). 20th-century strain accumulation on the Lesser Antilles megathrust based on coral microatolls. *Earth and Planetary Science Letters*, 579, 117343.
- Sasaki, K., Omura, A., Miwa, T., Tsuji, Y., Matsuda, H., Nakamori, T., ... & Nakagawa, H. (2006). ²³⁰Th/²³⁴U and ¹⁴C dating of a lowstand coral reef beneath the insular shelf off Irabu Island, Ryukyus, southwestern Japan. *Island Arc*, 15(4), 455-467.
- Shen, C. C., Li, K. S., Sieh, K., Natawidjaja, D., Cheng, H., & Wang, X. (2008). Variation of initial ²³⁰Th/²³²Th and limits of high precision U–Th dating of shallow-water corals. *Geochimica et Cosmochimica Acta*, 72(17), 4201–4223.
- Sowa, K., Watanabe, T., Kan, H., & Yamano, H. (2014). Influence of land development on Holocene Porites coral calcification at Nagura bay, Ishigaki island, Japan. *PLoS One*, 9(2), e88790.
- Suzuki, A., Yokoyama, Y., Kan, H., Minoshima, K., Matsuzaki, H., Hamanaka, N., & Kawahata, H. (2008). Identification of 1771 Meiwa Tsunami deposits using a combination of radiocarbon dating and oxygen isotope microprofiling of emerged massive Porites boulders. *Quaternary Geochronology*, 3(3), 226-234.
- Tsuji, Y. (1993). Tide influenced high energy environments and rhodolith-associated carbonate deposition on the outer shelf and slope off the Miyako Islands, southern Ryukyu Island Arc, Japan. *Marine Geology*, 113(3-4), 255-271.
- Weil-Accardo, J., Feuillet, N., Satake, K., Goto, T., Goto, K., Harada, T., ... & Shen, C. C. (2020). Relative sea-level changes over the past centuries in the central ryukyu arc Inferred from coral microatolls. *Journal of Geophysical Research: Solid Earth*, 125(2), e2019JB018466.
- Yamaguchi, T. (2016). A review of coral studies of the Ryukyu Island Arc to reconstruct its long-term landscape history. *Coral Reef Science: Strategy for Ecosystem Symbiosis and Coexistence with Humans under Multiple Stresses*, 55-63.
- Yamano, H., Inoue, T., Adachi, H., Tsukaya, K., Adachi, R., & Baba, S. (2019). Holocene sea-level change and evolution of a mixed coral reef and mangrove system at Iriomote Island, southwest Japan. *Estuarine, Coastal and Shelf Science*, 220, 166-175.
- Yamauchi et al., 1995:
山内秀夫・長谷川均・目崎茂和・前門 晃, 1995. サンゴ礁 潟の環境変化と保全. 日本自然保護協会(編) 口・ア介 ーラ・ファ 第 府助成成果報告 17.
- Yokoyama, Y., Maeda, Y., Okuno, J. I., Miyairi, Y., & Kosuge, T. (2016). Holocene Antarctic melting and lithospheric uplift history of the southern Okinawa trough inferred from mid-to late-Holocene sea level in Iriomote Island, Ryukyu, Japan. *Quaternary International*, 397, 342-348.
- Zachariasen, J. A. (1998). *Paleoseismology and paleogeodesy of the Sumatran subduction zone: A study of vertical deformation using coral microatolls* (Doctoral dissertation, California Institute of Technology).
- Zachariasen, J., Sieh, K., Taylor, F. W., & Hantoro, W. S. (2000). Modern vertical deformation above the Sumatran subduction zone: Paleogeodetic insights from coral microatolls. *Bulletin of the Seismological Society of America*, 90(4), 897-913.

Reviewer #2 (Remarks to the Author):

We thank the reviewer 2 for the careful reading of this work.

Reviewer #3 (Remarks to the Author):

This is an exciting paper that the authors present a comprehensive study of the history of megathrust earthquakes over the past five thousand years in the southern Ryukyu Trench using coral microatolls. The manuscript is well-organized and clearly written. I only have few minor comments and support publication.

We thank this reviewer for their helpful and constructive comments.

Lines 193-195: The inferred supercycles in the southern Ryukyu Trench are based on data from a single paleoseismic site. Are there any published studies using sediment cores from the seafloor, marine terraces, or tsunami boulders dated in Okinawa island that support the existence of these supercycles found in southern Ryukyu Trench?"?
Yes, there are several studies that investigate traces of past tsunamis or earthquakes in the Ryukyu arc.

Past tsunamis were mostly identified through the finding of tsunami boulders and tsunami deposits in the stratigraphy. The different studies are consistent between them, and we use their results in Figure 4 to compare interpreted megathrust earthquakes with tsunamis events.

Past earthquakes are more difficult to identify. There are Holocene marine terraces in Kikai island in the Central Ryukyus, but we believe that such earthquakes are too far to be associated with our study zone, and that their origin is strongly influenced by the subduction of the Amami plateau at the trench (Webster et al., 1998).

There is a consensus that an earthquake occurred in 2200 B.P. (which could correspond, in our study, to event S, now referred as event R in the revised version). Its coseismic motion has been evidenced by uplift of marine notches (Kawana, 1987; 1989).

Last, and closer to our study site, a study on turbidites sampled in the Hateruma basin, south of Hateruma and close to the trench, evidences several events affecting the sedimentation of the basin with a recurrence time of ~1000 years over 10 ka (Ujiie et al., 1997; Babonneau et al., 2019). This would be consistent with our interpretation of the past RSL in Nagura site, but the latter study has not been published yet and we are not certain of the origin of these events yet.

Those supporting evidences are mentioned in the manuscript, lines 272-274.

Lines 232-238: The Okinawa Trough is rifting southward at a rate of approximately 5 cm/year in this region (e.g. Nishimura et al., PEPI, 2004). As a result, the convergence

rate between the Philippine Sea plate and the southern Ryukyu forearc exceeds 8 cm/year (e.g. Chen et. al. <https://doi.org/10.1029/2022GL098218>). Authors need to refine interseismic modeling accordingly.

We modified the result of our elastic modeling accordingly.

References:

Babonneau, N., Guérin, C., Ratzov, G., Lallemand, S., Condomines, M., Su, C. C., ... & Mercier De Lepinay, B. F. (2019, December). Extreme events recorded in sediment cores of the Hateruma forearc basin, southern Ryukyus (MD214 EAGER Oceanographic Cruise). In AGU Fall Meeting Abstracts (Vol. 2019, pp. OS51C-1501).

Kawana, T. (1987). Holocene crustal movement in and around the Sekisei Lagoon in Okinawa Prefecture, Japan. *Earth Monthly*, 9, 129-34.

Kawana, T. (1989). Quaternary crustal movement in the Ryukyu Islands. *Earth Monthly*, 11, 618-30.

Ujiié, H., Nakamura, T., & Miyamoto, Y. (1998). Holocene turbidite cores from the southern Ryukyu Trench slope: suggestions of periodic earthquakes. *Oceanographic Literature Review*, 7(45), 1148

Webster, J. M., Davies, P. J., & Konishi, K. (1998). Model of fringing reef development in response to progressive sea level fall over the last 7000 years—(Kikai-jima, Ryukyu Islands, Japan). *Coral reefs*, 17, 289-308.

Reviewer #4 (Remarks to the Author):

Major points

Robustness and validity of the data

First of all I would like to note that I am no expert in paleosealevel changes and coral measurements. However, as an informed earthquake scientists with some expertise, I did my best to decipher the data presented. I found it hard to understand the presented data and follow several aspects of assumptions seemed very severe. Your descriptions makes some jumps in text and only to a limited degree refer to Figures, such that I needed to spend quite some time to follow and interpret what you are trying to show (e.g., sentences above Fig 3; are 3a and 3b a part of a longer record you that you only show for the first time in Fig 4? Then it makes more sense to first show the complete record (4) and then zoom into a few parts)).

We rewrote parts of the text and increased our references to the figures. We hope that this new version of our study is clearer now.

Our study focuses on the occurrence of several sudden drops in the RSL, some of which were interpreted from the corals slab analysis summarized in Figure 3, and other were interpreted from the RSL reconstruction and model in Figure 4. Figure 4 is a key figure that present our main result, but also interpretation of the RSL as we introduce the notion of the seismic supercycles. For that reason, we chose to start from zoom of few parts of the RSL record to an overall look of the RSL reconstruction and interpretation afterward.

Above figure 3 you write about nine sudden Relative Sea Level (RSL) drops with values ranging from 6 to 50 cm. If I would analyse Fig 3a without the interpretation lines you draw over the data I would probably not identify at least 3 events you do. For example:

- 2482: Where is the minimal data point of your event? I only see the red and black line you draw

We inferred this event C from the death of the microatoll NAG7_18A dated at 2482 ± 41 years BC. Its amplitude depends on the rate of the constant interseismic RSL rate we apply between events, 2.0 ± 0.6 mm/yr. The minimal data point of this event could be related to NAG7_18A thickness (>30 cm, Table S1a), but in our RSL reconstruction and event amplitude assessment, we do not represent it.

- 2357: Again no minimum point are any data at the time you draw

Event F dated at 2357 ± 8 years B.C. correspond to the death of coral NAG6_18Q. As for event C, we infer that the death of this coral could be related to a large drop of the RSL. In this case, we can have a more precise estimate of the amplitude of this drop as it would be at least 40cm, which is the external thickness of the NAG6_18Q coral.

- 2108: No minimum to connect to some other dots e.g., the blue ones seem unrelated to your interpretation. How do they fit?

The dated plateau (blue dots in Figures 3 and 4) has an age of 2128 ± 330 years B.C. As its age is closer to event J (now referred as event I in the revised version), we estimated that it is related to it and estimated the age of event from this plateau, as well as the RSL drop of ~ 11 cm observed in the NAG6_18H slice at 2137 ± 20 years B.C.

The calculated age of event J (now event I) is of 2137 ± 20 years B.C., which is explained by the large difference in age uncertainty between the dated plateau (330 years), and the drop in NAG6_18H (20 years). This result is consistent with the fact that given its large age uncertainty, this dated plateau could be associated with already identified events D to M (now events D to L), and therefore could be considered not significant.

Additionally, I could identify at least half a dozen more drops that you do not identify. For example:

- Between 2208 and 2240

There is indeed an apparent drop in the NAG6_18H coral record. However, this drop is not significant as it is related to bioerosion (there was mangrove roots there, as we stated in the Supplementary Materials).

Photograph of NAG6_18H showing the little mangrove tree causing the bioerosion documented in the slab (Figure S2a).

- Between I and J

There are variations of the RSL in the NAG6_18H record between events 2208 and 2137. However, these variations are small (less than 2cm between two consecutive years), and they may be related to erosion and/or climatic changes.

That being said, criteria to distinguish drops related to tectonic motion, to drops related to climatic events were documented by Taylor et al. (1987), and commonly used in microatolls studies ever since:

- Spatial extent: a local event is more likely of tectonic origin rather than a climatic event, which tend to be regional. Here we cannot use this criterion.
- Quick regrowth of the coral after the drop: a regrowth of the corals during the years directly following the drop would rather correspond to a climatic event than to a tectonic one, as vertical tectonic motion is expected to permanently change the RSL.
- Amplitude of the drop: an amplitude of at least ~10 cm is more likely related to tectonic. At the opposite, a small of amplitude (for example, 4 cm) does not permit to decipher the origin of the event.
- The occurrence of coeval events, either in earthquake catalog, other microatolls record or geological tools.

Here in NAG6_18H record, only the event at 2137 ± 20 years B.C. is documented as a tectonic motion: it has an important amplitude (~11 cm) and it induced a permanent

change the RSL. Furthermore, although the coral shows signs of unpreserved HLS/HLG growth bands before and after the drop, the RSL drop corresponds to a buried stepdown in elevation that was observed around the coral, which is very unlikely related to erosion.

- Also where from come the black lines beneath H and before G?

Those black lines refer to periods of relative submergence. We actually chose not to represent those before event G (as well as other relative submergence periods in NAG3_21E in panel b), as they highlight a drop that cannot be properly interpreted regarding its small amplitude (~4 cm).

Other relative submergence periods are shown as we infer that they could indicate the interseismic RSL variations. We modified Figure 3 to enhance its readability.

- Also in Fig. 4 I wonder what data support the drops and black lines that you draw in Fig 4a.

In Figure 4, drops are inferred from:

- Drops of the RSL of at least 10cm identified in the HLS curve from coral slab analysis.
- Death of sliced microatolls or dated plateau. Each of the dark blue square in Figure 4 represent a plateau that was dated from the external part of a microatoll that has thickness between 20 to 70 cm. Microatolls died from a tectonic event (i.e. uplift leading to a drop of the RSL) which is commonly proposed with fossil microatolls (Philibosian et al., 2022).

We applied a confidence index from 1 to 5 to indicate the robustness of our interpretation of the RSL drops (see Method section and Table S2b). Drops inferred from slab analysis have a confidence index of 5. The confidence index we apply to those events is of maximum 4 out of 5, because we cannot fully discard the hypothesis of a past climatic mass bleaching. Reconstruction of past climate is complex, and the studies in the region tend to show no anomalies that could account for the sudden death of microatolls. We apply a confidence index of 3 for events interpreted from the death of one fossil microatoll. Events related to fossil corals Yama_1 and Yama_2 observed by Yamaguchi (2016) have a confidence index of 2. Last event U has a confidence index of 1, as it corresponds to the hypothesis that plateau 1 was formed during the Meiwa event.

We downgraded by one point the confidence index of events with large uncertainties on their estimated amplitude.

- And how do you determine the values of the minima in your sawtooth pattern in Fig. 4a?

Figure 4a is a model of possible seismic supercycles we propose from the observation and simple hypothesis on the vertical distribution of dated corals along Nagura site, as well as the analysis of sliced microatolls. We applied a constant interseismic RSL rate of 2.0 ± 0.6 mm/yr from that interpreted in NAG3_18H slab analysis which is a reasonable hypothesis although we know that the interseismic rate can change through time. From a given dated coral, we applied this same rate constantly until the previous dated fossil coral. The difference in RSL at the older coral correspond to the addition of the difference in elevation between the two corals and the amount of RSL interseismic calculated along the time span between the two fossil corals. Our estimation understands uncertainties added in quadrature related to age, elevation and interseismic RSL rate.

We added a detailed description of our estimation in the Method section.

In general for many lines in that key graph it is not clear where they come from. Please explain what the lines are and where they come from.

In Figure 4, black lines show the RSL reconstruction, which is composed by interseismic period with RSL rate of 2.0 ± 0.6 mm/yr, and black vertical lines that represent drops of the RSL, e.g. inferred seismic events. Those events are indicated with dashed red lines with their reference letter at the bottom of the graph. Those red lines are the same in the panel b. Dashed black lines show possible other RSL reconstruction when RSL reconstruction are inferred. The dark blue line represents the actual HLS, measured in 2021. Large grey line represents the long-term RSL trend over 5 ka. This trend is estimated from slab records except that of NAG_B (which higher elevation could be accounted for other processes than RSL variations), and from dated plateau except the modern one, to approximate the current state of the seismic supercycle in light of our RSL reconstruction. Finally, black arrows show the extent in time of our inferred seismic supercycle, with dashed part representing the quiescence part and the solid part representing the failure sequence.

Even when you draw the interpretations for me I have a really hard time to make any sense out of those interpretations. Even after starring at it for more than 15 minutes I can not do it. It also makes me wonder whether you should draw so many lines on top of it, as the data themselves are not clear anymore.

In all, I do not find the presence of those nine events convincing at all.

We have simplified our figures and clarified our text. We hope this new version of our manuscript will be more reachable.

The same holds for figure 3b. I could interpret many events after $t=518$ and after $t=256$ that look the same as the lines that you do show in red. I also see 7 interpreted vertical lines that I think you want to show as vertical drops, but the text here refers to four sudden drops. All these points make me sincerely doubt your interpretation and the robustness of your story.

We thank the reviewer for this remark. We tried to simplify the figure by showing only by the main RSL drops and corrected the text accordingly. Other smaller drops that can be seen often in the slice's records were not shown because they have a smaller amplitude (<10 cm), which could indicate either a tectonic motion or a climatic event (Taylor et al., 1987). In addition, some of those drops are not significant as analysis of the slab and its original colony show that those drops correspond to erosion and/or climatic changes.

That questionability about the robustness of your data and interpretation is strengthened when reading that the uncertainty is ± 6 cm (caption Fig 2). Several drops that are in the data are around that 6 cm, so I do not think you can interpret that.

The calculated uncertainty we apply to plateau elevation as markers of past RSL are mostly related to (1) RTK-GNSS precision of ± 5 cm and (2) estimated HLS variability along the study site of ± 7 cm. These uncertainties do not apply to the slab analysis, which have a centimetric precision. Drops documented by those slices are therefore

very well constrained, whereas they are the smaller ones (Table S2b).

Other drops have important uncertainties, related to those previously mentioned, as well as uncertainties related to their age and the interseismic rate we apply between them.

To better highlight the robustness on the occurrence of sea-level drops recorded by the corals, we developed a scale of confidence index. For the drops with uncertainties larger than their amplitude, the confidence index is downgraded by 1.

As an example, for event N (now referred as event M) in 627 ± 212 B.C. in Table S2b, we apply an initial confidence index of 4 as this event corresponds to the death of NAG3_18C associated with 7 corals at similar elevation (plateau 16, Table S1c). As the amplitude of the drop was estimated at 15 ± 43 cm, we downgraded its confidence index to 3.

We precised this approach in the Method section.

Overinterpretation of a super cycle

This data interpretation is then used to suggest strong seismic hazard for the next centuries (end of the abstract / conclusion section). Based on the data, which could also represent other earthquakes, this is suggestive.

In Fig.4a you draw a large drop / event around 1300 BC.

- (a) *Based on what data is that drop / event justified? I do not see that in the figure.*

That drop is inferred from the death of part the plateau 12, in which a fossil concentric stepdown in a coral colony (GIANT_21) was dated at 1324 ± 23 B.C. Plateau 12 is composed of several colonies at similar elevation dated at two different ages (1324 ± 23 and 367 ± 8 B.C.) and here we cannot make the hypothesis that all undated fossil corals of the plateau died at the same time. We then downgraded the confidence index by one point.

In a second time, we estimated a very large drop of 166 ± 60 cm for that event, which is could be related to the lack of data and/or the RSL interseismic rate we apply.

- (b) *That very large event in the middle of your supercycle #2 does not really fit the supercycle thinking and process. Since supercycle #2 is your only complete one in your story line (Supercycle #1 you have the end and of #3 the potential approach of the end of another supercycle), I do not see one clear and complete supercycle. Hence I wonder how reliable your claim for recurrent supercycles is, and how likely it is.*

We agreed with the reviewer on this point.

The event at 1300 BC is in the middle of our interpreted quiescence period of supercycle 2. It is indeed less continuous and thus obvious than reference supercycles from Sieh et al. (2008). In the southern Ryukyus, periods of seismic quiescence are observed, during which earthquakes are infrequent yet distinctly interpreted. Here, quiescence periods differ from failure sequence by the frequency of interpreted seismic events.

We corrected our characterization of seismic supercycle here by only underlying the difference in events frequency.

Overall, following my comments above I do not think the "RSL signal is strikingly similar to what was observed in Sumatra by Sieh et al.". The true supercycle signal presented

here seems less evident.

The sentence has been modified to better align with the nature of the signal described, while maintaining the relevance of the comparison to Sumatra.

Potential M9.0 not justified

You suggest that a M9.0 earthquake can explain the coseismic uplift data by Kawana, but I do not agree with that interpretation:

(1) The calculation does not add up. Following your numbers I calculate a seismic moment M_0 of

$$M_0 = \text{rigidity} * \text{slip area} * \text{average slip} = 32e9 * 300e3 * 50e3 * 25 = 1.2e22 \text{ N m.}$$

Using empirical scaling relations that leads to a moment magnitude of

$$M_w = 2/3 \text{ LOG}_{10} (M_0) - 6.05 = 8.67$$

That is significantly smaller than a M9.0, as more than 10x less energy is released than what you suggest. A factor 10 is a large discrepancy.

To reach a M9.0 80 meters of slip would be required on average on the fault plane you use. So much slip has never been observed.

In fact, the width we used for the dislocation model is 150 km (Fault 1) and not 50 km to explain the Kawana (1987; 1989) coseismic deformations (red curve with a slip of 25 m) corresponding to a M9.0 earthquake.

(2) Also an average of 25 meters of slip over depths from 0 - 50 km is already a lot and is only rarely observed. I wonder if the 25 meter estimate is biased towards high values by the simplified dislocation modelling assumptions that you use. Tsunami modellers regularly observe that unrealistic values of elastic parameters need to be used to make seafloor displacements match slip estimates from seismological sources.

We never used this value of 25 m for the deeper portion of the slab (Fault 2), 25 m is for the entire megathrust 1 + 2 faults, the maximum slip for the deeper portion is 9 m.

We use a uniform slip for simplicity as our data are too sparse to resolve the slip of the shallower part of the megathrust. Slip on a restricted patch below the island could also explain the results. This is why we tried many forward models to explain our values. This results in large uncertainties on evaluation of the magnitude that range between 7.7 and 9.1 depending on the model used.

(3) In Figure 5b the top panel also shows that the 25 meter line is constrained by very little data, because all data are located very near to the hinge line.

We are not trying to constrain the 25-m model. We provided many forward models to explain our data (coseismic uplifts we inferred from microatolls) and the uncertainties are large on the magnitude (see results).

Moreover, coincidentally your data are only located right above the top of the deepest part of the deepest fault patch (segment 2 out of 2).

Yes, the data we have are just at this place, where we have topography and long-term deformation recorded by reef terraces. This is probably because this area is located just above a kink in the slab. We used the data we have and again uncertainties on magnitude are large because several models can explain the data.

Your simplified modelling required you to make gross approximations to make these estimates (see need major point). Those approximations in combination with the large extrapolations required lead to a large uncertainty on this 25m estimate and corresponding earthquake size.

We agreed that our modeling is simple, but we only have one observation site. We therefore modelled a single coseismic uplift datum. The subduction geometry is constrained in the SLAB 2.0 model (see method), which uses updated seismic refraction data and well-localized seismic events. We therefore modify only the slip of our dislocation. We are confident that this simple model is valid for modeling surface deformation measurement at a single point. More complex models involve more parameters, making the modeling procedure invalid. We estimate that if the data are perfect, we could use a ratio of 1 between the number of data and the number of parameters. However, there is always an element of uncertainty, so we need to increase this ratio as much as possible. For deformations, we should maintain a ratio of at least 2 to 3. Okada has 10 parameters. We are modifying two parameters (slip on two faults), with data that is under-resolved. We think that using more complex models would be not reasonable.

(4) By slightly moving the location of that second linear patch to the right (trenchward) a much smaller deep earthquake with 9 meters slip could also agree with all of your data (see Fig. 5b). Since the actual slab interface is curved and already not very well imaged and located, such a 10-15km shift is well within the uncertainties. Following your suggestion of the kink "likely being located between 25 to 35 km depth" that translates to a horizontal uncertainty of 24km (using an average dip of $(30+15)/2 = 22.5$ degrees). Hence twice as large as the shift required to fit all data with a deep event that slips 9m.

The subduction geometry is that of the SLAB2 model (see slab depth in panel a), and we modify only the slip. We have no reason to modify this geometry. The bend is located at the intersection between segments with different slopes, which is the only way to respect the SLAB2.0 geometry. It is always possible to adapt the model but we are using the updated available data for the geometry. The text was confusing, we modify this.

- How can you robustly separate where slip occurred?

We are not trying to separate the slip; we have generated a forward- model to explain our vertical displacement.

A shallow rupture between 0 and 33 km favors subsidence (see figure in the supplementary material). We need slip beneath the island (i.e. in the deeper portion of the megathrust) to explain the uplift.

We can not distinguish if only the deep segment ruptures, so we also modelled a slip on the entire megathrust between 0 and 60 km.

The segmentation simply depends on the geometry (the bend comes from the SLAB2.0 model).

- And what true evidence exists for the presence of a kink over a curved geometry?

The kink is a simplification of the SLAB2.0 geometry, which slope increases rapidly below a depth of 30 km. The kink is the intersection between planes with different slopes.

Simplified fully elastic modelling on a planar fault can not lead to accurate fault slip estimates

We believe that as the dip increases rapidly, it is possible to use models with a bend. We could have used many sub-segments to model a curve, but as the change in dip occurs in a very small area, we are confident that this would not have changed the result significantly.

Modelling of vertical motions is difficult and we hardly ever model vertical motions correctly. Not even at the shorter earthquake cycles time scales we observe (e.g., Govers et al., Reviews in Geophysics, 2018). This is partially because we do not understand the larger-scale geodynamic processes that cause them in combination with fault slip (e.g., Govers et al., Reviews in Geophysics, 2018). You follow a suggestion to balance our inadequate understanding and modelling simplifications by largely changing the dip of the fault (from 12.5 to 31 degrees, which is very significant!) over the most relevant depth intervals of 0-33km.

The change in slope is not due to our insufficient understanding and modeling simplifications. These slopes are from the SLAB2.0 model (see method) which is well constrained by seismicity, seismic refraction and seismic tomography in the region (see slab depth on the map in panel A). We simply used the available slab geometry. We added the depth of the slab in the maps for clarity.

These very large changes to compensate for largely simplified modelling assumptions (e.g., full elastic medium, back slip, planar fault) can in my opinion not lead to robust estimates of fault (back) slip rates.

Once again, we used simple models with few parameters to model the observations at a point. This is always the case in areas where uplift is determined with little data.

- *We know viscoelastic medium behaviour significantly influences vertical motions (e.g., Wang et al., Nature, 2012, Weiss et al., Science Advances, 2019).*

We agreed, but unfortunately our data set is too small for this (1 site). More data would be needed to distinguish these processes.

Furthermore, the main objective of this article is to model coseismic deformation and not interseismic deformation (including postseismic afterslip or viscoelastic relaxation). We agreed that there is uncertainty in modeling interseismic data and the locking depth is certainly not well estimated, but our aim was to show that the zone is partially coupled and can favor earthquakes.

An interesting study that could be carried out in the future would be to understand the long-term deformation recorded by corals, but again, our dataset is limited to a single site and therefore very limited to this objective.

- *As mentioned above, we know the interface is curved. The paper that you cite (Kanda & Simons, Tectonophysics, 2012) also explains that the hinge location and vertical motions are very sensitive to planar or curved faults. Since you have only a very narrow band of data very close to the hinge location, the usage of a planar versus curved fault is also said to have a very large impact on the location of the hinge location.*

The change in slope is very localized, the slab plunges rapidly beyond 30 km and thus

the kink approximation is valid.

Hence this would critically estimate your estimates, which required large extrapolation to that estimate the amount of slip (and hence earthquake size) somewhat. The impact of this assumption on the coseismic motions is also demonstrated by Ragon et al., GJI, 2018.

- We know geodynamic processes impact the vertical motions as well (e.g., Govers et al., *Reviews in Geophysics*, 2018).

- Additionally, the very distinct topography near the islands could play a role (e.g., Landers et al., *Tectonophysics*, 2020).

I know inverting for at least the first and/or second and/or third point requires state-of-the-art modelling that has become available since five years (see several works of Thea Ragon and Sylvain Barbot referred to below), but I think that level of robustness is justified for Nature Communications.

We agreed that there are uncertainties on our estimate but the results show at a first order that the rupture of a deep segment is required, they are large uncertainties magnitude (also because the models are in 2D) and we mentioned this in the legend (magnitude ranging between 7.7 and 9 depending on if the rupture continues in the shallower part of the megathrust).

Using more complex would be certainly very interesting but again our dataset is too limited to this goal.

Moderate points

- Novelty of the data

As written in the abstract geodetic measurements and slow slip events suggest coupled patches along the megathrust and there is evidence for great tsunamis. Hence the author provide another piece of the puzzle using a new and longer data set. However, it is not very clear from the text what the novelty of that data set is. The authors other paper in 2023 in GGG (below) have a similar introduction, but they then show similar data on a much shorter record from 1800 to 2020. Please better clarify in the introduction already that it is the duration of your record that makes the largest difference. You have that in the next paragraph, but then quantify the length of the other data sets in the first paragraph of page 2. For example for these two citations (i.e., 1 to 2 centuries rather than thousands of years)?

Weil-Accardo, J. et al. (2020). - last century

Debaecker, S. et al. (2023).: time range is difference; 1800-2020

We revised our introduction and specified the time span of the mentioned studies. We believe these changes make the contribution and originality of our dataset unambiguous.

Moreover, these texts in the abstract seem to contradict a sentence at the end of page 2, which states “ mainly aseismic “. It rather seems that observations are not in agreement and a new very long record is added to understand the earthquake potential and recurrence behaviour of this subduction segment.

That remark is correct, this is why we believe our study is important. In the abstract, it is specified that weak coupling is inferred from “geodetic measurements on the islands”, and recent microatolls studies and offshore geodetic data rather indicate

coupled patches. Our sentence at the end of page 2 was too general and we revised that the “mainly aseismic” was interpreted from inland geodesy studies.

- *Some terminology is not explained at a level suitable for the broad readership of Nature Communications. Examples include e.g., “total station” (p.3,4).*

We introduced an explanation of what is a total station at its first mention in the Methods section.

- *Some sentences are incredibly hard to follow for the reader. For example, see the top few at page 6 . The complete story in the “Seismic supercycles” section is not easy to follow and understand and has a lot of if’s present.*

We appreciated the reviewer’s comment and agreed that the clarity of the “Seismic supercycles” section is crucial. We have revised the beginning of this section to improve its readability and ensure that the narrative is more straightforward for a broad audience.

- *Moreover, a clear explanation of the vertical motions of the tectonics and the coral and what then the relative changes mean is missing.*

We included a description of vertical tectonic motions, coral response, and their interpretation in terms of relative sea-level changes in the Methods section. However, we agree that this part could benefit from additional clarification, and we have revised the relevant paragraph to make the physical mechanisms and their relation to the data more accessible to a broader readership (lines 349-352).

- *Additionally, an explanation of the physical mechanisms governing super cycles is missing. If you want to suggest they exist a brief explanation of why they can exist would be useful for the reader. Papers such as Herrendorfer et al., Nature Geoscience, 2015, explain how stress build up near the down dip limit can explain quiescence periods, whereas subsequent earthquakes can transfer the stresses loaded from below. That transfer loads the entire mega-thrust and prepares it for complete rupture in a super event that you anticipate.*

We appreciate this suggestion, and we have added a brief synthesis of these mechanisms to the revised manuscript for clarity.

- *P.6: Sudden coseismic uplifts of 210 cm are extremely large and much larger than the largest coseismic uplifts we have measured for e.g., Tohoku. This also contradicts other values in the paper, so is this reasonable?*

Could you also add mean, min, max values there, such that we can better interpret your values?

A coseismic uplift of 210 cm, now 262 cm (prior event S, now event R) is on the highest bound of what is observed during large earthquakes (i.e. 2.50 m for the 2010 Maule Earthquake, Farias et al., 2010). This estimate is based on our RSL reconstruction and it is possible that due to coral absence, we missed some evidence of uplifts in our survey. However, the absence of coral can also indicate that a coral reef needs time to recover after a major earthquake. That is why we have considered the possibility that undated plateaus could be younger than event S (now event R), and with a constant interseismic rate between events, the amount of uplift that occurred in event R would correspond to three small uplifts events instead of a bigger one. We used the same

hypothesis for event M (now event L, dashed lines in Figure 3.).

It is also possible that the interseismic rate varied through time (Meltzner et al., 2015), and includes post-seismic signal that we cannot constrain. The interseismic rate that we applied in our RSL reconstruction is well constrained by a well-preserved microatoll on a limited time scale. This may lead to a under or overestimation of the coseismic movement from curves of seismic cycle.

Dated plateaus in Figure 4 correspond to several fossil of microatolls measured at the same elevation. Their shape varies, but the ones we dated have a thickness of at least 20 cm (Table S1a). Their death is related whether to a large uplift, or to an important climatic event. To our knowledge, no large and sudden climatic variations were documented in the last 5ka in the Ryukyus. In this study, we document evidence of uplifts that support the scenario of an uplift causing a sudden decrease of the RSL and consequently the death of the microatolls. We investigate here this scenario.

- P/6 You always use inter seismic rates of 1.4 mm/yr, but for the period from 1771-2018 you use 3.0 mm.yr without much explanations of how you get to these rates. You even name them “arbitrary”. Since this is key for some of your calculations latter it would be good to understand how you get to those estimates and what their uncertainty and the impact thereof is.

Sometimes I wonder wether your work simply needs to be described with more details and requires a longer journal.

We thank the reviewer for this insightful comment. The interseismic rate of 3.0 mm/yr used for the 1771–2018 period is indeed based on recent RSL observations over the past two decades in the southern Ryukyus (Debaecker et al., 2023), which we consider better constrained than longer-term estimates for this period. We have revised the text to make this origin clearer and to avoid the potentially misleading term “arbitrary.”

We also clarified in the Methods section that the uncertainty in this rate, as well as the other interseismic RSL rate we use, affects the calculation of the amplitude of the event, via the formula we detailed there. This localized influence does not significantly alter the broader interpretation or timing of the seismic cycles we present.

Regarding the reviewer’s final point, we respectfully note that the level of detail currently provided is consistent with the format and expectations of this journal. While further elaboration is always possible, we believe that the current manuscript is appropriate for the scope and format of Nature Communications, and sufficiently rigorous to support our main conclusions.

Minor points

- Fig. S2b is missing axis labels (please also check for other axis labels)

Axis label is indicated in the legend box for all graphs. We choose to indicate it there for a better readability.

References:

Debaecker, S., Feuillet, N., Satake, K., Sowa, K., Yamada, M., Watanabe, A., ... & Shen, C. C. (2023). Recent relative sea-level changes recorded by coral microatolls in Southern Ryukyus Islands, Japan: implication for the seismic cycle of the megathrust. *Geochemistry, Geophysics, Geosystems*, 24(6), e2022GC010587.

- Fariás, M., Vargas, G., Tassara, A., Carretier, S., Baize, S., Melnick, D., & Bataille, K. (2010). Land-level changes produced by the M w 8.8 2010 Chilean earthquake. *Science*, 329(5994), 916-916.
- Kawana, T. (1987). Holocene crustal movement in and around the Sekisei Lagoon in Okinawa Prefecture, Japan. *Earth Monthly*, 9, 129-34.
- Kawana, T. (1989). Quaternary crustal movement in the Ryukyu Islands. *Earth Monthly*, 11, 618-30.
- Meltzner, A. J., Sieh, K., Chiang, H. W., Wu, C. C., Tsang, L. L., Shen, C. C., ... & Briggs, R. W. (2015). Time-varying interseismic strain rates and similar seismic ruptures on the Nias–Simeulue patch of the Sunda megathrust. *Quaternary Science Reviews*, 122, 258-281.
- Philibosian, B., K. Sieh, J.-P. Avouac, D. H. Natawidjaja, H.-W. Chiang, C.-C. Wu, H. Perfettini, C.-C. Shen, M. R. Daryono, and B. W. Suwargadi (2014), Rupture and variable coupling behavior of the Mentawai segment of the Sunda megathrust during the supercycle culmination of 1797 to 1833, *J. Geophys. Res. Solid Earth*, 119, 7258–7287, doi:10.1002/2014JB011200.
- Sieh, K., Natawidjaja, D. H., Meltzner, A. J., Shen, C. C., Cheng, H., Li, K. S., ... & Edwards, R. L. (2008). Earthquake supercycles inferred from sea-level changes recorded in the corals of west Sumatra. *Science*, 322(5908), 1674-1678.
- Tan, F., Horton, B. P., Ke, L., Li, T., Quye-Sawyer, J., Lim, J. T., ... & Meltzner, A. J. (2024). Late Holocene relative sea-level records from coral microatolls in Singapore. *Scientific Reports*, 14(1), 13458.
- Taylor, F. W., Frohlich, C., Lecolle, J., & Strecker, M. (1987). Analysis of partially emerged corals and reef terraces in the central Vanuatu arc: Comparison of contemporary coseismic and nonseismic with Quaternary vertical movements. *Journal of Geophysical Research: Solid Earth*, 92(B6), 4905-4933.
- Yamaguchi, T. (2016). A review of coral studies of the Ryukyu Island Arc to reconstruct its long-term landscape history. *Coral Reef Science: Strategy for Ecosystem Symbiosis and Coexistence with Humans under Multiple Stresses*, 55-63.

Reviewer #1 :

1. Erosion and sea-level tendency

It is now much clearer with preserved/eroded HLG/HLS illustrated in Figure S2b. The interseismic rates are also now inferred using the authors' best estimate of the interseismic RSL trend using the longest, best-preserved record from NAG3_18H. However, eroded HLGs were still used to quantify coseismic uplift estimates, which we draw caution to.

While it is true that eroded HLG/HLS have been used to estimate RSL changes (as cited in the examples by Sieh et al. 2008 and Meltzner et al. 2015), the focus for this is to quantify interseismic RSL rates. The premise for doing so is that the coral only experiences a diedown once it grows close enough to sea level that a temporary negative sea-level anomaly/ drop in sea level can cause a diedown. Therefore, the HLG immediately preceding a diedown (i.e., highest point on the coral before a diedown) best approximates HLS, even if the highest points are eroded. Additionally, the pre-diedown HLG is less affected by the magnitude of a given diedown event (which can be influenced by the severity of a sea-level anomaly), compared to the post-diedown HLS. (See Meltzner et al., 2010, section 3.3.2)

We understand the reviewer's concern. We agree with the reviewer that our microatoll slices clearly show signs of erosion.

We are aware that there must be an erosion-related bias, but we propose conducting the exercise for long and/or important periods.

As we explained, we use rates from all points, on the basis that there are no significant differences between rates estimated from preserved and eroded records. Although this observation relies solely on one measure (NAG3_18H sea-level increase rate), it is also supported by our hypothesis that, at first order, erosion occurred continuously and from above. The effect of erosion may have minimised the decrease rate estimated here, but to our knowledge, this is the best approximation our data can offer.

In Table S2a, we distinguish these RSL rates from those calculated from pre-diedown HLG or from preserved HLS bands. We state that the latter two are the best constrained. Additionally, we clarify that RSL rates that are estimated either partially or entirely from eroded HLS/HLG bands are considered "apparent", as they may not accurately reflect the true RSL trend value.

In the examples regarding Sieh et al. 2008 and Meltzner et al 2015, coseismic uplifts were only estimated where there were preserved HLS points to indicate how far the corals died down to. Without a preserved HLS/ diedown, it is not clear if there truly was a diedown that later got eroded, or that there was no diedown to begin with.

We agree that there are uncertainties in quantifying the coseismic uplift when the upper surface of microatolls shows signs of erosion. For several events we have identified a **preserved** HLS band that marks the extent of the coral diedown, and we use the highest point preceding the diedown, eroded or not. We now specify it in the **Method section**. "We estimated the amplitude of the identified die downs using the difference in elevation between the highest HLG point (eroded or not) preceding the diedown⁶⁵, and of the HLS point (eroded or not) marking the die down itself."

We also considered the largest diedowns to be tectonic events, focusing only on large RSL drops of at least 10 cm (Zachariasen, 1998; except for event E in NAG6_18Q which is correlated with the death of NAG6_21C, and event O in

NAG3_18H for which we infer that the 9 cm amplitude is significantly underestimated). As stated by the reviewer, these events may be exaggerated by temporary sea-level anomalies or erosion, but given the large amplitude of the RSL drop, we interpret that most of it is related to tectonic motion.

We recommend that events G and H should be assigned a lower confidence index due to the lack of preserved HLS and lack of agreement with coeval records. During the timing of events G and H when NAG_M was inferred to have died down, NAG6_18H recorded continued upward growth, even though NAG6_18H was at a higher elevation. The two coeval coral records do not seem to agree, even considering the +/- 7 cm HLS variability, so it seems possible that events G and H in NAG_M reflect erosion horizons, rather than diedowns. The concentric nature of this “step-down” marking event G is also not clearly illustrated in the field photo of NAG_M provided in the rebuttal:

Therefore, we recommend that event G is downgraded from confidence index of 5 (the most confident) to a lower confidence, and hedge the wording regarding the “major sea-level decrease” and “second sudden sea-level decrease” in supplementary lines 142-144.

This is unlike event D, where coeval diedowns were recorded in both NAG6_18Q and NAG6_21C (despite NAG6_21C not having a preserved diedown); and event E, where NAG6_21C died down completely around the same time when NAG6_18Q recorded a diedown (despite there being no preserved HLS in NAG6_21C).

We agree with the reviewer that the slices, HLS and photographs presented in the paper do not allow us to confirm with certainty that the event G is a major sea-level decrease and that the coral died in two steps (partly after G and entirely after H). We acknowledge that the provided photograph is sufficiently convincing, despite being the best available.

While it is clear that both corals did not respond equally to event G, as the reviewer noted, event H corresponds to a major change in the morphology of NAG6_18H (Colony B) that was growing in a higher position. The RSL drop (and inferred coseismic uplift) that caused the death of NAG_M brought NAG6_18H to its HLS, enabling it to record RSL, although those bands are now eroded.

We emphasise that event H is also coeval with the death of plateau P9, which was dated using a third coral, NAG6_21A (Table S1b, Fig. 3). However, we cannot explain the difference in elevation between the lower lobe of NAG_M and NAG6_18H. We consider it highly unlikely that this event is related to erosion, or to a climatic diedown. However, as we cannot firmly demonstrate this, we have downgraded the confidence index of event G from 5 to 4 and removed the adjective “major” to qualify this RSL drop.

Supplementary text lines 117-119: the two emergence events inferred are from eroded parts of NAG_B – there may not have been a diedown as the HLS is not preserved.

The emergence event recorded at 301 years B.C. in the left radius of NAG_B is inferred from a preserved HLS band. It is therefore interpreted as a diedown. We agree with the reviewer that, when examining a single slice, a small drop in the RSL identified from eroded growth bands alone may not necessarily indicate a diedown. However, we have two slices from opposite sides of the microatoll that both show the drop of the RSL at ~264 years BC. It is highly unlikely that erosion alone or predominantly affected the entire uppermost external part of the microatoll.

Figure S2a: The upward growth of a coral does not necessarily indicate sea-level rise tendency (as shown by black lines in Figure S2a indicating “apparent RSL increase”). The coral could still be catching up to sea level, during which upward growth reflects the coral growth rate, not sea level. Instead, it is the stepping upwards of successive concentric rings/ diedowns that indicate RSL rise, not the increase in elevation of successive growth bands. Similarly, RSL fall should be interpreted as successively lower concentric rings/ HLS, not successively lower growth bands. We agree that the figure and legend were confusing. Our intention was to indicate the periods of free growth indicating a RSL increase (either sudden or continuous, although we cannot always estimate a rate on them) and we made a mistake by including the period of free growth during the initial growth period of a microatoll. To avoid any further confusion, we have deleted the black and blue arrows in Figure S2a that corresponded to the initial growth period of the coral and apparent decrease in RSL, respectively. However, we have kept the RSL increase periods inferred from preserved HLS bands in NAG_B and NAG3_18H, and updated the legend. In Table S2a, we deleted the apparent RSL increase rates inferred from HLG points only, as we agree with the reviewer that these rates were not consistent.

Supplementary text lines 138-139: the words ‘first HLS’ were removed, and eroded HLG/HLS shown in Figure S2b, but the text and interpretation did not change (i.e., erosion is still interpreted as RSL fall, as indicated by the blue arrow in Figure S2b).

Supplementary text lines 180-181: similar comment to the above; RSL decrease of -1.4 mm/yrs is inferred from eroded HLG, but HLS are not preserved

Supplementary text line 191: similar comment to the above; RSL decrease is inferred from eroded HLG, but HLS are not preserved

The blue arrows in Figure S2b illustrate the inferred impact of erosion on uplift estimation.

As explained in the first response, we estimated RSL trends from periods including eroded points, but we now acknowledge that these estimations are not well-constrained.

We now refer to these estimations as “apparent” RSL decrease or increase periods.

2. Offset of NAG3_21E and NAG_B not explained by HLS variability applied

It is unclear from the text how the HLS variability of +/- 7 cm was derived from two corals with coeval diedowns. Which corals / diedowns was this based on?

This observation is based on NAG6_21C and NAG6_18Q RSL record, which we now specify in the Method section: “When a drop was observed in several slices (e.g. NAG6_21C and NAG6_18Q, Fig. 3), we observe an average range of 15cm between measurements, leading to an estimated uncertainty of ±7cm.”.

In lines 102-104 of the main text, the current explanation for why NAG3_21E is lower than NAG_B seems to suggest that this offset could be explained by natural variability in the coral HLG/HLS. However, the elevation difference between the preserved HLS of the two corals is nearly 35 cm at 301 yrs BC based on figure 3, which is larger and seems to contradict the the +/- 7 cm applied as the HLS variability. Could NAG3_21E have slumped? Or NAG_B be ponded at a higher elevation?

±7cm of variability is inferred from the elevation difference of RSL records from NAG6_21C and NAG6_18Q. We focused our estimation on these coeval corals as they recorded two RSL drops and grew close to each other, which discard any potential influence of geomorphologic features as suggested for NAG_M and NAG6_18H, and for NAG3_21E and NAG_B.

This estimation relies only on two corals, so while it is the best our data can provide, it remains uncertain. We considered it important to propose that variability of the coral record could be greater than what we initially estimated.

Ponding of NAG_B coral cannot be ruled out, and we acknowledged this possibility as suggested by the reviewer.

“Although NAG3_21E is coeval with NAG_B, it is 30-40 cm lower and did not record RSL drops of similar magnitude. Variations of tens of centimeters in altitude among corals of the same generation have been observed elsewhere, reflecting the variability of the coral record²⁸. Ponding of NAG_B coral could explain such differences, although we did not observe any traces of past geomorphic features supporting this hypothesis.”

Alternatively, is it possible that NAG3_21E was growing too deep compared to NAG_B, such that the uplift event that killed NAG_B (event Q) was not large enough to kill NAG3_21E? If so, then this would provide an upper limit on the amount of uplift for event Q. This would require that the 301 yrs BC and 329 yrs BC inferred diedowns were not diedowns. From figure S2a, the 342 yrs BC diedown is not shown on NAG3_21E, and the 301 yrs BC diedown in NAG3_21E is also not as clear as that in NAG_B.

As the reviewer stated, the possibility that NAG3_21E grew deeper compared to NAG_B implies that it did not record RSL until ~256 B.C.

We agree that the 342-years B.C. diedown observed in NAG_B is not evident in NAG3_21E in Fig. S2a, since there is no preserved HLS band and the RSL record shows no drop of the RSL, may this be related to erosion or not. A preserved HLS band is interpreted at 301 years B.C. in NAG3_21E although it is not the clearest. However, at 329 years B.C., the growth band clearly shows a preserved HLS, as there is local upward (and lateral) growth upon it (Figure S2a). This rules out the hypothesis that NAG3_21E was initially ball-shaped until ~256 B.C.

Furthermore, NAG_B died at the same time as NAG3_18B (Table S1b), which is located at roughly the same elevation and position as NAG3_21E.

The difference in elevation between the two coeval corals could be related to large HLS variability, ponding, compaction or another unidentified process. Unfortunately, in our opinion, we cannot clearly explain it based on our observations.

Minor comments

1. Line 99: Should be “glacial isostatic adjustment” not “glacio-isostatic adjustment”
Corrected

2. Lines 181-183: Please specify which corals were used to infer possible coseismic subsidence (it’s indicated in figure 4 caption as “potential sudden RSL increase event”) but could be described better in the main text. It will also be helpful to add the inferred subsidence event(s) to table S2b.

This subsidence event is inferred from the difference in elevation of NAG6_21C, NAG6_18Q, and NAG6_18H, NAG_M corals, This is already specified in the main text “At the mangrove site (Fig. 3a), NAG6_21C and NAG6_18Q are older than NAG_M and NAG6_18H, yet lie below them. This can indicate a sudden increase in RSL of at least 35 cm, occurring between the RSL records of NAG6_18Q and NAG6_18H, i.e. between 2307 and 2357 years B.C.

We also now mention it in Table S2b, Figs. 3 and 4.

3. Lines 234-236: Consider adding that different glacial isostatic adjustment models could also produce variable RSL predictions.

We have chosen not to go into too much detail about the GIA, since we have established that the GIA alone cannot account for the observed deformations, and that an important part of the RSL signal over the last 5 ka must be related to tectonic process. It is this tectonic process that we focus on here at first order. However, as the reviewer noted, there are different GIA models and we cannot and do not attempt to precisely quantify it.

4. Supplementary text line 119: 26’ +/- 8 yrs BC seems to be a typo
Corrected.

5. Supplementary lines 190-191: NAG6_21C was described as starting to record HLS variations at 2436 years BC, but there was no preserved diedown/ HLS here.

We consider unpreserved HLS/HLG growth band as we refer to the oldest age from which the coral could have recorded RSL variations. We modified the sentence to “it recorded HLS variations at 2436 ± 9 years B.C. at the sooner, with a period of apparent RSL decrease of -1.8 ± 1.4 mm/year.”.

6. Figure 3: Similar to figure 4, also distinguish the confidence index in the vertical lines that indicate diedown events?

Done.

7. Figure 4: Rather than bolding letters with confidence index 5, consider having distinct line styles for the different confidence indices.

Done.

We would like to take this opportunity to point out that we have corrected an error in Figure 4. Between the last event, U, and the present time, we now use the same rate of RSL change as for the rest of the RSL reconstruction: 2.0 ± 0.6 mm/year.

8. Please check that table S2b is consistent with figure 3. For example, events N and O are indicated as >15 cm and > 18 cm in figure 3 and in the main text, but reported as ~15 cm and ~18 cm in table S2b. Please check for other events as well.

Done. We also corrected the amplitude for events D and E.
As explained in a previous response, event O amplitude now ranges between 9 and 18 cm.

9. NAG3_18H, events N and O:

□ It seems odd that the erosion line was not extrapolated to the diedown for event N
In Figure S2a, there are growth bands on both sides of the initial growth part of the coral, up to the growth band dated at 541 ± 14 years BC. Therefore, we considered that there are varying degrees of certainty about the high erosion hypothesis in this area, which is why we only draw it up to 541 years BC.

□ Why is the vertical downward dashed arrow for event O (which indicates the tectonic diedown) drawn from the HLG/ring crest **after** the diedown, not the pre-diedown HLG?

The vertical arrow showing the event indicates the growth band used for dating the diedown. Since some of the pre-diedown HLG are eroded, presumably highly eroded in the cases of events N and O, drawing it from there can be confusing.

We adjusted the length of the vertical arrows to the scale we use consistently for all the other events documented in Figure S2a.

□ Rather than crudely estimating the amount of erosion based on the dashed blue erosion lines drawn, since the uplift amounts are reported and described as minimum uplift estimates anyway, would it be better to simply take the difference between the highest (eroded) pre-diedown HLG and the preserved HLS elevation as the minimum uplift estimate? Alternatively, perhaps extrapolate the pre-diedown HLG from the RSL trend prior to the diedown, similar to the examples cited in the rebuttal for Sieh et al. 2008 and Meltzner et al. 2015?

We thank the reviewer for this remark, and we used the pre-diedown HLG elevation to further estimate the amplitude of each identified RSL drop, including events N and O, and added these reference points to Fig. S2b.

For event O, extrapolating the growth period before the eroded part prior to the diedown, as suggested by the reviewer, leads to an amplitude estimate of 18 cm.

We have updated the values in the supplementary text.

Lines 150-152 : “If the apparent RSL increase period last until 445 ± 12 years B.C., the coral would have recorded a RSL drop of 18 cm (instead of 9 cm, considering the highest eroded HLS/HLG prior to the diedown⁵, Fig. S2b).”

10. The complete die-off of any individual coral microatoll colony may not always require a tectonic uplift event.

By “temporally discontinuous” records, we refer to the fact that even in relatively tectonically stable locations, coral microatoll ages often cluster into age clusters. This

may be due to biological life spans of the coral microatolls, as according to Smithers (2011):

“Large microatolls demonstrate continuity of growth for several centuries, but most microatolls are < 2 m in diameter. This probably reflects the relatively low probability of extended survival in shallow reef environments where corals are most vulnerable to a range of debilitating conditions. Microatolls on the northern GBR surveyed by Scoffin and Stoddart (1978, p.105) averaged 0.5 m in diameter, although other than stating that they “vary in size from a few centimeters to a few meters” no statistical description of the variance was offered.”

Example from Hallmann et al. 2018 with coral microatolls from different sites in French Polynesia clustering in different time periods.

Example from Tan et al. 2024 showing fossil corals in Singapore clustering in time periods, with gaps between age clusters.

We thank the reviewer for clarifying this previous remark. We agree that the death of one or more corals cannot prove the tectonic or climatic nature of an event. This is why we did not apply a 5-confidence index to these interpreted events.

Meanwhile, we have more arguments in favour of a tectonic origin for these clusters (the staircase morphology of the site is consistent with long-term uplift of the coast; there is evidence of coseismic uplift in the coral analysis; there are tsunami deposits; and there have been previously documented past earthquakes). By contrast, we have hardly any arguments in favour of a climatic origin (mostly due to a lack of data). Once again, we agree with the reviewer that this reasoning is not conclusive, but we would like to emphasise that it is more logical to interpret the motion as being due to seismic activity rather than to climate change.

References

Hallmann, N., Camoin, G., Eisenhauer, A., Botella, A., Milne, G. A., Vella, C., ... & Fietzke, J. (2018). Ice volume and climate changes from a 6000 year sea-level record in French Polynesia. *Nature communications*, 9(1), 285.

Meltzner, A. J., Sieh, K., Chiang, H. W., Shen, C. C., Suwargadi, B. W., Natawidjaja, D. H., ... & Galetzka, J. (2010). Coral evidence for earthquake recurrence and an AD 1390–1455 cluster at the south end of the 2004 Aceh–Andaman rupture. *Journal of Geophysical Research: Solid Earth*, 115(B10).

Meltzner, A. J., Sieh, K., Chiang, H. W., Wu, C. C., Tsang, L. L., Shen, C. C., ... & Briggs, R. W. (2015). Time-varying interseismic strain rates and similar seismic ruptures on the Nias–Simeulue patch of the Sunda megathrust. *Quaternary Science Reviews*, 122, 258–281.

Sieh, K., Natawidjaja, D. H., Meltzner, A. J., Shen, C. C., Cheng, H., Li, K. S., ... & Edwards, R. L. (2008). Earthquake supercycles inferred from sea-level changes recorded in the corals of west Sumatra. *Science*, 322(5908), 1674–1678.

Smithers, S. (2011). Microatoll. In D. Hopley (Ed.), *Encyclopedia of Modern Coral Reefs: Structure, Form and Process* (pp. 691–696). Springer Netherlands. https://doi.org/10.1007/978-90-481-2639-2_111

Tan, F., Horton, B. P., Ke, L., Li, T., Quye-Sawyer, J., Lim, J. T., ... & Meltzner, A. J. (2024). Late Holocene relative sea-level records from coral microatolls in Singapore. *Scientific Reports*, 14(1), 13458.

References:

Zachariasen, J. A. (1998). *Paleoseismology and paleogeodesy of the Sumatran subduction zone: A study of vertical deformation using coral microatolls* (Un-published doctoral dissertation). California Institute of Technology.

Reviewer #5 (Remarks to the Author):

The manuscript "Evidence for megathrust earthquakes (...)" provides clear evidence for paleoseismic activity along the Ryukyu trench that can reasonably be attributed to megathrust earthquakes. The interpretation of the coral record seems robust. However, the paper is poorly written and the figures are of poor quality. I recommend minor revisions to improve the quality of the manuscript.

Line 26: replace "infer" with "imply". Write "Furthermore, historical and geological studies show evidence of great tsunamis."

Done.

But perhaps indicate when and where, as the sentence, as is, is vague.

The main goal of our abstract is to present the context in a general way, while the evidence of great tsunamis is detailed in the first paragraph of the main text.

Line 27. "Here, we use". Use present tense and active voice.

Corrected.

Line 28: "relative sea level". Singular.

"We use fossil microatolls in Ishigaki Island to reconstruct relative sea level in the Holocene. The coral record reveals several emergence episodes clustering 5-to-4 and 3-to-2 thousand years ago (ka) compatible with rapid uplift from megathrust earthquakes."

Line 33: remove "would". "The paleoseismic behavior implies a strong seismic hazard for the upcoming centuries. The devastating 1771 Meiwa earthquake and associated tsunami may mark the onset of a seismic supercycle."

We revised the sentences in the abstract following the reviewer's suggestions.

Line 36: remove "," in the middle of the sentence. Remove "strong".

Done.

Line 43: "evidence". Singular. Remove "," before "and".

Done.

Throughout the text: There are problems with virtually every sentence of the manuscript. I will stop correcting, but this is a major issue. There are many useful tools to correct and improve the writing. See DeepSeek, ChatGPT, Mistral, and the like. The language and writing must be seriously improved.

We have carefully proofread and corrected the text.

Line 49: "evidence" -> "record". "To this goal" -> "Toward this goal"

Done.

Line 97: "motion". Singular.

Done.

Line 98: "motions"->"displacement".

Done.

Line 215: "2 ky apart"

Line 216: "over the last 6 kr".

The abbreviations "kr" and "ky" are uncommon and mostly used in astrophysics. We therefore use the standard "ka" notation in the revised manuscript.

Line 238: Permanent or irrecoverable deformation is often attributed to splay faulting and folding in the accretionary prism. Ishagaki is a forearc high built by such deformation. See a recent review:

Qiu Q. and S. Barbot, "Tsunami excitation in the outer wedge of global subduction zones". *Earth-Science Review*, 2022.

Yes there are splay faults in the accretionary wedge, and we agree that it is difficult to assess the source of permanent deformation here.

According to the megathrust model of the Ryukyus presented in the mentioned paper, the splay faults are located very close to the trench and the splay faults are located very close to the trench and connect to the megathrust at a depth of about 10 km, that is, far from the islands. Activating such small faults would induce a highly localized uplift directly above the structure, resulting in negligible or near-zero displacement at the islands.

From Qiu and Barbot (2022).

Figure 1: add contours of slab2.0 depth. **Done.**

Is it "Ruykyu" or "Ryukyus"? Probably singular.

We use plural form "Ryukyus" when referring to it as a subject.

Use similar color themes for the main map and the two insets.

We kept the black-and-white theme for the two insets since they are meant for location information only, whereas the colored main map contains the main information of this figure.

Fix the width of the black rectangles on the side. Some have a black frame, others don't. **Done.**

Date the aseismic events. **Done.**

Figure 2: can probably be split into two figures.

We chose to keep a single figure to better highlight the correspondence between the corals location and their vertical distribution.

Figure 2c: "Distance along plot (km)" What is "plot". Is it a plot of land? Or plot as in "figure"? Perhaps find a more suitable axis label.

Corrected.

All the figures are pixelated. Hopefully, vector-graphic figures will be provided eventually.

We improved the image quality.

References

Qiu, Q., & Barbot, S. (2022). Tsunami excitation in the outer wedge of global subduction zones. *Earth-science reviews*, 230, 104054.

We thank the authors for taking the time and effort to address the comments raised. We find that the manuscript is now much clearer in its assumptions and inferences, providing important data to enhance understanding of seismogenic nature of the Ryuku subduction zone. It is especially interesting that the clusters of high-confidence abrupt uplifts coincided with the ages of tsunami deposits, with lack of both coral data and tsunami data between the inferred supercycles. We only have a few remaining minor comments and recommend publication thereafter.

Main text

1. Line 101: please specify over what time period the quoted GIA rate is referring to.
Done.

2. There are places where there seems to be missing line spacing between paragraphs. Please check to ensure that the spaces are added accordingly (e.g., between lines 102–103, 135–136...)
Done.

3. Line 405: Here it says you use Zach-3 method, but in table S2a the footnote mentions you use the Zach-1 method. Please check to make sure it is consistent.
We precise in the Method section that we use Zach-3 method for RSL reconstruction, and Zach-1 method for slice analysis (as Zach-3 method could not be applied in most slices).

4. Line 436: Earthquakes inferred from within the coral slices are assigned with confidence rating of 5. While we appreciate that the authors downgraded the confidence ranking of event G to grade 4 to address our earlier comment, the confidence rating now is inconsistent with the description in the text (because event G is inferred from the slab but is assigned grade 4 and not 5). We suggest adding a new confidence rank to differentiate between diedowns inferred from *preserved* HLG/HLS (ranking of 6) in slabs versus those inferred from *eroded* HLG/HLS in slabs (ranking of 5).

We thank the reviewer for this suggestion. We applied it and modified the definition of the confidence index as well as Table S2b.

In the method section, we write : “Earthquakes identified within coral slices receive a confidence index of 6 when affected growth bands are preserved from erosion, and of 5 otherwise.”

5. Figure 1: panel labels are missing (e.g., 1a, 1b)
Figure 1 and its caption have been modified accordingly.

6. Figure 4: where are the dotted lines?
We were referring to dashed black lines of the RSL history reconstruction. We corrected this mistake.

7. Figure 4: we appreciate the attempt to address our earlier comment by introducing different line thicknesses for different confidence rankings. However, the different line thicknesses are more difficult to read than the bolded letters initially. Could you use different line symbols (e.g., dotted, dashed, solid, twodash) to represent different confidence rankings, and add a legend for this?
Done.

8. Figure 4: please arrange the red line for event R to be on top of the black vertical line as now it is difficult to see the red line.

Done.

9. Figure 5: please indicate in the caption which corals/ uplift events were used to constrain the elastic dislocation models. Maybe I missed this but was this mentioned explicitly in the text? If not, please add a line to clarify this.

Done.

We now specify in the caption : The colored points with gray error bars are inferred uplift events with their associated uncertainties (see Table S2b).

Supporting text

1. Line 209: change “at the sooner” to “at the soonest”

Done.

2. Table S2b: to make it easier for the reader, please add to the description in the table header that a confidence ranking of 5 is the best, so that the reader does not have to look for this in the text.

Done.